**Registered report**

# Value-free random exploration is linked to impulsivity

**Magda Dubois** [1,2] ✉ **& Tobias U. Hauser** [1,2]

Deciding whether to forgo a good choice in favour of exploring a potentially more rewarding alternative is one of the most challenging arbitrations both in human reasoning and in artificial intelligence. Humans show substantial variability in their exploration, and theoretical (but only limited empirical) work has suggested that excessive exploration is a critical mechanism underlying the psychiatric dimension of impulsivity. In this registered report, we put these theories to test using large online samples, dimensional analyses, and computational modelling. Capitalising on recent advances in disentangling distinct human exploration strategies, we not only demonstrate that impulsivity is associated with a specific form of exploration−value-free random exploration−but also explore links between exploration and other psychiatric dimensions.

## Protocol registration

The Stage 1 protocol for this Registered Report was accepted in principle on 19/03/2021. The protocol, as accepted by the journal, can be found at https://doi.org/10.6084/m9.figshare.14346506.v1.

That human and non-human animals differ in their impulsivity is one of the earliest and most influential observations of inter-individual differences[1]. Impulsivity is often described as 'acting without thinking' and is traditionally assessed using self-report questionnaires[1,2]. It is a broad and heterogenous construct[1,3–5] whose relevance not only comes from the observation of a substantial variation among a 'healthy' population, but also its importance in psychiatry. More recently, highly influential reinterpretations of psychiatric disorders have proposed impulsivity as an overarching symptom conglomerate encompassing multiple psychiatric disorders, such as addictions, manias, and−almost archetypically−attention-deficit/hyperactivity disorder[6] (ADHD).

Despite the relevance of impulsivity, relatively little is known about the neurocomputational mechanisms that underlie this trait. Impulsivity has been linked to imbalances in catecholamine functioning[7–13] but how these imbalances affect behaviour remains unknown. One suggestion about the function and role of impulsivity is as elegant as it remains speculative. Using simulations, Williams and Taylor[14] suggested that impulsivity is characterised by heightened exploration behaviour, meaning that impulsive participants are more

likely to forego certain high valued outcomes for the benefit of exploring lesser known choice options that may hide even higher valued outcomes. Even though such a behaviour could be detrimental for an impulsive individual, the authors demonstrated that such behaviour could be of great benefit on a societal level[14]. Several theoretical accounts have embraced this concept and demonstrated how increased exploration can arise due to catecholamine imbalance[6,12,15–17], how this may be implemented neurally[12], and how an excessive exploration can explain other behaviours observed with impulsivity[12,15], such as delay discounting[18–20].

Outside of theoretical work, however, empirical evidence so far is still sparse. Only the first few studies using ADHD patients[21] or looking at ADHD symptoms in youths[22] have found empirical evidence for heightened exploration in impulsivity using computational methods. Their insights are particularly limited as recent work on exploration-exploitation trade-offs has shown that exploration itself is not a homogenous concept. Empirical work in healthy participants has clearly demonstrated that humans deploy multiple different exploration strategies and that these strategies differ in sophistication and

[1]Max Planck UCL Centre for Computational Psychiatry and Ageing Research, London, UK. [2]Wellcome Centre for Human Neuroimaging, University College London, London, UK. ✉e-mail: magda.dubois.18@ucl.ac.uk

computational demand[23–25]. In particular, one can distinguish between sophisticated and complex exploration strategies, such as upper confidence bound[26] (UCB), which take the expectation as well as the uncertainty of all possible choice options into account[25,27–30], versus heuristic strategies, which require relatively less computation. Amongst the latter, we and others have found evidence for novelty exploration, a strategy that focuses only on completely unknown choice options[23,29,31]. In addition, there is evidence for exploration strategies that deliberately omit all existing knowledge to choose all options equally likely, termed value-free random exploration[23]. Such 'value-free' random exploration ignores all available information (i.e., expectation and uncertainty of choices) and thus forgoes any costly computation. This is in contrast to more refined 'value-based' random exploration that adds stochasticity during choice value computation or directed exploration, which biases choice towards information gain[24,25]. Even though such mechanisms are suboptimal, their low computational demand has made them popular in artificial intelligence[32] (i.e., $\epsilon$-greedy) and we have found clear signatures in humans[23].

Whether exploration mechanisms (and if so, which ones) are altered in impulsivity remains unknown. However, this is of critical importance as recent animal[33] and human[23] work have demonstrated a specific role of catecholamine functioning in different forms of exploration. In particular, we have shown that only value-free random exploration is sensitive to noradrenaline (but not dopamine) functioning[23], which is a neurotransmitter that has repeatedly been suggested to be critical in impulsivity disorders[6,7,12,17,34–38].

In this study, we put the large body of theoretical work to test and exploited the recent advances in the exploration literature to empirically investigate the link between impulsivity and exploration. Here, we investigated impulsivity as a broad spectrum across the general population and also with respect to a more specific ADHD-related impulsivity. We used a preregistered, dimensional approach via large sample online testing to provide a clear answer. We advanced on a method that has recently proven the most promising for detecting meaningful mechanisms underlying psychiatric symptoms[39–42]. We ran a big data dimensional study using our recently developed exploration task[23], which was designed to disentangle the exploration strategies that have been put forward in the literature, and which allowed us to provide an answer to whether exploration behaviour is linked to impulsivity. To determine not only whether, but also which, exploration mechanism predicts impulsivity, we made use of computational modelling. Supported by previous findings that impulsivity is associated to increased avoidance of mental effort[43] and that ADHD is associated to increased value-free random exploration[22], we tested our hypothesis that it is specifically value-free random exploration (captured by our model parameter $\epsilon$) which correlates with impulsivity measures (cf. Table 1), therefore determining which of these mechanisms is impaired in impulsive participants. In addition, our data allowed us to explore how exploration impairments may be linked to other psychiatric domains (e.g., to OCD and other avoidance of uncertainty disorders[44,45]; to depression, anxiety and anhedonia[46–48]) using data-driven methods.

## Results

To capture different forms of exploration, we used our previously lab-validated Maggie's Farm task[23] (cf. Fig. 1), which is essentially a 3-armed variant of the Horizon task[24]. In this task, participants had to choose which bandit (depicted as trees) to draw a sample from (i.e., pick an apple) in order to maximise a sum of reward (represented by the apples' size; Fig. 1a). To help them with their decision, at the beginning of each trial, participants had some information about how good each bandit was in the form of 'initial samples' (i.e., apples that have been picked before). Bandits carried either a lot, some, or no prior information (i.e., 3, 1 or 0 initials samples) and had either a

standard or a low reward mean. In effect, there were 4 different types of bandits: the certain-standard bandit (standard mean, 3 initial samples), the standard bandit (standard mean, 1 initial sample), the novel bandit (standard mean, 0 initial samples) and the low-value bandit (low mean). A real-life example would be having to choose between four different ice-cream flavours in an Italian city: chocolate, which you have enjoyed 3 times in the past, Toblerone, which you have enjoyed once in the past, hibiscus, which you have never tried, and spinach, which you have disliked once in the past. The decision horizon (cf. below), represents how often you will come back to this exact same ice-cream shop (e.g., the number of vacation days left, assuming you have once ice-cream per day). On each trial, 3 out of those 4 bandit types were used. In the analysis, the bandit with the highest mean reward of prior samples (either 1 or 3) is referred to as the 'high-value bandit'.

This task allowed to distinguish between complex exploration strategies and exploration heuristics, namely, value-free random exploration and novelty exploration (cf. Methods for detail). We manipulated the number of prior samples and the rewards of the bandits. This allowed us to capture complex exploration strategies, because they take expected values and the uncertainty of the expected values into account. Value-free random exploration is a computationally very light heuristic that does not take any prior knowledge into account, de facto choosing randomly between options, even those known to be bad (e.g., associated to a low reward prior sample). The low-value bandit is thus a signature of such a heuristic and therefore allows quantification of its contribution. Similarly, the novel bandit allows us to capture novelty exploration, a heuristic which targets entirely novel options.

To promote and assess exploration, we manipulated the number of choices per trial (i.e., decision horizon; Fig. 1b), similarly to the Horizon Task[24]. Participants could perform either one draw, encouraging exploitation (short horizon condition), or six draws, encouraging more substantial explorative behaviour (long horizon condition) as in the latter condition, the newly gained information could be subsequently exploited. Going back to the ice-cream example, knowing that you will come back to the same place many times will encourage you to explore different flavours (i.e., other than chocolate), as it can help guide your future choices. In the analysis, if not stated otherwise, we compared the short horizon's single draw to the long horizon's first draw in alignment with previous studies using the same manipulation[23,24]. All tests were two-tailed. Detailed statistics for all measures can be found in Supplementary Table 2.

### Step 1.1. Are exploitation and exploration horizon-modulated? Yes

To assess whether the horizon manipulation promoted exploration, we analysed which bandit participants chose in the long (versus short) horizon condition. We found, as hypothesised, that participants chose bandits with a lower expected value (computed as the mean of the bandits' initial samples) in the long horizon compared to the short horizon, a sign of increased exploration in the condition where they could benefit from it (expected value of chosen bandit: Wilcoxon signed-rank two-tailed test: $V = 110057$, $p < 0.001$, Wilcoxon effect size: $r = 0.265$; Supplementary Fig. 3a). Further analysis revealed that this was driven by multiple behavioural shifts. We found a reduced frequency of picking the high-value bandit in the long horizon ($V = 157079.5$, $p < 0.001$, $r = 0.797$; Fig. 2a; Hypothesis 1.1.a. in Table 1), showing that participants forego the option with the best expected outcome. We found that this exploration was goal-directed, with participants choosing bandits they knew less about (lower number of initial samples, i.e., more informative) in the long horizon (number of initial samples of chosen bandit: $V = 160109.5$, $p < 0.001$, $r = 0.796$; Supplementary Fig. 3b). Concretely, they increasingly chose both the low-value bandit ($V = 34420$, $p < 0.001$,

## Table 1 | Design Table

| | Question | Hypothesis | Sampling plan (e.g., power analysis) | Analysis Plan | Interpretation given to different outcomes | Outcome |
|---|---|---|---|---|---|---|
| Analysis Step 1: task effect replication | Are exploitation and exploration horizon-modulated? | 1.1.a: Less exploitation in the long horizon: High-value bandit frequency: SH > LH 1.1.b: More value-free random exploration in the long horizon: Low value bandit frequency: SH < LH 1.1.c: More novelty exploration in the long horizon: Novel bandit frequency: SH < LH | Pilot data lowest effect size: 0.410. To detect with 95% power, a similar effect size requires a sample of $N = 83$. | Paired samples t-test (or Wilcoxon signed-rank test if the Shapiro normality assumption is violated). | 1.1.a: No effect: no evidence that our horizon manipulation modulated overall exploitation. Opposite effect: overall exploitation is increased in the long horizon. 1.1.b: No effect: no evidence that our horizon manipulation modulated value-free random exploration. Opposite effect: value-free random exploration is increased in the short horizon. 1.1.c: No effect: no evidence that our horizon manipulation modulated novelty exploration. Opposite effect: novelty exploration is increased in the short horizon. | 1.1.a: Hypothesis confirmed: Less exploitation in the long horizon. 1.1.b: Hypothesis confirmed: More value-free random exploration in the long horizon. 1.1.c: Hypothesis confirmed: More novelty exploration in the long horizon. Answer to research question: Yes, exploitation and exploration are horizon-modulated. |
| | Is exploration beneficial for participants? | 1.2.a: Lower initial reward in the long horizon: Reward of first sample: SH > LH 1.2.b: Higher reward overall in the long horizon: Reward averaged over samples: SH < LH | Pilot data lowest effect size: 0.835. To be detected with 95% power, a similar effect size requires a sample of $N = 21$. | | 1.2.a: No effect: no evidence that participants sacrificed a higher initial outcome in the long horizon. Opposite effect: participants optimised initial reward in the long horizon. 1.2.b: No effect: no evidence that participants took advantage of the information gained in the long horizon. Opposite effect: information gain negatively impacted reward. | 1.2.a: Hypothesis confirmed: Lower initial reward in the long horizon. 1.2.b: Hypothesis confirmed: Higher reward overall in the long horizon. Answer to research question: Yes, exploration is beneficial for participants. |
| | Do participants use exploration heuristics? | 1.3: Average BIC score: complex model + $\epsilon$ + $\eta$ > other models | Pilot data effect size: 1.304. To be detected with 95% power, a similar effect size requires a sample of $N = 10$. | | 1.3: No effect: no evidence that participants combine complex models with heuristics. Opposite effect: Participants are not using complex models with heuristics. | 1.3: Hypothesis confirmed: The average BIC score was higher for the complex models with both heuristics compared to other models. Answer to research question: Yes, participants use exploration heuristics. |
| | Are exploration heuristics used more in the long horizon? | 1.4.a: Value-free random exploration is used more in the long horizon: $\epsilon$-greedy parameter: SH < LH 1.4.b Novelty exploration is used more in the long horizon: Novelty bonus $\eta$: SH < LH | Pilot data lowest effect size: 0.446. To be detected with 95% power, a similar effect size requires a sample of $N = 71$. | | 1.4.a: No effect: no evidence that $\epsilon$ was modulated by the horizon. Opposite effect: value-free random exploration is increased in the short horizon. 1.4.b: No effect: no evidence that $\eta$ was modulated by the horizon. Opposite effect: novelty exploration is increased in the short horizon. | 1.4.a: Hypothesis confirmed: Value-free random exploration is used more in the long horizon. 1.4.b: Hypothesis confirmed: Novelty exploration is used more in the long horizon. Answer to research question: Yes, exploration heuristics are used more in the long horizon. |
| Analysis Step 2: impulsivity | Is impulsivity linked to value-free random exploration? | 2.1: Value-free random exploration is positively associated to BIS: $\epsilon$-greedy parameter and low-value bandit frequency correlates positively with the BIS total score. | Previous study[22] correlation for a similar measure: R = 0.26. To be detected with 95% power, a similar effect size requires a sample of $N = 190$. | Bivariate and partial (correcting for age and IQ) Pearson correlation and a repeated-measures ANOVA with within factor horizon and between participants variable [impulsivity/ADHD-symptoms] | 2.1: No effect: no evidence for an association between value-free random exploration and general impulsivity as measured by the BIS. Opposite effect: Value-free random exploration is negatively associated to BIS. | 2.1: Hypothesis confirmed: Value-free random exploration is positively associated to BIS. Answer to research question: Yes, impulsivity is linked to value-free random exploration. |
| | Are ADHD symptoms linked to value-free random exploration? | 2.2: Value-free random exploration is positively associated to ASRS: $\epsilon$-greedy parameter and low-value bandit frequency correlates positively with the ASRS total score. | | | 2.2: No effect: no evidence for an association between value-free random exploration and ADHD as measured by the ASRS total score. Opposite effect: Value-free random exploration is negatively associated to ASRS total score. | 2.2: Hypothesis confirmed: Value-free random exploration is positively associated to ASRS. Answer to research question: Yes, ADHD symptoms are linked to value-free random exploration. |

Summary of preregistered hypotheses from our Stage 1 Registered report (the full protocol can be found at: https://doi.org/10.6084/m9.figshare.14346506.v1) with an additional 'outcome' column describing the observed effect. SH: Short horizon condition, LH: Long horizon condition.

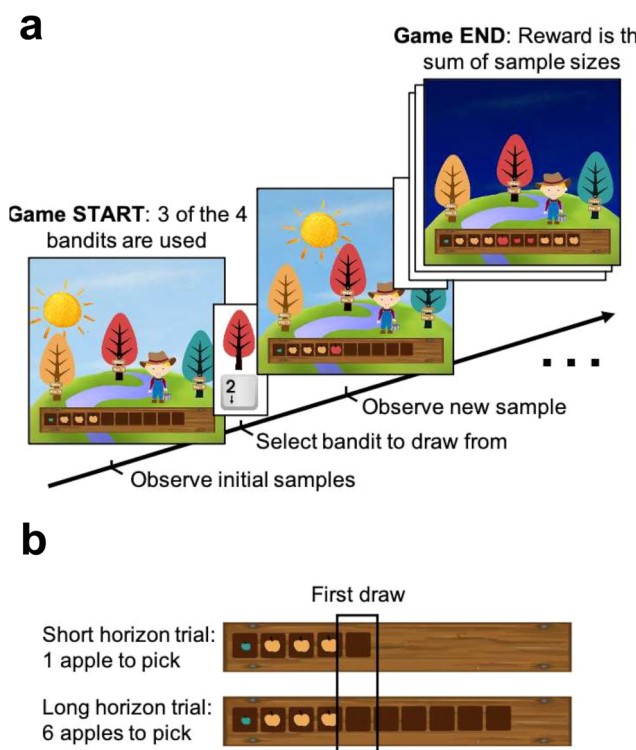

**Game END:** Reward is the sum of sample sizes

**Game START:** 3 of the 4 bandits are used

Observe new sample
Select bandit to draw from
Observe initial samples

**First draw**

Short horizon trial: 1 apple to pick

Long horizon trial: 6 apples to pick

**Fig. 1 | Exploration task.** In the Maggie's farm task, participants had to choose from three bandits (depicted as trees) to maximise their overall reward. The rewards (apple size) of each bandit followed a normal distribution with a fixed sampling variance. **a** At the beginning of each trial, participants are provided with some initial samples (number varied depending on the bandits present on that trial) on the wooden crate at the bottom of the screen and participants had to select which bandit they want to sample from next. **b** Depending the condition, they can either perform one draw (short horizon) or six draws (long horizon). The empty spaces on the wooden crate (and the position of the sun) indicate how many draws they have left. Image adapted from our previous study[23].

$r = 0.425$; Fig. 2a; Hypothesis 1.1.b. in Table 1) as well as the novel bandit ($V = 10355$, $p < 0.001$, $r = 0.750$; Fig. 2a; Hypothesis 1.1.c. in Table 1), for which there were no initial information available. Our findings thus match our preregistered hypotheses of an increase in exploration in the long horizon.

### Step 1.2. Is exploration beneficial for participants? Yes
To evaluate whether participants were able to use their exploration beneficially, we looked at their performance (i.e., the outcomes they obtained). In alignment with the above analyses, we observed that participants obtained a lower reward (i.e., apple size) in the first draw of the long horizon (i.e., when we observed increased exploration) compared to the single draw in the short horizon ($V = 131612$, $p < 0.001$, $r = 0.53$; Supplementary Fig. 3c; Hypothesis 1.2.a. in Table 1). To assess the long-term benefits of exploration, we calculated the long horizon average reward (across 6 draws) and found that this was higher than the short horizon reward ($V = 264$, $p < 0.001$, $r = 0.864$; Supplementary Fig. 3c; Hypothesis 1.2.b. in Table 1). This indicated that participants made good use of the additional information earned by exploring as observed in previous studies[22,23] in alignment with our preregistered hypotheses. For an analysis of score per trial and per block cf. Supplementary Fig. 4.

### Step 1.3. Do participants use exploration heuristics? Yes
To assess more formally which exploration strategies were being used, we turned to computational modelling, which allows us to tease apart different exploration strategies. In line with our previous studies[22,23],

we found that participants used a mixture of computationally demanding (i.e., Thompson sampling) and two heuristic exploration strategies (i.e., value-free random exploration $\epsilon$ and novelty exploration $\eta$) as captured by the winning model (comparison of BIC average scores: Thompson+$\eta$+$\epsilon$ vs Thompson model: $V = 3089$, $p < 0.001$, $r = 0.835$; Hypothesis 1.3. in Table 1; Thompson+$\eta$+$\epsilon$ vs UCB+$\eta$+$\epsilon$ model: $V = 46440$, $p < 0.001$, $r = 0.389$; Supplementary Fig. 5a). The pilot data (cf. Supplementary Information) and our preregistered hypotheses (cf. Table 1) predicted the same winning model.

### Step 1.4. Are exploration heuristics used more in the long horizon? Yes
Next, we were interested to assess which exploration strategies were deployed more in the long horizon, which is why we examined the winning model's (Thompson+$\eta$+$\epsilon$) fitted parameters. We found an increase in the $\epsilon$-greedy parameter in the long horizon, which captures the contribution of value-free random exploration ($V = 35367$, $p < 0.001$, $r = 0.503$; Fig. 2b; Hypothesis 1.4.a. in Table 1). Similarly, the novelty bonus $\eta$, which captures the intrinsic reward of selecting a novel option, was also increased in the long horizon ($V = 10334$, $p < 0.001$, $r = 0.76$; Fig. 2b; Hypothesis 1.4.b. in Table 1). This thus confirms our preregistered hypothesis of a flexible deployment of these exploration heuristics (cf. Table 1). In addition, we found that the prior variance, capturing complex, uncertainty-related exploration, was also increased in the long horizon (prior variance fitted parameter: $V = 54537$, $p < 0.001$, $r = 0.306$; Fig. 2b), which supports the notion that the long horizon facilitates the exploration strategies we assessed in this task.

### Step 2.1. Is impulsivity linked to value-free random exploration? Yes
Next, we looked at the link between impulsivity and exploration. First, we characterised general impulsivity as a broad concept, and expected it to be linked with value-free random exploration. For this, we used the total score on the Barratt Impulsiveness Scale (BIS), the most commonly administered self-report measure for impulsiveness[49]. We assessed its link to the model parameter and behavioural measure of value-free random exploration, the $\epsilon$-greedy parameter and the low-value bandit picking frequency. As hypothesized (cf. Table 1), we found a significant association between the BIS total score and the $\epsilon$-greedy parameter ($r(578) = 0.171$, $p < 0.001$, Fig. 3a; accounting for age and IQ: $r(573) = 0.117$, $p = 0.005$; Hypothesis 2.1. in Table 1; cf. Methods for details and Supplementary Table 1 for demographics), which was also reflected by a correlation between the BIS total score and the low-value bandit frequency ($r(578) = 0.174$, $p < 0.001$, Fig. 3b; accounting for age and IQ: $r(573) = 0.117$, $p = 0.005$; Hypothesis 2.1. in Table 1). In line with these results, when performing a repeated-measures ANOVA with the horizon as within-participant factor, we found a main effect of impulsivity on how frequently the low-value bandit was chosen (BIS main effect: $F(1,578) = 18.103$, $p < 0.001$, partial eta squared $\eta_p^2 = 0.03$; horizon main effect: $F(1,578) = 113.614$, $p < 0.001$, $\eta_p^2 = 0.164$; BIS-by-horizon interaction: $F(1,578) = 0.773$, $p = 0.380$, $\eta_p^2 = 0.001$) and on the $\epsilon$-greedy parameter (BIS main effect: $F(1,578) = 17.454$, $p < 0.001$, $\eta_p^2 = 0.029$; horizon main effect: $F(1,578) = 125.804$, $p < 0.001$, $\eta_p^2 = 0.179$; BIS-by-horizon interaction: $F(1,578) = 0.084$, $p = 0.772$, $\eta_p^2 < 0.001$), but no significant interaction effects, suggesting that this exploration strategy was increased in both horizons.

In summary, these findings confirmed our preregistered hypothesis that value-free random exploration is linked to general impulsivity traits in this large convenience sample, and our exploratory analyses (cf. Step 4. Exploratory analyses) showed that it was not associated with any other exploration strategy. Detailed correlations can be found in Supplementary Table 5. Detailed correlations with all questionnaires can be found in Supplementary Table 10 and Supplementary Table 11.

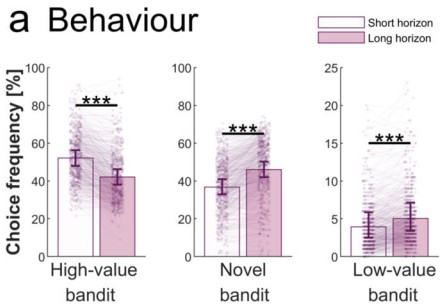

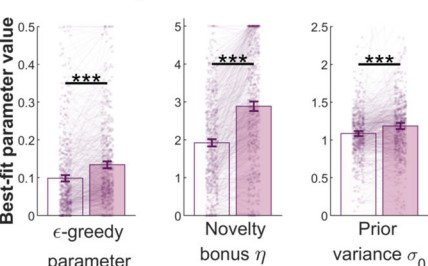

**Fig. 2 | Increased exploration in the long horizon. a** Behavioural horizon effects: in the long (versus short) horizon participants sampled less from the high-value bandit (two-sided Wilcoxon signed-rank two-tailed test: $V = 157079.5$, $p = 1.536e\text{-}81$, Wilcoxon effect size $r = 0.797$) and more from the novel ($V = 10355$, $p = 1.811e\text{-}72$, $r = 0.750$) and low-value bandit ($V = 34420$, $p = 2.817e\text{-}24$, $r = 0.425$). **b** Model parameters: in the long (versus short) horizon participants had higher value of $\epsilon$ (i.e., value-free random exploration; $V = 35367$, $p = 9.831e\text{-}34$, $r = 0.503$), $\eta$ (i.e., novelty exploration; $V = 10334$, $p = 7.411e\text{-}75$, $r = 0.076$) and $\sigma_0$ (their uncertainty about a bandit's mean before seeing any samples; $V = 54537$, $p = 1.868e\text{-}13$, $r = 0.306$). The

parameters were fitted to each participant's first draw, and they were fitted to each horizon separately. ***$p < 0.001$. For detailed statistics cf. Supplementary Table 2. For details about model parameters cf. Supplementary Table 4. Data are shown as mean ± 95% CI and each dot/line represent one participant. Sample size for statistics: $N = 580$ human participants. Source data are provided as a Source Data file. Bar values: High-value bandit: Short Horizon (SH): 52.134, Long Horizon (LH): 42.088; Novel bandit: SH: 36.801, LH: 46.073; Low-value bandit: SH: 3.928, LH: 5.026; $\epsilon$-greedy parameter: SH: 0.099, LH: 0.134; Novelty bonus $\eta$: SH: 1.919, LH: 2.884; Prior variance $\sigma_0$: SH: 1.085, LH: 1.186.

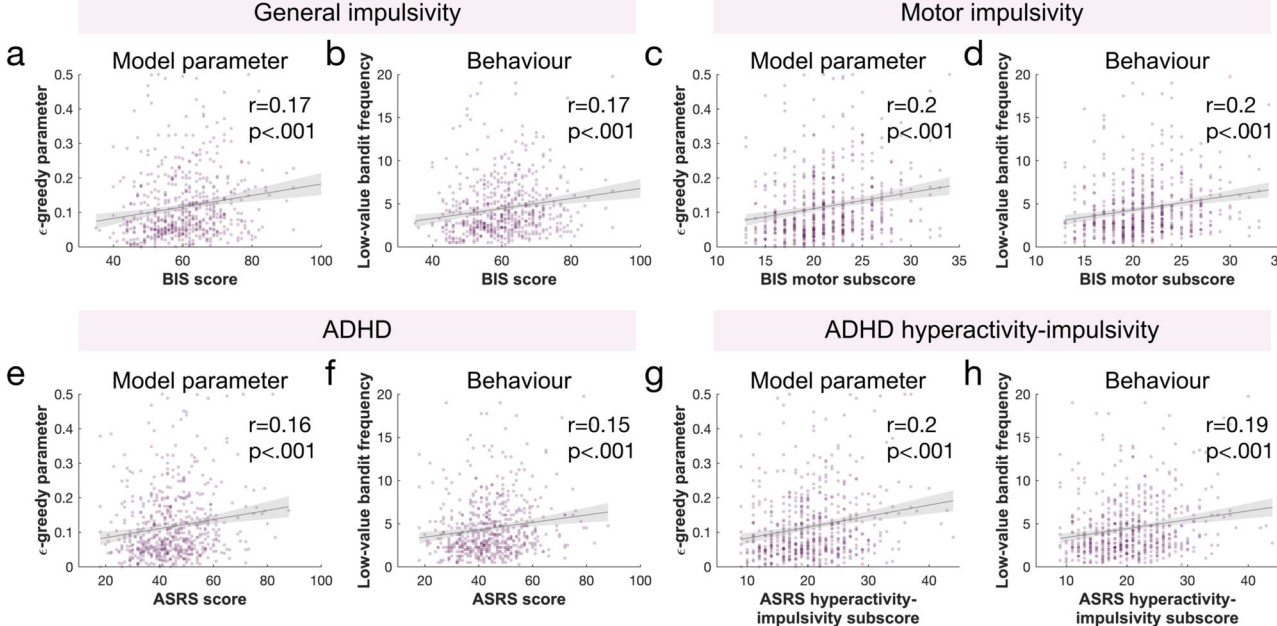

**Fig. 3 | Value-free random exploration linked to impulsivity.** General impulsivity (as measured by the BIS[49]) was significantly associated to value-free random exploration. This was observed both in **a** the model parameter, the $\epsilon$-greedy parameter (Pearson's correlation: $r(578) = 0.171$, $p = 3.398e\text{-}05$) and in **b** the behaviour, the frequency of picking the low-value bandit ($r(578) = 0.174$, $p = 2.442e\text{-}05$). The motor subscore of the BIS was most closely associated with value-free random exploration, both in **c** the model parameter (Bonferroni corrected ($n = 3$): $r(578) = 0.198$, $p_{cor} = 5.318e\text{-}06$, $p_{unc} = 1.772e\text{-}06$) and **d** the behaviour ($r(578) = 0.205$, $p_{cor} = 2.249e\text{-}06$, $p_{unc} = 7.495e\text{-}07$). Similarly, ADHD-

related impulsivity (as measured by the ASRS[72]) was significantly associated to value-free random exploration, both in **e** the model parameter ($r(578) = 0.157$, $p = 1.466e\text{-}04$) and in **f** the behaviour ($r(578) = 0.151$, $p = 3.069e\text{-}04$). The hyperactivity-impulsivity subscore of the ASRS was most tightly associated to novelty exploration, both in **g** the model parameter (Bonferroni corrected ($n = 2$): $r(578) = 0.205$, $p_{cor} = 1.352e\text{-}06$, $p_{unc} = 6.764e\text{-}07$) and **h** the behaviour ($r(578) = 0.193$, $p_{cor} = 6.319e\text{-}06$, $p_{unc} = 3.159e\text{-}06$). The filled lines represent the 95%CI. Sample size for statistics: N = 580 human participants. Source data are provided as a Source Data file.

## Step 2.2. Are ADHD symptoms linked to value-free random exploration? Yes

After having established the association between value-free random exploration and general impulsivity, we sought to investigate the association more specifically, focusing on ADHD symptoms. Based on our previous preliminary findings of a positive association between ADHD traits and value-free random exploration[22] in youths, we thus investigated how the ASRS total score is linked to this form of exploration. A correlation of $r = 0.63$ (cf. Supplementary Fig. 10) between the above BIS total score and the ASRS total

score suggests that they are similar, but not entirely overlapping constructs.

As hypothesized in our preregistration (cf. Table 1), we found an association between the ASRS total score and the $\epsilon$-greedy parameter ($r(578) = 0.157$, $p_{unc} < 0.001$, Fig. 3e; accounting for age and IQ: $r(573) = 0.115$, $p_{unc} = 0.006$; Hypothesis 2.2. in Table 1). Likewise, the same effect was present when looking at the association with its behavioural equivalent, the low-value bandit frequency ($r(578) = 0.151$, $p_{unc} < 0.001$, Fig. 3f; accounting for age and IQ: $r(573) = 0.104$, $p_{unc} = 0.012$; Hypothesis 2.2. in Table 1).

Similar to the above findings, we did not find any interaction with horizon, neither in the low-value bandit (ASRS main effect: F(1,578) = 13.187, $p < 0.001$, $\eta_p^2 = 0.022$; horizon main effect: F(1,578) = 113.468, $p < 0.001$, $\eta_p^2 = 0.164$; ASRS-by-horizon interaction: F(1,578) = 0.025, $p = 0.875$, $\eta_p^2 < 0.001$) nor the $\epsilon$-greedy parameter (ASRS main effect: F(1,578) = 14.609, $p < 0.001$, $\eta_p^2 = 0.025$; horizon main effect: F(1,578) = 126.34, $p < 0.001$, $\eta_p^2 = 0.179$; ASRS-by-horizon interaction: F(1,578) = 2.549, $p = 0.111$, $\eta_p^2 = 0.004$). Our exploratory analyses (cf. Step 4. Exploratory analyses) showed that it was not associated with any other exploration strategy. These results thus confirmed our preregistered hypothesis that value-free random exploration is linked to ADHD symptoms. Detailed correlations can be found in Supplementary Table 5.

## Step 3. Preregistered exploratory analyses

### Step 3.1. Investigating subscales of impulsivity and ADHD
To explore the association between value-free random exploration and general impulsivity further, we performed an exploratory analysis of the BIS subscores (i.e., attentional, motor, and non-planning impulsivity), correcting for multiple comparisons using Bonferroni correction ($N = 3$). Those subscores allow to differentiate between attentional impulsiveness, an 'inability to focus attention or concentrate', motor impulsiveness, 'acting without thinking', and non-planning impulsiveness, a lack of 'futuring' or 'forethought'[49].

We found that the BIS motor subscore was associated with value-free random exploration in all indicators of that exploration heuristic (Bonferroni corrected ($n = 3$): $\epsilon$-greedy parameter: r(578) = 0.198, $p_{cor} < 0.001$, $p_{unc} < 0.001$, Fig. 3c [accounting for age and IQ: r(573) = 0.159, $p_{cor} < 0.001$, $p_{unc} < 0.001$]; frequency of low-value bandit: r(578) = 0.205, $p_{cor} < 0.001$, $p_{unc} < 0.001$, Fig. 3d [accounting for age and IQ: r(573) = 0.165, $p_{cor} < 0.001$, $p_{unc} < 0.001$]). We did not observe any robust association with the BIS non-planning subscore when correcting for age and IQ (using the $\epsilon$-greedy parameter: r(578) = 0.120, $p_{cor} = 0.012$, $p_{unc} = 0.004$ [accounting for age and IQ: r(573) = 0.058, $p_{cor} = 0.501$, $p_{unc} = 0.167$]; using the low-value bandit: r(578) = 0.128, $p_{cor} = 0.006$, $p_{unc} = 0.002$ [accounting for age and IQ: r(573) = 0.065, $p_{cor} = 0.364$, $p_{unc} = 0.121$]) or with the BIS attentional subscore (using the $\epsilon$-greedy parameter: $r = 0.095$, $p_{cor} = 0.067$, $p_{unc} = 0.022$ [accounting for age and IQ: r(573) = 0.067, $p_{cor} = 0.331$, $p_{unc} = 0.11$]; using the low-value bandit: r(578) = 0.086, $p_{cor} = 0.118$, $p_{unc} = 0.039$ [accounting for age and IQ: r(573) = 0.054, $p_{cor} = 0.592$, $p_{unc} = 0.197$]). This suggests that it is the motor dimension of general impulsivity, i.e., acting without thinking, that is related to value-free random exploration the most. Detailed correlations can be found in Supplementary Table 6.

Next, we further explored how the value-free random exploration is associated with the two ADHD subdomains (as assessed by the ASRS), namely inattention and hyperactivity-impulsivity (Bonferroni correcting for $N = 2$ tests). We found that value-free random exploration was linked to the hyperactivity-impulsivity subscore (using the $\epsilon$-greedy parameter: r(578) = 0.205, $p_{cor} < 0.001$, $p_{unc} < 0.001$, Fig. 3g [accounting for age and IQ: r(573) = 0.152, $p_{cor} = 0.001$, $p_{unc} < 0.001$]; using the low-value bandit: r(578) = 0.193, $p_{cor} < 0.001$, $p_{unc} < 0.001$, Fig. 3h [accounting for age and IQ: r(573) = 0.136, $p_{cor} = 0.002$, $p_{cor} = 0.001$]) but not reliably with the ASRS inattention subscore (using the $\epsilon$-greedy parameter: r(578) = 0.087, $p_{cor} = 0.074$, $p_{unc} = 0.037$ [accounting for age and IQ: r(573) = 0.061, $p_{cor} = 0.292$, $p_{unc} = 0.146$]; using the low-value bandit: r(578) = 0.085, $p_{cor} = 0.082$, $p_{unc} = 0.041$ [accounting for age and IQ: r(573) = 0.057, $p_{cor} = 0.348$, $p_{unc} = 0.174$]). This suggests that value-free random exploration is more closely linked to the impulsivity-hyperactivity dimension of ADHD than the other subdomains. Detailed correlations can be found in Supplementary Table 6.

### Step 3.2. Investigating exploration across transdiagnostic dimensions
Thus far, we exclusively focused on our hypothesised association between exploration and impulsivity / ADHD symptoms. However, to be able to explore the wider associations with other symptom dimensions, we additionally collected data from further questionnaires, in the same spirit as previous transdiagnostic dimensional approaches[39,41,42].

As specified in our preregistration, we conducted a factor analysis across all items of the collected questionnaires (including BIS and ASRS). This factor analysis of individual questionnaire items revealed three distinct latent factors (Fig. 4a) which we labelled as "anxious-depression", "uncertainty-related distress" and "impulsivity" factor, in accordance with the strongest individual item loadings (cf. Fig. 4b). For correlations between questionnaires and factors cf. Supplementary Fig. 10.

As we had initially expected, our two impulsivity-related questionnaires (BIS and ASRS) primarily loaded onto one factor (labelled as impulsivity factor). We thus explored the association between this impulsivity factor and value-free random exploration. We found that value-free random exploration was more closely related with the impulsivity factor than with each questionnaire separately (i.e., BIS and ASRS). We found an association between the impulsivity factor and the $\epsilon$-greedy parameter (correcting for multiple comparison using Bonferroni correction across 4 parameters x 3 factors, i.e., $N = 12$; r(578) = 0.257, $p_{unc} < 0.001$, $p_{cor} < 0.001$, Fig. 5a; accounting for age and IQ: r(573) = 0.204, $p_{unc} < 0.001$, $p_{cor} < 0.001$) as well as between the impulsivity factor score and the low-value bandit frequency (correcting for multiple comparison using Bonferroni correction across 3 bandits x 3 factors, i.e., $N = 9$; r(578) = 0.247, $p_{unc} < 0.001$, $p_{cor} < 0.001$, Fig. 5b; accounting for age and IQ: r(573) = 0.191, $p_{unc} < 0.001$, $p_{cor} < 0.001$). Together, our results suggest that value-free random exploration is associated with a general impulsivity that spans across multiple questionnaires.

Having established the link with value-free random exploration, we now explored whether impulsivity was also linked to other forms of exploration. When linking the impulsivity factor with the parameters capturing the other exploration strategies, we did not observe any significant correlation (Bonferroni correction with $N = 12$), neither with the novelty bonus $\eta$ (r(578) = 0.051, $p_{cor} = 1$, $p_{unc} = 0.223$; accounting for age and IQ: r(573) = 0.058, $p_{cor} = 1$, $p_{unc} = 0.167$; Fig. 5e), nor the prior variance $\sigma_0$ (r(578) = −0.02, $p_{cor} = 1$, $p_{unc} = 0.631$; accounting for age and IQ: r(573) = 0.01, $p_{cor} = 1$, $p_{unc} = 0.816$; Fig. 5e), or the prior mean $Q_0$ (r(578) = −0.006, $p_{cor} = 1$, $p_{unc} = 0.891$; accounting for age and IQ: r(573) = −0.016, $p_{cor} = 1$, $p_{unc} = 0.701$; Fig. 5e). This suggests that the impulsivity is first and foremost linked with value-free random exploration.

As a second step, we explored whether exploration correlates with the other factors identified in the factor analysis. Similar to previous studies[40–42,50,51], we retrieved a factor, labelled anxious-depression, which was mainly capturing depression, social anxiety and trait anxiety questions (SDS, LSAS and STAI-Y2 questionnaires respectively). As for the third factor, we obtained a factor that was mainly capturing intolerance of uncertainty (IUS questionnaire), labelled as uncertainty-related distress.

First, we looked at the anxious-depression factor and all exploration strategies as captured by the model parameters (correcting for multiple comparison using Bonferroni correction across all parameters x factors, i.e., $N = 12$). Our exploratory analysis revealed that the anxious-depression factor correlated positively with the novelty bonus $\eta$ (r(578) = 0.14, $p_{cor} = 0.008$, $p_{unc} < 0.001$, Fig. 5c; accounting for age and IQ: r(573) = 0.126, $p_{cor} = 0.03$, $p_{unc} = 0.002$). None of the other parameters was linked to the anxious-depression factor ($\epsilon$: r(578) = −0.047, $p_{cor} = 1$, $p_{unc} = 0.262$; accounting for age and IQ: r(573) = −0.078, $p_{cor} = 0.73$,

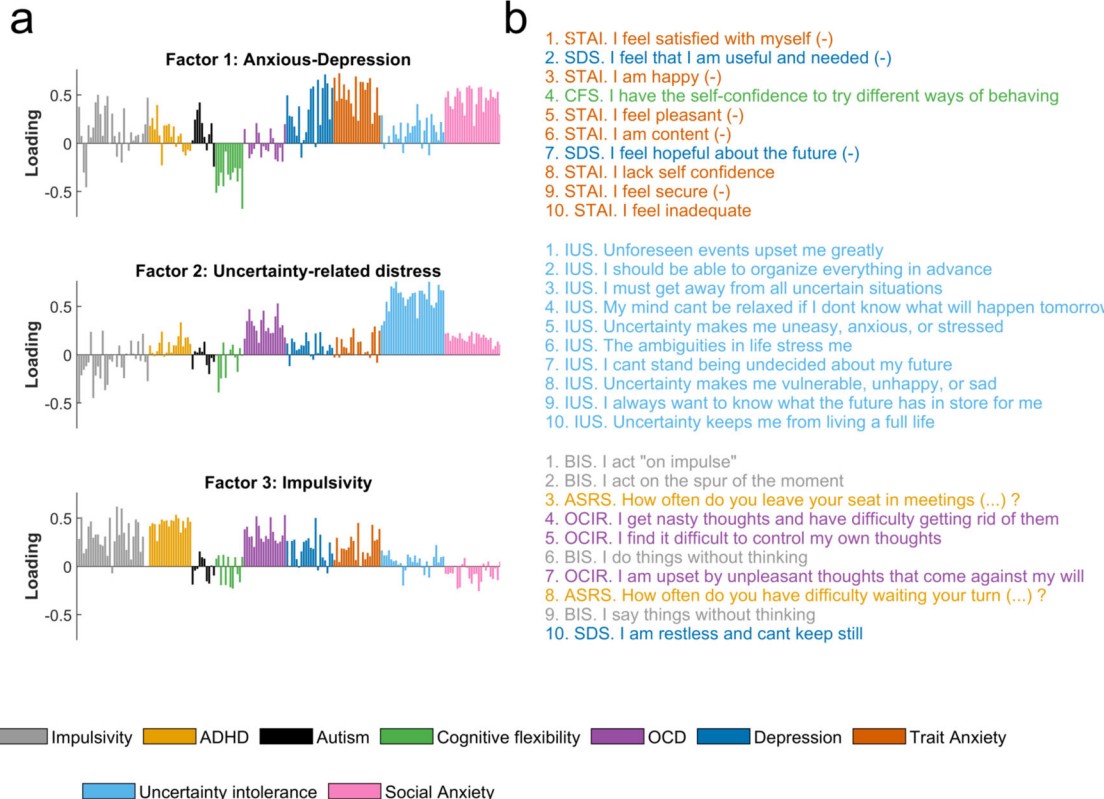

**Fig. 4 | Transdiagnostic parcellation of symptoms. a** Three latent factors were identified when performing a factor analysis on individual questionnaire items. **b** The 10 most loading items for each factor illustrate the aspects that contribute to each dimension. Abbreviations: ASRS: Adult ADHD Self-Report Scale, BIS: Barratt Impulsiveness Scale, LSAS: Liebowitz Social Anxiety Scale, STAI: State-Trait Anxiety Inventory, IUS: Intolerance of Uncertainty, OCIR: Obsessive-Compulsive Inventory-Revised, SDS: Zung's Self-rating Depression Scale, CFS: Cognitive Flexibility Scale, AQ10: Autism spectrum Quotient. (-) indicates reversed items. For correlations between each questionnaire total score cf. Supplementary Fig. 10. For loading weights of individual items on each factor cf. Supplementary Table 10–18. Source data are provided as a Source Data file.

$p_{unc} = 0.061$; $\sigma_0$: $r = 0.099$, $p_{cor} = 0.203$, $p_{unc} = 0.017$; accounting for age and IQ: $r(573) = 0.107$, $p_{cor} = 0.121$, $p_{unc} = 0.01$; $Q_0$: $r(578) = 0.073$, $p_{cor} = 0.925$, $p_{unc} = 0.077$; accounting for age and IQ: $r(573) = 0.052$, $p_{cor} = 1$, $p_{unc} = 0.209$; Fig. 5e).

This pattern was also reflected in the behavioural indicators when looking at the correlation between the 3 factors and the bandit picking frequencies. Correcting for multiple comparisons (Bonferroni correction with N = 9), we found that with the anxious-depression factor, the novel bandit frequency was increased ($r(578) = 0.19$, $p_{cor} < 0.001$, $p_{unc} < 0.001$, Fig. 5d; accounting for age and IQ: $r(573) = 0.17$, $p_{cor} < 0.001$, $p_{unc} < 0.001$), and in turn the high-value bandit frequency decreased ($r(578) = -0.174$, $p_{cor} < 0.001$, $p_{unc} < 0.001$; accounting for age and IQ: $r(573) = -0.138$, $p_{cor} = 0.008$, $p_{unc} = 0.001$; Fig. 5f). We did not observe any correlation with the low-value bandit frequency ($r(578) = -0.058$, $p_{cor} = 1$, $p_{unc} = 0.165$; accounting for age and IQ: $r(573) = -0.093$, $p_{cor} = 0.232$, $p_{unc} = 0.026$; Fig. 5f). Together our results demonstrate that the anxious-depression factor is associated with an increase in novelty exploration.

Lastly, we explored whether the uncertainty-related distress factor was associated with any exploration strategy. We did not observe any significant association (after correcting for multiple comparisons) in neither in the model parameters ($\epsilon$: $r(578) = 0.107$, $p_{cor} = 0.119$, $p_{unc} = 0.01$; accounting for age and IQ: $r(573) = 0.072$, $p_{cor} = 0.997$, $p_{unc} = 0.083$; $\eta$: $r(578) = 0.001$, $p_{cor} = 1$, $p_{unc} = 0.99$; accounting for age and IQ: $r(573) = -0.002$, $p_{cor} = 1$, $p_{unc} = 0.97$; $\sigma_0$: $r(578) = -0.006$, $p_{cor} = 1$, $p_{unc} = 0.877$; accounting for age and IQ: $r(573) = 0.009$, $p_{cor} = 1$, $p_{unc} = 0.821$; $Q_0$: $r(578) = 0.054$, $p_{cor} = 1$, $p_{unc} = 0.197$; accounting for age and IQ: $r(573) = 0.048$, $p_{cor} = 1$,

$p_{unc} = 0.251$; Fig. 5e) nor in the behaviour (low-value bandit: $r = 0.075$, $p_{cor} = 0.625$, $p_{unc} = 0.069$; accounting for age and IQ: $r(573) = 0.035$, $p_{cor} = 1$, $p_{unc} = 0.399$; novel bandit: $r(578) = 0.039$, $p_{cor} = 1$, $p_{unc} = 0.343$; accounting for age and IQ: $r(573) = 0.037$, $p_{cor} = 1$, $p_{unc} = 0.371$; high-value bandit: $r(578) = -0.09$, $p_{cor} = 0.266$, $p_{unc} = 0.03$; accounting for age and IQ: $r(573) = -0.073$, $p_{cor} = 0.732$, $p_{unc} = 0.081$; Fig. 5f).

Taken together, these findings thus suggest that – as hypothesized – impulsivity is associated with value-free random exploration. In addition, we also find a non-hypothesised association between the novelty exploration heuristic and an anxious-depression factor. Detailed correlations can be found in Supplementary Table 7.

### Step 3.3. Associations with cognitive flexibility and autism

We did not observe any correlation between autism and value-free random exploration (the AQ10[52] total score with the low-value bandit frequency: $r(578) = 0.025$, $p = 0.545$; accounting for age and IQ: $r(573) = 0.022$, $p = 0.592$; with the $\epsilon$-greedy parameter: $r(578) = 0.023$, $p = 0.584$; accounting for age and IQ: $r(573) = 0.02$, $p = 0.631$), nor between cognitive flexibility and value-free random exploration (the CFS[53] total score with the low-value bandit frequency: $r(578) = -0.042$, $p = 0.317$; accounting for age and IQ: $r(573) = 0.002$, $p = 0.954$; with the $\epsilon$-greedy parameter: $r(578) = -0.038$, $p = 0.361$; accounting for age and IQ: $r(573) = 0.004$, $p = 0.927$).

### Step 4. Non-preregistered exploratory analyses

The analyses mentioned below were not part of the preregistration.

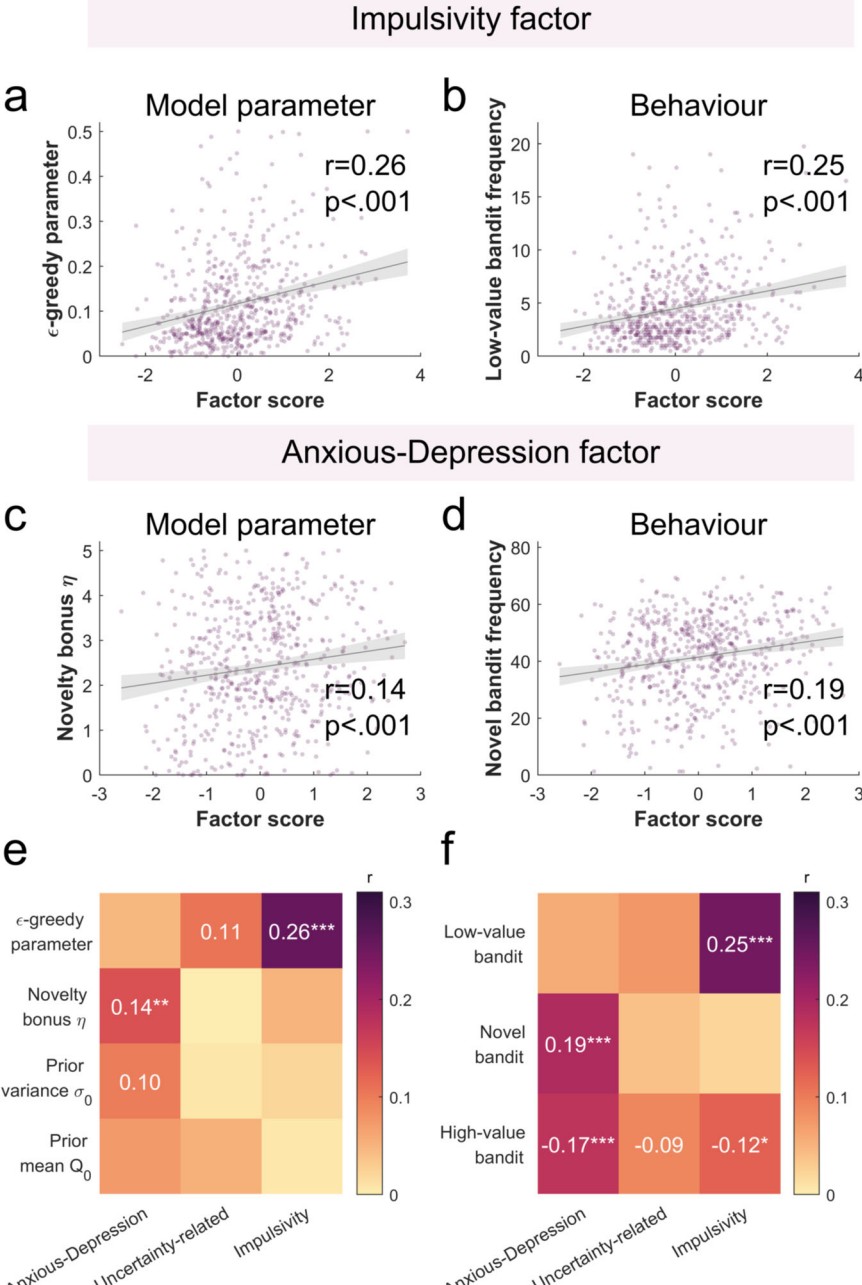

**Fig. 5 | Exploration associations with transdiagnostic psychiatric factors.** The factor analysis-derived impulsivity factor was significantly associated to value-free random exploration. This was observed both in **a** the model parameter, the $\epsilon$-greedy parameter (Bonferroni corrected ($n = 12$); Pearson's correlation: r(578) = 0.257, $p_{unc}$ = 3.352e-10, $p_{cor}$ = 4.023e-09), and in **b** the behaviour, the frequency of picking the low-value bandit (Bonferroni corrected ($n = 9$): r(578) = 0.257, $p_{unc}$ = 1.561e-09, $p_{cor}$ = 1.405e-08). Similarly, the anxious-depression factor was significantly associated with the novelty exploration, both in **c** the model parameter, the novelty bonus $\eta$ (Bonferroni corrected ($n = 12$): r(578) = 0.14, $p_{unc}$ = 7.041e-04, $p_{cor}$ = 0.0084), and in **d** the behaviour (Bonferroni corrected ($n = 9$): r(578) = 0.19, $p_{unc}$ = 4.084e-06, $p_{cor}$ = 3.675e-05), the frequency of picking the novel bandit. Pearson correlations for each factor and **e** model parameters as well as **f** behavioral task measures (i.e., bandit picking frequencies). Significant uncorrected correlations are displayed. Significant corrected correlations are indicated with asterisks (*$p < 0.05$, **$p < 0.01$, ***$p < 0.001$). Sample size for statistics: $N = 580$ human participants. Source data are provided as a Source Data file.

## Step 4.1. Improved performance following exploration is not strategy-specific

To investigate whether the improved performance following exploration (cf. Supplementary Fig. 3c) was specific to an exploration strategy, we split the data by their first choice (i.e., high-value bandit, novel bandit or low-value bandit; cf. Supplementary Fig. 3d). The higher outcome (in the long run) following exploration was irrespective of the exploration strategy used (i.e., novel or low-value bandit).

## Step 4.2. No association between the BIS and other exploration strategies

We explored whether the BIS score was correlated with any of the other exploration strategies (uncertainty-driven Thompson exploration, novelty exploration; correcting for multiple parameters N = 4). We did not observe any association with any of these parameters ($\eta$: r(578) = 0.061, corrected $p_{cor}$ = 0.565, uncorrected $p_{unc}$ = 0.141; accounting for age and IQ: r(573) = 0.073, $p_{cor}$ = 0.32, $p_{unc}$ = 0.08; $\sigma_0$: r(578) = 0.012, $p_{cor}$ = 1, $p_{unc}$ = 0.767; accounting for age and IQ:

r(573) = 0.041, $p_{cor}$ = 1, $p_{unc}$ = 0.331; $Q_0$ : r(578) = 0.029, $p_{cor}$ = 1, $p_{unc}$ = 0.491; accounting for age and IQ: r(573) = 0.026, $p_{cor}$ = 1, $p_{unc}$ = 0.531), while the correlation with value-free random exploration (as detailed above) remained significant ($\epsilon$: r(578) = 0.171, $p_{cor}$ < 0.001, $p_{unc}$ < 0.001; accounting for age and IQ: r(573) = 0.117, $p_{cor}$ = 0.019, $p_{unc}$ = 0.005).

### Step 4.3. No association between the ASRS and other exploration strategies

We explored whether the ASRS score was correlated with any of the other exploration strategies (correcting for multiple parameters N = 4). We did not observe any robust association with any of the other parameters ($\eta$: r(578) = 0.083, $p_{cor}$ = 0.187, $p_{unc}$ = 0.047; accounting for age and IQ: r(573) = 0.087, $p_{cor}$ = 0.152, $p_{unc}$ = 0.038; $\sigma_0$: r(578) = 0.015, $p_{cor}$ = 1, $p_{unc}$ = 0.713; accounting for age and IQ: r(573) = 0.038, $p_{cor}$ = 1, $p_{unc}$ = 0.369; $Q_0$ : r(578) = 0.03, $p_{cor}$ = 1, $p_{unc}$ = 0.466; accounting for age and IQ: r(573) = 0.022, $p_{cor}$ = 1, $p_{unc}$ = 0.603), while the correlation with value-free random exploration remained significant ($\epsilon$: r(578) = 0.157, $p_{cor}$ = 0.001, $p_{unc}$ < 0.001; accounting for age and IQ: r(573) = 0.115, $p_{cor}$ = 0.023, $p_{unc}$ = 0.006).

### Step 4.4. Analysis of 2nd winning model

Value-free random exploration (captured by the $\epsilon$-greedy parameter) was similar in the 1st winning model (Thompson+$\epsilon$+$\eta$) and in the 2nd winning model (UCB+$\epsilon$+$\eta$), both in the short horizon (Pearson correlation: r(578) = 0.87, p < 0.001; Supplementary Fig. 7a) and in the long horizon (r(578) = 0.85, p < 0.001; Supplementary Fig. 7b). Similarly, novelty exploration (captured by the novelty bonus $\eta$) was similar across both models, both in the short horizon (r(578) = 0.71, p < 0.001; Supplementary Fig. 8a) and in the long horizon (r(578) = 0.72, p < 0.001; Supplementary Fig. 8b).

Similar to the 1st winning model, we observed an association between value-free random exploration (i.e., $\epsilon$-greedy parameter) as captured by 2nd winning model, and impulsivity. We observed a significant association between $\epsilon$ and the BIS total score (r(578) = 0.155, p < 0.001; controlling for age and IQ: r(573) = 0.101, p = 0.015), between $\epsilon$ and the ASRS total score (r(578) = 0.167, p < 0.001; controlling for age and IQ: r(573) = 0.126, p = 0.006), between $\epsilon$ and the impulsivity factor (cf. Fig. 4; r(578) = 0.250, p < 0.001; controlling for age and IQ: r(573) = 0.196, p < 0.001).

Additionally, we observed an association between novelty exploration (i.e., novelty bonus $\eta$) and the anxious-depression factor (r(578) = 0.124, p = 0.003; controlling for age and IQ: r(573) = 0.101, p = 0.015).

### Step 4.5. Further analysis of impulsivity factor and value-free random exploration association

In line with the previous impulsivity results, when performing a repeated-measures ANOVAs with horizon as the within-participants factor, we found a main effect of impulsivity (i.e., impulsivity factor score) on the low-value bandit (impulsivity main effect: F(1,578) = 37.664, p < 0.001, $\eta_p^2$ = 0.061; horizon main effect: F(1,578) = 113.474, p < 0.001, $\eta_p^2$ = 0.164; impulsivity-by-horizon interaction: F(1,578) = 0.059, p = 0.808, $\eta_p^2$ = 0) and on the $\epsilon$-greedy parameter (impulsivity main effect: F(1,578) = 40.872, p < 0.001, $\eta_p^2$ = 0.066; horizon main effect: F(1,578) = 126.418, p < 0.001, $\eta_p^2$ = 0.179; impulsivity-by-horizon interaction: F(1,578) = 2.906, p = 0.089, $\eta_p^2$ = 0.005), but no horizon interaction.

## Discussion

In this preregistered study, we investigated how impulsivity is related to exploration, and more specifically, how a computationally light exploration heuristic, value-free random exploration, is associated with different measures of impulsivity. Using a behavioural task and computational modelling we demonstrate that inter-individual variability in value-free random exploration usage is associated with general impulsivity in a large-sample online study.

We and others have previously shown that humans deploy a multitude of different strategies for exploration[22–25,30,54] that all approximate an optimal exploration strategy, which is intractable in open-ended decision problems. In our current data, we confirmed that our participants utilised a mixture of resource-requiring complex strategies and computationally light heuristics. The resource-demanding strategies (such as Thompson sampling or UCB) demand keeping track of expected means and uncertainties across the different choice options. The computationally lighter heuristic strategies, namely value-free random exploration (captured by $\epsilon$-greedy) and novelty exploration (captured using a novelty bonus $\eta$), although being less optimal, require substantially less computational power, making them very useful in practice. Using model comparison as well as model simulations, we were able to demonstrate the presence of both complex and heuristic exploration strategies. The winning model, combining complex Thompson with novelty ($\eta$) and value-free random ($\epsilon$) exploration, was not entirely distinguishable from the 2nd winning model, combining complex UCB with novelty and value-free random exploration, but was well distinguishable from other models (cf. confusion matrix, Supplementary Fig. 6b) with relatively high confidence regarding its generative origins (cf. inversion matrix, Supplementary Fig. 6c). This suggests that the two complex exploration strategies make similar predictions in our task, preventing us to disentangle them properly. However, we capture similar amounts of value-free random exploration, irrespective of the complex model used, demonstrating the robustness of our result. Our results therefore show that participants supplemented complex strategies (UCB or Thompson sampling) with two heuristic strategies. Given that we find an association between value-free random exploration and impulsivity irrespective of the complex model used, this does not impact the conclusions in the given study.

Impulsivity is a crucial construct across both general and clinical populations, but the links to specific computational mechanisms are still far from clear[4]. Based on previous theoretical[6,12,14–17] and some experimental work[21,23], exploration is believed to be increased in impulsivity[14] and especially in ADHD. Here, we extend these previous studies by identifying that it is value-free random exploration specifically which is increased, whilst other forms of exploration were not found to be robustly linked. This form of exploration is the computationally least demanding as it simply ignores all existing information. This is well aligned with a notion of impulsivity as 'acting without thinking', which is also captured in the motor impulsivity scale of the BIS. The latter showed a much closer association with value-free random exploration than the other attentional and non-planning impulsivity BIS subscores, which capture the inability to concentrate or a lack of forethought. We did not find a significant association between this form of exploration and a measure of global cognitive flexibility (cf. Supplementary Information), supporting the idea that cognitive flexibility and planning inabilities might of different neurocognitive constructs. However, it would be interesting to investigate whether value-free random exploration is related to more specific tasks, such as set shifting, inhibition or other decision making and learning tasks, given that cognitive flexibility in itself is a relatively heterogeneous construct[55].

From our results, it remains unclear which brain processes exactly mediate value-free random exploration. Interestingly, we have previously found that this form of exploration is modulated by noradrenaline functioning[23], a neurotransmitter which plays an important role in impulsivity-related disorders such as ADHD[6,7,12,17,34–38], which could be a potential mechanism. Previous findings that linked noradrenaline functioning to what is traditionally seen as motor impulsivity support this notion[56]. This form of exploration may be also related to brain circuits generally seen to be linked to noradrenaline functioning

(for a detailed discussion of noradrenaline and executive functions, see Chamberlain & Robbins[56]). In particular, anterior cingulate cortex would be a candidate as it is heavily innervated by noradrenaline and linked and linked to similar exploratory behaviour[33]. In addition, fronto-striatal loops including orbito-frontal and dorso-lateral pre-frontal cortex may also be involved, as they have often been found to be involved in tasks that are modulated by noradrenaline related to set shifting[56,57]. However, the precise neural processes underlying value-free random exploration needs to be examined in more detail.

Given that value-free random exploration ignores all prior information, it begs the question why humans use this strategy in exploration. Interestingly, inducing randomness or noise has often been shown to benefit a system both in living species and in machines, supporting the importance of such strategies[12,58–62]. Here, the main benefit of value-free random exploration is that it does not require demanding computations, allowing exploration even with restrained neural resources[63] or a limited ability/willingness to engage with mentally effortful computations[43]. Exploring in a seemingly random way can be beneficial, either at an individual or a group level, in many different contexts. For example, in the case of an absence of prior knowledge[14], increased stochasticity can help to speed up learning. Additionally, in a case of imprecise or even inaccurate prior knowledge, random exploration ignores such erroneous priors and prevents them from penalizing future decision-making. Introducing stochasticity can also be beneficial in the case of dynamic environments e.g., where values can change drastically and thus agents should not rely solely on their expectations[62]. Our findings of such exploration heuristics are also well aligned with recent findings showing that limiting cognitive resources impacts the use of exploration strategies[64], and shifts in exploration strategies can be induced by applying constraints such as time pressure[65]. Overall, our findings suggest at least two roles for exploration in impulsivity: a more flexible way of exploration which does not rely on (potentially wrong) prior knowledge and a way to circumvent mental effort. Importantly, value-free random exploration is used by all participants in a goal-directed manner (i.e., they used it more when exploration was beneficial). This means that participants adapt their usage of value-free random exploration to the demands of the task.

Because impulsivity is a feature of multiple psychiatric disorders, we investigated it in a transdiagnostic, population-based dimensional manner. This approach allowed us to capture a more general dimension of impulsivity rather than a sub-trait of a specific disorder. To obtain a transdiagnostic impulsivity factor, we performed a factor analysis similar to previous studies[39–42]. Such an approach also helps to reduce the noise that is present when investigating individual questionnaires. We identified an impulsivity factor, capturing both impulsivity questionnaires (BIS, ASRS) as well as some aspects of OCD (as captured by OCI-R that was also related to value-free random exploration, cf. Supplementary Fig. 11). Interestingly, this factor was associated to value-free random exploration to an even stronger degree than the individual impulsivity questionnaires.

In addition to the impulsivity factor, we also identified an anxious-depression factor, but unlike previous studies we did not find a separate compulsivity factor. This is most probably due to the fact that we did not use the exact same set of questionnaires (previous studies included more compulsivity-related questionnaires). As some previous studies have suggested that depression and anxiety are associated to abnormal exploration[45,47,48], we explored these possible links in our dataset. After controlling for multiple comparisons, we indeed observed an association between the anxiety-depression factor and our parameter capturing the intrinsic value of novelty, the novelty bonus $\eta$. We did not observe any association between the third, uncertainty-related factor, nor any specific exploration strategy. Our findings suggest that those with increased anxiety-depression traits deployed the novelty-related exploration heuristic more eagerly. This

is aligned with previous findings showing increased exploration in participants with higher levels of anxiety[66,67]. It is believed that this is because exploration aids in overcoming long-term uncertainty, and an uncertainty aversion is commonly reported in anxiety[68]. Targeting novelty in exploration might be a way to save cognitive resources as one does not need to compute expected values and uncertainties of the other options, but instead can be simply guided by what has not been encountered before. This strategy thus seems deployable even under increased stress and anxiety. Even though we have rigorously controlled for multiple comparisons, we believe an independent replication of this somewhat unexpected result would be desirable. Moreover, it would be interesting to assess whether the deployment of such novelty exploration is more closely linked to apathy or anhedonia, as they are both important features of depression.

We did not find any direct association between the trans-diagnostic factors and our complex exploration strategy (here: Thompson sampling). It needs to be noted that our task was optimised to detect the exploration heuristics. As a consequence, the complex exploration strategies make relatively similar predictions (cf. Supplementary Information). It is thus possible that in other tasks (e.g., by varying the generative bandit variance[25,65,69]; or larger decision spaces[30]), the coexistence of Thompson and UCB exploration is clearer and may be more directly linked to one of the trans-diagnostic dimensions. However, this is unlikely to impact the impulsivity findings presented here, as we find them irrespective of the complex strategy we are using in our computational models (cf. Supplementary Information). In addition, alternative exploration strategies, such as repeating one's previous choice could provide additional insight[65].

In this registered report, we demonstrated that transdiagnostic impulsivity is associated with value-free random exploration. By pre-registering and peer-reviewing our specific hypotheses using a previously-validated task[22,23] and a well-defined dimensional approach[39–41,70], we were able to demonstrate this specific association. Our results aid in understanding the adaptivity of impulsivity and are important for the understanding of behaviour in the general and in clinical populations given the high prevalence of impulsivity. Nonetheless, future studies should investigate the validity of those effects in clinically diagnosed patient populations.

## Methods

### Ethics information

The study has been approved by the UCL research ethics committee (REC No 15301/001) and written informed consent was obtained from all participants. Participants were reimbursed for their participation on an hourly basis and received a bonus according to their performance (proportional to the sum of obtained rewards). The total compensation was bound between £8.25 and £12.0 per hour.

### Design

**Task**. Participants were recruited online on Prolific Academic (www.prolific.ac), which manages the participant allocation and their reimbursement. Participants signed an online consent form and were redirected to the task.

We deployed a multi-armed bandit task which we have recently developed[23], and which allows us to capture different forms of exploration. On each trial, participants had to choose between different bandits (depicted as trees; cf. Figure 1) which one they want to draw a sample (i.e., pick an apple) from and therefore obtain a reward (the apple's size). Participants were instructed to maximise their score (i.e., sum of apple sizes) in order to maximise their overall reimbursement (i.e., they were instructed that they will receive a cash bonus proportional to their performance). Prior to the participants' first choice, bandits display varying levels of information about the plausible rewards they carry. Information is given in the form of 'initial samples', i.e., apples that have been picked before. We varied the

number of initial samples that were displayed for each bandit (identifiable by colour) to dissociate different forms of exploration (cf. below). The initial samples of each bandit are drawn from their generative normal distributions (cf. Supplementary Information for detail), meaning that initial samples carry important information for future choices as the mean of already observed bandits can be estimated[24].

To induce changes in exploration, similar to the horizon task[24] and our previous studies[22,23], we manipulated the number of samples they could draw from a given set of bandits[24]. This decision horizon varied between two conditions (intermixed trials): they could either perform one draw (short horizon condition) or six draws (long horizon condition). The long horizon promotes exploration as obtained information can subsequently be used[24]. Although there would be no interest for an optimal agent to explore in the short horizon, humans still show signs of exploration even when it is not beneficial, though to a much lesser extent[14,58]. In fact, exploration in the short horizon has previously been observed in humans[23,24,71].

We constructed the reward and information of each bandit to be able to assess the contributions of different exploration strategies that have previously been put forward[23–25]. Each bandit $i$ is from one of four generative groups characterised by different means $\mu_i$ and number of initial samples, following the same procedure as other studies[24]. The size of the apple is determined by its radius (cf. Supplementary Fig. 1). Manipulating the amount of information participants have before they make their choice (i.e., initial samples) avoids a potential reward-information confound[24]. The samples of each bandit are then sampled from a normal distribution with a fixed sampling variance $\mathcal{N}(\mu, 0.8)$, truncated to [2, 10], and rounded to the closest integer. Each mean $\mu$ was sampled from $\mathcal{N}(\mu_{overall}, 1.4)$, with an "overall mean" $\mu_{overall}$ specific to each bandit type. The overall mean was computed similarly to previous studies:[24] On each trial we set the overall mean for one of the bandits, the 'certain-standard bandit', to be either 4.5 or 6.5. We determine the overall mean of the 'standard bandit' by adding a number sampled uniformly from [−2, −1, +1, +2] to the certain-standard bandit overall mean. Similarly, we determine the overall mean of the 'novel' bandit by adding a number sampled uniformly from [−2, −1, +1, +2] to either the certain-standard bandit overall mean or the standard bandit overall mean. By doing this, we make sure that the means of those 3 bandits are comparable. This results in the means of the standard bandit and novel bandit spanning a slightly larger range compared to the certain-standard bandit means (cf. Supplementary Table 3). To make sure that the 'low-value' bandit mean was always the smallest, it's overall mean is computed by subtracting 1 to the minimum of the above-mentioned average means. Bandits also carry different amounts of information: The certain-standard bandit provides 3 initial samples, the standard bandit provides 1 initial sample, the novel bandit does not provide any initial samples and the low-value bandit provides 1 initial sample. Even though the absolute range of reward is set, randomly scaling each reward mean around the certain-standard bandits' reward mean allows to maintain uncertainty about the overall average reward on each trial similarly to previous studies[24,27]. On each trial, the average value of the certain-standard bandit initial samples is compared to the value of the standard bandit initial sample. The bandit with such a higher value is referred to as the (expected) 'high-value' bandit. For detailed comparison between those average rewards cf. Supplementary Table 19. At the beginning of each trial, the initial samples of the presented bandits are sampled from their respective distributions. We ensured that the initial sample from the low-value bandit is the smallest by resampling from this bandit in the trials where it is not the case, similar to our previous study. For detailed information about the value of initial samples, first draw and later draws cf. Supplementary Table 3. The order of all initial samples is then permuted to avoid biases. Additionally, to be able to compute choice consistency which is specifically reduced in value-free random

exploration[23], each trial is duplicated. Overall, each participant is asked to play 400 trials (200 in each horizon condition). The trees' positions (left, middle or right) as well as their colour (8 sets of 3 different colours) where shuffled between trials.

The task has originally been developed in a lab setting[23] and has now been adapted for online use. We have adjusted the instructions, making them as clear as possible while keeping the participants' attention. Following the initial task instructions, to make sure that they understood what they need to do, they were asked to answer 5 questions. Similar to previous online studies[40–42], failing to correctly answer these questions guided the participant back to the instructions until all correct answers are given. To make sure that participants understood that the apples from the same tree are always of similar size (generated following a normal distribution), participants additionally performed several training trials. In this training, based on three displayed apples of similar size, they had to guess, between two options, which apple is the most likely to come from the same tree and receive feedback about their choice. If participants gave a wrong answer in at least 3 of the 10 trials, they were asked to restart the training. Task pilot data (N = 61, cf. below) demonstrated comparable effects and effect sizes (cf. Data analysis, Step 1) to our previous lab-based data[23].

**Behavioural analysis nomenclature.** For the behavioural analysis, we categorized each bandit according to the number and size of initial samples (apples shown before the first draw). The bandit with the highest sampling mean, carrying either a lot or some prior information (i.e., 3 or 1 initial samples; for further split cf. Supplementary Information), is referred to as the 'high-value bandit'. The bandit for which no prior sample was shown is named the 'novel bandit', and the bandit with one initial sample from a substantially lower generative mean (trials were constructed to have sufficient number of such trials[23]) is called the 'low-value bandit'. The high-value bandit is an evident signature of exploitation (choosing maximal expected value), the novel bandit is captured amongst other by 'novelty exploration' which is biased towards options for which nothing is known, and the 'low-value bandit' appeals to the value-free random exploration alone as it is the only strategy which does not take expected values into account[23].

**Assessing psychiatric symptoms.** After completing the task, participants were asked to fill-in several self-report questionnaires. To assess impulsivity, our key dimension of interest, we used the Adult ADHD Self-Report Scale[72] (ASRS) and the Barratt Impulsiveness Scale[49] (BIS). In addition, we collected further questionnaires to investigate additional psychiatric dimensions (cf. Data analysis, Step 3). These entail the Liebowitz Social Anxiety Scale[73] (LSAS), the State-Trait Anxiety Inventory[74] (STAI-Y2), Intolerance of Uncertainty Scale[75] (IUS), Obsessive-Compulsive Inventory-Revised[76] (OCI-R), and Zung's Self-rating Depression Scale[77] (SDS), in accordance with similar previous approaches[40–42], as well as the Cognitive Flexibility Scale[53] (CFS) and the Autism spectrum Quotient[52] (AQ-10). To control for confounding factors, such as intelligence and medication, participants additionally completed the International Cognitive Ability Resource sample test[78] (ICAR) and were asked whether they take psychoactive medication and/or medication to increase attention/concentration on a regular basis. As a measure of data quality, attention checks are added to every questionnaire to make sure that participants read the questions[79]. Failure in 1 or more attention check resulted in the participants' exclusion from data analysis.

**Blinding and randomisation do not apply for this study.** The full code (written using the open source React JavaScript library) of the task can be found online (https://github.com/MagDub/MFweb-app).

## Sample
### Power analyses
The analysis consisted of two preregistered steps addressing separate research questions. Step 1 consisted of expanding our pilot data (cf. Data analysis) and replicating the main characteristics of the previously lab-based task[23] in an online setting. In Step 2 we assessed our main research questions and looked at associations between exploration measures and impulsivity traits. Lastly, an exploratory factor analysis was conducted at Stage 2 and is reported subsequently. In Step 3 the factor analysis across all questionnaires allowed us to explore the relationship between psychiatric dimensions and exploration measures more broadly.

For Step 1's sample size estimation, in which we attempted to replicate the task main effects, we collected online pilot data ($N = 61$ after exclusion). A total of 4 hypotheses were tested (hypothesis 1.1 to 1.4; cf. Table 1 and Data analysis for details). The lowest effect size across all tests in the pilot study (Wilcoxon signed-rank effect size = 0.410) was used for our power analysis, which suggested that a sample of $N = 83$ is sufficient to reach 95% power for all hypotheses. For a summary of the statistics performed on all measures on the pilot data cf. Supplementary Table 20. Importantly, Wilcoxon signed-rank tests was used instead of paired t-tests if the Shapiro normality assumption was violated.

For Step 2's sample size estimation, where the link between exploration and impulsivity was investigated, the correlation coefficient of our previous study using the same task[22] was used for our power analysis. In this prior study, a Pearson correlation of $r = 0.26$, $p < 0.001$ was observed between an impulsivity measure[80] (the Conners ADHD questionnaire) and value-free random exploration in youths. Assuming a similar correlation in adults, our power analysis suggested that a sample of $N = 190$ is sufficient to reach 95% power (G*Power analyses suggest a similar sample size of $N = 186$). This moderate size correlation factor is in line with previous studies linking BIS-measured impulsivity to behaviour (e.g., with delay discounting[81–84]). Similarly, for Step 2's Stage 2 exploratory analysis in which we looked at the correlation between value-free random exploration and the three subdomains of BIS, G*Power analyses suggested that a sample size of $N = 228$ is sufficient to reach a 95% power at a significance corrected for multiple comparisons using Bonferroni correction.

However, assuming a lower association strength and taking into account previous dimensional analyses using exploratory factor analysis (similar to our exploratory Step 3), we additionally considered the correlation coefficients obtained from these previous big data dimensional studies. These previous studies have observed correlations from $r = 0.15$ (negative association between dogmatism and metacognitive sensitivity[39]) up to correlations of $r = 0.25$ (association between confidence and compulsivity[41]). The relatively small effect sizes can be explained by the higher noise associated with large online samples as well as the lack of precision of behavioural and questionnaire measures. In this study we account for these facets and consider the study as a first step to establish associations between measures by conducting thorough effect size and power calculations. The lowest correlation ($r = 0.15$) was used to extend our power analysis, which suggested that a sample of $N = 580$ is sufficient to reach 95% power (cf. Supplementary Fig. 12; G*Power analyses suggested a similar sample size of $N = 571$). Taking all steps together, to reach at least 95% power across all measures, we collected a total sample of $N = 580$ participants. A sensitivity power analysis (performed in G*Power) predicted that with such a sample size, we would be able to detect an effect size (Minimal Detectable Effect, MDE) of 0.15 with 95%. Importantly, the lower bound of the 95% Confidence Interval of each pilot data measures' effect size was above this MDE (cf. Supplementary Table 20), ensuring a detectable effect.

Effect sizes as well as hypothetical sample sizes to reach a 95% power can be found in Table 1 (details about each measure in the pilot data can be found in Supplementary Table 20). For power analysis, we used the G*Power Software[85] for the t-tests in Step 1 (using the pilot data). Power analysis for the correlations in the further steps was performed using simulations in MATLAB. The 'matter' gallery of the 'cmocean' colourmap was used for the figures[86,87]. We obtained summary statistic scores or correlations from previous studies and used bootstrapping to simulate data. Concretely, for t-tests, $n$ simulated participants were sampled from each group normal distribution: $N$(m1, std1) and $N$(m2,std2) and significance was assessed using paired t-test on those 2 data sets. For correlations, $n$ simulated participants were taken from the bivariate distribution of mean = [0, 0] and covariance matrix = [1, R; R,1]. To assess power of a given sample size, we assessed the number of significant tests ($p < 0.05$) of a total number of $N = 10000$ simulations. The summary statistics for Step 1 were taken from our pilot data (cf. Supplementary Information), for Step 2 from our previous study in youths[22], and for Step 3 from previous big data studies[39,41].

**Participant recruitment.** To take part in the study, participants had to be above 18 years of age and have their current residence in the UK. To ensure data quality, participants were excluded according to the following criteria: data was incomplete, the mean score (i.e., apple size) was lower than 5.5 indicating participants were performing at chance level[40] (cf. Supplementary Fig. 2b), the first draw mean reaction time was faster than 1500 ms (based on our pilot data and previous study[23]) indicating participants were not allocating much thought to their choice (cf. Supplementary Fig. 2c) and if participants failed at least one attention check during the questionnaires meaning that they were not reading the questions[40,41,79]. According to these exclusion criteria, $N = 77$ participants were excluded (cf. Supplementary Fig. 2) and replaced prior to data analysis in order to reach a final sample of $N = 580$ ($N = 3$ participants were excluded out of $N = 64$ in the pilot data).

## Data analysis
### Step 1
This step aims at replicating the main characteristics of the previously lab-based task[23] in an online setting. Here, we report results from our pilot data set (N = 61), which we collected online using the exact same online task to estimate the effect sizes. The analysis follows the pipeline which we have successfully used in our previous studies[22,23]. In line with previous studies investigating horizon-dependent exploration[23,24,71], we only investigated the first draw of each horizon in the main analysis. This allowed us to compare between horizon conditions preventing biases of collected reward and unequal variance.

**Participants explore more when it is worth it.** To assess whether the horizon manipulation promoted exploration, we analysed whether participants explored more in the long (versus short) horizon condition, in which additional information can inform later choices. To this end, we assessed which bandit participants chose on their first draw. Replicating our previous studies[22,23], we expected several exploration markers to differ. We predicted that participants would choose bandits with a lower expected value (computed as the mean of the bandits' initial samples) in the long horizon (pilot data: t(60) = 3.585, $p = 0.001$, 95% confidence interval of the mean: $CI_M = [0.047, 0.165]$, effect size: Cohen's d = −0.459, 95% confidence interval of the effect size $CI_{ES} = [−0.727, −0.195]$). This is reflected by the frequency of picking the high-value bandit, which we predicted to decrease in the long horizon (pilot data: t(60) = 8.45, $p < 0.001$, 95%$CI_M = [6.92, 11.211]$, d = −1.082, 95%$CI_{ES} = [−1.407, −0.769]$). Similarly for the frequency of picking the low-value bandit, we predicted it to increase in the long horizon (pilot

data: $t(60) = -3.446$, $p = 0.001$, $95\%\text{CI}_M = [-1.568, -0.416]$, $d = 0.441$, $95\%\text{CI}_{ES} = [0.178, 0.708]$). We predicted this exploration to be goal-directed, with participants choosing bandits they know less about (lower number of initial samples, i.e., more informative) in the long horizon (pilot data: $t(60) = 9.625$, $p < 0.001$, $95\%\text{CI}_M = [0.184, 0.281]$, $d = -1.232$, $95\%\text{CI}_{ES} = [-1.576, -0.903]$). This is largely reflected by the frequency of the novel bandit, which we predicted to increase in the long horizon (pilot data $t(60) = -8.586$, $p < 0.001$, $95\%\text{CI}_M = [-11.178, -6.954]$, $d = 1.099$, $95\%\text{CI}_{ES} = [0.784, 1.427]$).

**Participants use exploration beneficially.** To evaluate whether participants were able to use exploration beneficially, we looked at their performance (i.e., the outcomes they obtained). We first compared the reward (i.e., apple size) obtained in the short horizon with the first reward obtained in the long horizon. As the latter is driven by exploration, we expected it to be lower (pilot data: $t(60) = 6.522$, $p < 0.001$, $95\%\text{CI}_M = [0.059, 0.112]$, $d = -0.835$, $95\%\text{CI}_{ES} = [-1.134, -0.545]$). As observed in previous studies[22,23], we expected them to make good use of the additional information earned by exploring, and therefore their long horizon average reward (across 6 draws) was expected to be higher than the short horizon reward (pilot data: $t(60) = -16.096$, $p < 0.001$, $95\%\text{CI}_M = [-0.245, -0.191]$, $d = 2.061$, $95\%\text{CI}_{ES} = [1.626, 2.524]$).

**Participants explore using heuristics.** To formally assess which exploration strategies were being used, we turned to computational modelling. Similar to the behavioural analysis, only the first draw of each trial was analysed. We compared 16 models that make different predictions about the usage of exploration strategies (cf. Supplementary Information). Similar to our previous studies[22,23], we expected participants to use a mixture of computationally demanding (i.e., Thompson sampling and/or UCB) and heuristic exploration strategies (i.e., value-free random exploration and novelty exploration) captured by the winning model (pilot data: BIC average score: Thompson $+\eta+\epsilon$ vs Thompson model: $t(60) = -10.187$, $p < 0.001$, $95\%\text{CI}_M = [-72.866, -48.946]$, $d = 1.304$, $95\%\text{CI}_{ES} = [0.967, 1.657]$). Model comparison was computed using the commonly used Bayesian Information Criterion (BIC). The winning model, i.e., the model with the lowest BIC score, was used for subsequent analyses. All models that were not significantly different than the 1$^{st}$ winning model would have been used for subsequent analysis to demonstrate the generalisability of the effect (similar to previous studies[22]). Model fitting was performed using the maximum a posteriori probability (MAP) estimate, which allows incorporation of prior beliefs. All the parameters besides participants' initial estimate of a bandit's mean ($Q0$; prior mean) and the contribution of each model in the hybrid model ($w$) were free to vary as a function of the horizon as they capture different exploration forms (cf. Supplementary Information for details).

**Participants rely more on heuristics in the long horizon.** To assess the changes in exploration strategy, we examined the winning model's fitted parameters. Those parameters were fitted to the first draw of all trials of each participant. We expected the $\epsilon$-greedy parameter, which captures the contribution of value-free random exploration, to be increased in the long (versus short) horizon (pilot data: $t(60) = -3.23$, $p = 0.002$, $95\%\text{CI}_M = [-0.058, -0.014]$, $d = 0.413$, $95\%\text{CI}_{ES} = [0.152, 0.679]$). Similarly, we expected the novelty bonus $\eta$, which captures the intrinsic reward of selecting a novel option, to be increased in the long horizon (pilot data: $t(60) = -9.43$, $p < 0.001$, $95\%\text{CI}_M = [-1.265, -0.822]$, $d = 1.207$, $95\%\text{CI}_{ES} = [0.881, 1.548]$).

**Step 2**

In this step we tested our main hypothesis about value-free random exploration being linked to impulsivity and ADHD traits. Our key measure of interest is the mean $\epsilon$ parameter[22] - measuring value-free

random exploration - and how it is related to our specific questionnaire measures. For the correlations, we used the Pearson correlation coefficient and we performed both a bivariate correlation as well as a partial correlation to control for age and IQ[22]. The IQ score was computed as the sum of the correct answers on the ICAR sample test[78]. Additionally, we also performed repeated-measures ANOVAs with within factor horizon and a between participants variable [impulsivity/ADHD-symptoms] to assess these effects further.

**Step 2.1.** First, we looked at impulsivity within a broad spectrum, and expected it to be linked with value-free random exploration. For this, we used the total score on the Barratt Impulsiveness Scale (BIS). The BIS is the most commonly administered self-report measure for assessment of impulsiveness[49], and has already been used in online studies[40,41]. We looked at the correlation between the BIS total score and the low-value bandit frequency, and between the BIS total score and the $\epsilon$-greedy parameter. These associations allowed us to conclude that value-free random exploration is linked to impulsivity traits in general, which has implications for impulsivity disorders beyond ADHD.

Considering that impulsivity is a broad heterogenous construct[1,3–5], in Stage 2 we performed an exploratory analysis of the three subdomains of BIS (i.e., attentional, motor, and non-planning behaviour[88]) similarly to previous studies[89]. We investigated whether value-free random exploration is linked to a specific subdomain by looking at the correlations with each of them. Specifically, we looked at the correlation (corrected for multiple comparisons using Bonferroni correction) between the low-value bandit frequency and the BIS subdomains: attentional, motor and non-planning, as well as the correlation (corrected for multiple comparisons using Bonferroni correction) between the $\epsilon$-greedy parameter and the BIS subdomains: attentional, motor and non-planning.

**Step 2.2.** Second, we looked at ADHD symptoms across our sample and expected to find an association of higher ADHD scores being related to increased value-free random exploration. This analysis extends our previous preliminary findings showing a positive association in youths (9–18 year olds) between ADHD traits (the Conners ADHD questionnaire[80]) and value-free random exploration[22]. We looked at the correlation between the ASRS total score and the low-value bandit frequency, and between the ASRS total score and the $\epsilon$-greedy parameter. It allows a definitive answer to the hypothesis whether ADHD symptoms are linked to value-free random exploration[12,21,23]. The ADHD measure we used was the total score on the Adult ADHD Self-Report Scale (ASRS), a questionnaire which was developed by the World Health Organization and is used for screening ADHD in the general population[72]. In Stage 2 we additionally performed an exploratory analysis of the sub-scales of the ASRS (i.e., inattention, hyperactivity-impulsivity). Specifically, we looked at the correlation (corrected for multiple comparisons using Bonferroni correction) between the low-value bandit frequency and the ASRS sub-scales: inattention and hyperactivity-impulsivity. We also examined the correlation (corrected for multiple comparisons using Bonferroni correction) between the $\epsilon$-greedy parameter and the ASRS sub-scales: inattention and hyperactivity-impulsivity.

**Step 3 (Stage 2)**

In Stage 2, we performed a further exploratory step. In order to investigate whether there exists a latent trans-diagnostic structure which can help to explain exploration differences, we performed a factor analysis. First, we used the raw scores from all questionnaire items as variables to reduce their dimensionality similarly as previous studies[40–42]. Factor analysis was conducted using the fa() function from the Psych package in R, with an oblique rotation (oblimin; we draw the reader's attention to the fact that the factanal() function, which does

not allow for such rotation, was erroneously mentioned in the Stage 1 protocol). The number of factors was based on the Cattel's criterion[90], using the Cattell-Nelson-Gorsuch test (nFactors package in R). Factors were labelled based on the items which loaded the most strongly in a consensus discussion among the authors.

First, we expected our two impulsivity questionnaires (BIS and ASRS) to primarily load onto one factor, and we expected this factor to be at least as much associated with value-free random exploration as the impulsivity/ADHD questionnaires alone (cf. above). In addition to the hypothesized increase in value-free random exploration, we investigated using multiple comparison whether impulsivity correlates with other forms of exploration (e.g., complex strategies).

As a second step, we investigated whether exploration correlates with other factors. In particular, similar to previous studies[40–42,50,51], we expected to retrieve a depression / anxiety dimension, on which depression, social anxiety and anxiety would load onto (SDS, LSAS and STAI-Y2 questionnaires respectively) and a compulsivity dimension, on which OCD and uncertainty intolerance traits load onto (OCIR and IUS questionnaires) respectively. Indeed, previous research has found that impulsivity and compulsivity only show a modest overlap[63], which is also why previous studies that used factor analyses have found that these items load onto different factors[40,42]. Previous studies have demonstrated increases in exploration in OCD patients[44,91], but it is not clear which exploration strategy is concerned. We therefore looked at the correlation (corrected for multiple comparisons using Bonferroni correction) between the compulsivity dimension and each exploration free parameter (depending on the model). Similarly, studies have demonstrated abnormality in exploration in patients with depression[47], anxiety[48] and other disorders related to avoidance of uncertainty[45]. However, different exploration strategies have not been tested. We therefore looked at the correlation (corrected for multiple comparisons using Bonferroni correction) between the depression/anxiety dimension and each exploration free parameter (depending on the model). We also investigated two separate questions. First, we looked at the correlation between the autism scale, AQ-10 total score and value-free random exploration, as autism has overlapping symptoms with ADHD[92]. An association between the autism score and our impulsivity measure would have resulted in further analysis using partial correlations. Second, we looked at the correlation between the cognitive flexibility scale (CFS) and value-free random exploration, as cognitive flexibility is thought to play a role in the exploration-exploitation trade-off[93,94].

## Reporting summary

Further information on research design is available in the Nature Research Reporting Summary linked to this article.

## Data availability

The raw (anonymized) and processed data are available at Github https://github.com/MagDub/Mfweb-data and Zenodo: https://doi.org/10.5281/zenodo.6522060[95]. The pilot data are available at Github https://github.com/MagDub/Mfweb-pilot_data and Zenodo: https://doi.org/10.5281/zenodo.6522062[96]. Source data are provided with this paper.

## Code availability

Code for power simulations, computational modelling and data analysis can be found on Github: https://github.com/MagDub/MFweb-data_analysis and Zenodo: https://doi.org/10.5281/zenodo.6445661[97].

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

## Acknowledgements

We thank Vasilisa Skvortsova for her help with implementing the exploration task online. M.D. is a predoctoral fellow of the International Max Planck Research School on Computational Methods in Psychiatry and Ageing Research. The participating institutions are the Max Planck Institute for Human Development and the University College London (UCL). T.U.H. is supported by a Wellcome Sir Henry Dale Fellowship (211155/Z/18/Z), a grant from the Jacobs Foundation (2017-1261-04), the Medical Research Foundation, and a 2018 NARSAD Young Investigator Grant (27023) from the Brain and Behaviour Research Foundation. T.U.H. has also received funding from the European Research Council (ERC) under the European Union's Horizon 2020 research and innovation programme (grant agreement No 946055). The Max Planck UCL Centre is a joint initiative supported by UCL and the Max Planck Society. The Wellcome Centre for Human Neuroimaging is supported by core funding from the Wellcome Trust (203147/Z/16/Z).

## Author contributions

Conceptualization, M.D., and T.U.H.; Methodology, M.D., and T.U.H.; Software, M.D.; Formal Analysis, M.D., and T.U.H.; Investigation, M.D.; Data Curation, M.D.; Writing – Original Draft, M.D., and T.U.H.; Writing – Review & Editing, M.D. and T.U.H.; Supervision, T.U.H.; Funding Acquisition, M.D., and T.U.H.

## Competing interests

The authors declare no competing interests.
