## [Peer Review File · Nature Communications]

Value-free random exploration is linked to impulsivityReview Study Plan (Stage 1)

Reviewers' Comments:

Reviewer #1 (Remarks to the Author):

This is an interesting and original research plan based on a generally sound literature review, suitable pilot data and adequate statistical power analysis. The relationship between what the author call "impulsivity" and different facets of their dissection of "exploration" is important and may significantly illuminate the neurocognitive substrates of several neuropsychiatric disorders. The table of hypotheses, possible outcomes and interpretations is in many ways exemplary. The methodology looks straight forward and would be fairly easy to replicate. Whether of course the data obtained by this online methodology in reality conforms to these outcomes is difficult to say and I would have to reserve judgement about its future publishability in *Nature Communications*, although I am sure the project is valid and worthwhile. I have a few specific substantive points and a more minor query:

Specific points

1. Impulsivity itself is a diverse phenomenon as the authors' reference to reference 1 (and other more recent uncited references) makes clear. Thus, the exercise may in the end devolve to finding out which aspect of exploration relates to which aspect of impulsivity. The Barratt Impulsiveness Scale is a rather broad-brush instrument that taps into several constructs and often does not correlate highly with the type of objective discounting measures analogous to what is being deployed by the authors.
2. It was at times a little unclear to me whether the authors were really as interested in impulsivity as ADHD which of course includes a broader range of behavioral change. Can this be clarified? I do see that the proposed research is highly relevant to ADHD.
3. Impulsivity is often related to a loss of inhibitory cognitive control and exploration to cognitive flexibility; inhibitory control and cognitive flexibility are generally found to be correlated rather than anti-correlated and so it could be hypothesised in some quarters that impulsivity is related to cognitive inflexibility rather than cognitive flexibility.
4. One must remember that although significant relationships are often found between these constructs they are often very small (e.g. $r=0.1-0.25$) and thus account for rather little of the shared variance. How do the authors consider this?
5. A suggestion is made that OCD patients are less exploratory but the recent Kanen paper cited appears to show reduced stickiness in that situation, i.e. increased 'exploration'. How does that affect your possible theorising?

Reviewer #2 (Remarks to the Author):

I am delighted to review this Stage 1 registered report.

As outlined in the abstract, the authors' intention is 'Capitalising on recent advances in disentangling distinct human exploration strategies, we will assess whether impulsivity, or related psychiatric dimensions, are associated with specific forms of exploration.'

(1) Potential significance if implemented well.

This is a worthwhile endeavor.

Computational psychiatry studies have advanced our understanding of how the component processes of cognition go wrong in psychopathology. Initial studies used only a single trait (e.g. impulsivity) or diagnostic group (e.g. ADHD). More recently, a great advance has been the use of multiple dimensional measures of symptomatology and application of dimension reduction techniques to pull out latent factors of psychopathology and relate them to computational parameters obtained by modelling participants' performance on cognitive tasks, in particular from the decision-making field. There are still only a few examples of this important new approach. Work by Gillan/Daw and Gershman among others have related (i) performance of a task designed to differentiate model-based from model-free decision-making and (ii) performance on an effort-based decision-making task to latent dimensions extracted using factor analysis of items from a range of self-report measures of

psychopathology (the factors extracted being compulsive behavior and intrusive thought, anxiety/depression and social withdrawal). To my knowledge this approach has not been applied to relate psychiatric dimensions to exploration/exploitation – hence this proposed study has the potential to make an important addition to the literature.

Implementation concerns

(2) Task. The authors have designed a novel task to tease apart different forms of exploration (in addition to examining preference for exploitation). This task builds on a tradition of exploration/exploitation tasks recently published (e.g. work by Nathaniel Daw, Bob Wilson among others). The authors reference a previous paper in which they first present this task – I found this on bioRxiv (<https://www.biorxiv.org/content/10.1101/2020.02.20.958025v3>) as far as I can tell it is not peer reviewed / published as yet.

On each trial, participants are presented with 3 out of 4 possible ‘trees’ (bandits) and one of two possible horizons (i.e. they will get to choose between the trees once or six times). Three bandits have approximately the same reward value distribution (means and variance matched). One [novel] shows no prior draws (i.e. no information as to bandit value on a given trial), one [high value, standard] shows one prior draw and one [high value, high information] shows three prior draws. The 4th bandit ‘low value’ has a much lower reward value distribution (the mean is I believe set to be below the minimum of the means of the other 3 bandits) and one prior draw.

The authors argue that increased drawing from the high value bandit ‘is an evident signature of exploitation (choosing maximal expected value)’ – here as far as I can tell they are collapsing the one and three samples shown bandits (that I call ‘standard’ and ‘high information’ above). Meanwhile, novelty-based exploration is indexed by increased selection of the novel bandit at the long horizon and ‘tabula-rasa exploration’ by increased selection of the low-value bandit.

The potential problem I see here is that reward value and information might be confounded. The ‘high value’ bandit also has the highest mean information (either 1 or 3 samples versus 1 or 0 for the other two bandits). In addition, the authors collapse across trials regardless of which 3 bandits are shown. The apparent preference towards one bandit might actually reflect a preference away from the other bandits presented and it is hard to disentangle with this collapsing procedure.

I think the task would be greatly improved by orthogonalizing reward value and information –by inclusion of novel (0 sample) low value and 3 sample low value (i.e. low value, high information) bandits. If the authors now compare low and high value trials they are not confounded by information – so preference towards the high value ones might now more reliably index exploitation versus exploration.

2) Modeling. The authors do not include sections on model comparison, parameter recovery, model identification etc in this registered report. I believe this is important to do as my understanding is that this is the stage at which reviewers are meant to evaluate the modeling.

The authors do however state that these sections can be found in their prior paper (the bioRxiv preprint) – I hope it is therefore okay to comment on these sections as presented there.

2a) Model comparison – the authors previously compared a ‘Thompson’ model, a ‘UCB’ model and a hybrid model with or without the addition of a tabula-rasa ϵ -greedy parameter and a novelty bonus η . Within each class, the model with both added parameters performs best. Critically, there is very little difference in performance between the three classes of model at a group level (looking at the model version with both these additional parameters in each case). In a supplementary figure in the bioRxiv preprint, the authors report that 27 participants were best fit by the Thompson class model, 22 by the UCB class model and 10 by the hybrid model. Model performance in each case is evaluated using MLE and cross validation (10fold, conducted by dividing up trial into subsets.)

My concerns here are –

i)-Cross validation– participants are highly likely to be learning across trials and performance is hence unlikely to be independent across trials.

ii) Modeling learning effects - relatedly, ideally the authors would include parameters in their model to examine influence of trial number and the interaction of this with the other parameters – so addressing whether participants change their preference for different forms of exploration with time on

task.

iii) It is not possible to tell from the information presented if 27 participants are just slightly better fit by the Thompson model and 22 by the UCB or if there are two clear subsets of participants using different strategies. If the latter, then it would be suboptimal to fit these participants using a model that does not capture how they are performing the task. This is particularly an issue when relating parameter estimates (used to capture aspects of behavioral performance) to individual differences (e.g. psychopathology indices) as it could falsely give the impression that certain individuals are engaging less or more than others in a given form of exploration if the modelling of a key parameter does not capture their performance successfully.

iv) One way to address this would be to switch to a hierarchical modelling framework and allow for there to be 2 different models adopted by different subgroups of participants – then to compare this against only allowing for 1 model.

v) In addition, for novelty and tabula-rasa parameters, it would be useful to correlate the estimates of these from the Thompson versus UCB models across participants to see if they are equivalent. (Even if they are, this still leaves the issue however of how to examine information-based exploration equivalently across participants best fit by the Thompson vs UCB models).

vi) In the current paper, in figure S4, the authors state that the Thompson model does best, however it is hard to see why they claim this from the figure. Here again, it would be helpful to know how many individuals are best fit by each model and to also see the results once a hierarchical modelling framework is used.

2b) Model identification. No model identification analysis is provided. This addresses the issue of whether we can even tell the models apart enough for the above comparisons to be meaningful. This is done by simulating data with one model and then fitting it with all three models (UCB & n & e; Thompson & n & e; Hybrid & n & e) and seeing how often the model that was used to simulate the data is recovered (i.e. gives the best fit).

2c) Parameter recovery – only 4 parameter values are used for this according to the bioRxiv preprint (despite the figure legend saying 10) – it would be better for values to be drawn randomly across the whole range. In a hierarchical framework, these values could be drawn from population level posterior distributions for each parameter.

3) Relating impulsivity to model parameter estimates – multiple comparisons.

The authors describe 3 measures (ADHD symptoms and scores on the BIS -11 and ASRS) they will relate to model parameter estimates. They also state they will perform bivariate correlations and partial correlations controlling for age and IQ. This is 6 tests – have the authors power corrected for performing multiple comparisons? Are the authors uniquely predicting a relationship with tabula-rasa exploration? Using the model based parameter? If they also plan to interrogate the relationship with other model parameters and/or non model-based indices of tabula-rasa exploration (e.g. choice of the low value bandit in the long vs short horizon) then will they multiple comparison correct for this? This also needs detailing.

4) A broader analysis of psychopathology.

a) In their table of planned comparisons, the authors seem to suggest they expect to find separate impulsivity and compulsivity factors. On what basis? Are appropriate measures included to facilitate this?

b) On a related note, it looks like the authors are not planning to include all the self-report measures used by both Gillan and Gershman. Why is this? It might well change the ability to identify the same latent factors. Have the authors any pilot data on the latent factors they can extract based on the subset of measures they plan to use?

c) Do the authors plan to do a hypothesis-free comparison of all latent factor scores (3+?) to all indices of exploration/exploitation (all model parameters? Also non model based indices?) – can they set this out now so it is apparent how many tests are being conducted and whether power analyses have been corrected for this.

d) Are the authors hoping to be able to claim that a given model parameter is related to one latent factor and not another? If so, given they plan to use oblique rotation (e.g. a correlated factors

approach) they should do power calculations taking into account the anticipated degree of correlations between factors.

For this, it would help to have pilot data on the self-report measures, the latent factors extracted and their degree of correlation.

Reviewer #3 (Remarks to the Author):

This is a signed review (Chris Chambers, Cardiff University).

I enjoyed reading this Stage 1 Registered Report – it tackles an important question in the area of impulsivity and cognitive control, with implications for theory and the field of computational psychiatry. The study is carefully designed and the manuscript is clearly written.

My main comments are methodological and address the mapping between the hypotheses, statistical tests, and falsifiability. All issues in my review are addressable through revision.

1. In reviewing the design table, it was not always clear to me which outcomes would be considered to confirm or disconfirm the specific hypotheses. For example, taking the first hypothesis (that participants will select bandits with lower expected values in the long vs short horizon), the authors appear to propose five t-tests – thus the scientific hypothesis does not map directly on to a statistical hypothesis – and the “interpretation” column does not make clear which tests within this nested structure must reveal evidence of a difference (or not) for the hypothesis to be confirmed or disconfirmed. I also didn’t understand the language (used here and elsewhere) “absence of horizon effect may reflect meaningful between-subject variation” – does this imply that the authors consider the hypothesis irrefutable and that any non-significant outcome instead indicates that a true effect is obscured by individual differences? If so (and I accept I may have misunderstood) then I think this is problematic from the point of view of falsifiability. To make this clear, I suggest plotting the predicted results for each of the DVs (or listing them as directional formulae, e.g. $X > Y$) and then making clear which outcomes (or combination of outcomes) must be observed to confirm or disconfirm the overall prediction. The authors may also want to list specific statistical tests as sub-hypotheses (e.g. hyp 1a, 1b) and then explain which sub-hypotheses must be confirmed for the overarching “hypothesis 1” to be confirmed.

2. Also on the first hypothesis (and as implied above I would suggest numbering these predictions to make more readily trackable), I didn’t understand how this prediction was tested: “In terms of bandit types, they select the low-value and novel bandit more, and the high-value bandit less.” Which test interrogates this hypothesis? Please ensure a complete and precise correspondence between every prediction and the statistical test(s) that of that prediction.

3. Further down the design table, turning to the question of whether participants deploy exploration heuristics, the authors write in the interpretation column: “All our previous studies show clear effects, thus an absence of effect is not expected. If we will not find evidence for some of the strategies, we will focus subsequent analyses on the strategies we find evidence for. Moreover, we will explore whether high impulsive subjects have a different winning model from the low impulsive subjects.” This feels a little off balance to me. I can accept that the hypothesis may have a strong prior, but the interpretation column should be free of such baggage and simply make clear what evidence would demonstrate the hypothesis to be supported or unsupported, rather than explaining why that might or might not be (which is best reserved for the Discussion). I would also remove comments about the plans to explore the data given different outcomes – this is a given (and could be reported in the Results as an exploratory post hoc analysis) but in my view doesn’t add any information to inform the precision, precommitment or falsifiability for the confirmatory plans.

4. Most of the power analyses draw on point effect size estimates from the pilot data. In my view, power analyses should generally not be based solely on the effect size estimate of a pilot study but on

the minimal plausible yet theoretically informative value or the lower bound estimate extracted from a set of previous studies (including the pilot). Basing the power analysis on a single point value is inadvisable because the estimate of the effect size from a small pilot study is consistent with a range of values, including effects much smaller than the estimate that may nonetheless be theoretically informative. That said, this issue is mitigated by the fact that the actual sample size (580) will be much higher than needed for most of the hypothesis tests. I would therefore suggest also reporting a sensitivity power analysis for the hypothesis tests that are based on the pilot data, to report what effect size the test has 95% power to detect at the actual N of 580. These effect sizes will be substantially smaller than the observed effect sizes in the pilot study and, e.g. are likely to be below the lower bound 95% CI (see point 7).

5. The authors will use a number of assessment instruments and then propose to covary for individual scores in step 2 or include them in a factor analysis in step 3. I would like to see a little more precision in the specification of which scores will (and won't) be included in the factor analysis in step 3. At present, the analysis in step 3 is described relatively vaguely compared to steps 1 and 2. It may be that step 3 is best reserved for Stage 2 exploratory analyses and removed entirely from the Stage 1 manuscript, but I will defer to reviewers who are more expert on FA to offer a view.

6. Will excluded participants be replaced? I would suggest doing so to ensure that the final sample meets the required minimum sample size.

7. In Table S1, could the authors also report the 95% confidence interval around the effect size estimate? I'm not sure what is currently reported in the CI ranges, but a CI around the effect size would be informative.

Reviewer #4 (Remarks to the Author):

I'm very grateful for the opportunity to read and review this pre-registered report. The research questions are clear and engaging, and cover a nice balance between replication and exploration. I applaud the authors for opting to go the route of preregistering the hypotheses and analyses. Below, I provide feedback and suggestions about how to improve the manuscript. I separate my feedback to address the experiment, the models, the analysis, and then some minor comments.

Experiment:

There are several aspects of the task that were not clear to me. First of all, is it the case that participants experienced 200 trials where they made one choice (short horizon) and 200 trials where they made six choices (long horizon)? This would mean that there is substantially more data for the long horizon choices, which may violate some of the statistical assumptions (i.e., your paired t-tests assume equal variance). And since there is no value to exploration in the short horizon trials, why then do you still see some novelty/exploration?

Secondly, how are the mean rewards for each tree determined? Do they vary from trial to trial? What is the range of reward values? Does the global maxima stay fixed across trials? These are important considerations, since participants could use meta learning to determine whether exploring the novel option pays off (e.g., if the pre-samples of the high value tree are already at the max apple size, you already know that the novel tree won't be better). In order to properly measure exploration, there needs to be uncertainty about the reward values. Some previous studies have used random scaling of reward values from round to round (e.g., Wu et al., 2020) to maintain uncertainty about which option has the highest mean reward. It would also be helpful to see the influence of trial and horizon on score, as a diagnostic plot. The communication of reward values to participants was also unclear. Are reward values indicated by the radius, circumference, or the area of the apple? Is the color of the apple reflecting its size (e.g., as a visual aid similar to Wu et al., 2018), or is it indicating which tree it comes from? The color mapping could have some unintended effects, since green is associated with

unripe apples.

Lastly, how long do you expect the experiment to take? Based on $(200 \text{ trials} * 1 \text{ choice in short horizon cond}) + (200 \text{ trials} * 6 \text{ choices in long horizon cond}) = 1400$ choices given by participants, averaging at 3000 ms per choice (Fig. S1b), that would be 70 minutes just for response times in the main task, without any instructions. It's also not clear how long the questionnaires will take, but I worry about decision fatigue and drop out. This is likely to disproportionately affect ADHD participants, who by definition have difficulties with sustained attention. This may lead to a self-selection bias and a majority healthy control population in your final sample. And although I feel there are already too many questionnaires, I wonder if you would want to replace one with an autism scale, since that is the closest to ADHD, and provide valuable differentiation. Also, you may want to ask participants if they are taking medication, because that may mask ADHD-like behavior.

Models:

As the authors themselves admit, what was supplied in the supplementary information is not sufficient for understanding/reproducing the models. I hope that will be addressed in a later version. The simple inclusion of the description that their models are time-invariant Kalman filters would have been helpful, instead of directing the reader to a different preprint by the authors, which largely contains the same textual description of the models. The concept "tabula rasa" exploration should also be explained in the introduction of the current report, since it involves important theoretical commitments.

In the "Subjects explore using heuristics" section, a t-test is reported, but it's not clear what is being compared. Is this the winning model compared against the next closest model? Fig. S4 suggests that the model results are rather inconclusive, and doesn't seem to match the test statistics. Also, if you are comparing likelihoods (bounded between $[0,1]$), the normality assumptions of the t-test may not be met. You would do better to use a t-test against the negative log likelihoods, or better yet, to use a Bayesian model selection framework and compute the protected exceedance probability (Rigoux et al., 2014).

Along these lines, I was a bit surprised to learn that your UCB model has as many as 4 different forms of exploration. The softmax tau, the information bonus gamma, the tabula rasa exploration epsilon, and the novelty bonus eta. From eyeballing Fig. S4, this model seems to perform quite similarly as your winning model. While the introduction is quite committed to interpreting the model results in terms of novelty vs. tabula rasa exploration, I'm not sure the pilot results justify the exclusion of these other two forms of exploration (information bonus and softmax temperature). I think the ideal method would be to run model recovery and parameter recovery on your pilot data (see Wilson & Collins, 2019) to find out if your experiment can in fact distinguish between these models and whether these parameters are in fact capturing distinct and interpretable phenomenon. This would substantially improve the interpretability of your model results.

Analysis:

The analysis plan seems detailed, and both the hypotheses and interpretations of alternative outcomes seem well thought-out. I would suggest however, that t-tests might not be the ideal mode of analysis for all of these research questions. While the 3 separate steps of analysis make a lot of sense, there are potential interactions between variables that would be missed. I believe a mixed effects regression framework would be superior in many ways. For instance, you could see if your ADHD/impulsivity variables interact with the exploration and horizon length variables from step 1. A mixed effects approach would also be beneficial in step 3 because you could treat participants as random intercepts to account for individual biases in the survey results. Additionally, when comparing parameter estimates, a Wilcoxon signed-rank test would provide a rank-based alternative to a paired t-test, which wouldn't require assumptions of normality that might be violated by your bounded parameter estimation procedure. Lastly, I was wondering how you intend to avoid overfitting or spurious correlations in your factor analysis? With a sample size of 580 and a large number of survey questions, some type of statistical correction seems crucial.

Minor comments

- There are parts where the writing is fragmented or grammatically confusing. I know this isn't the final paper, but I believe this journal has rather high standards, and any future submission would benefit from more proofreading and editing
- Heuristic strategies require <relatively> less computation. I would avoid using the absolute term "computationally light"
- What is an "ordinary t-test"? Do you mean unpaired?
- Please report 95% CI in future plots, since your large sample size will make SEM meaningless
- Why the 1500ms RT exclusion criteria? Maybe I missed something, but the rationale wasn't clear. Also, the average 3000 ms RT seems really long, but perhaps I'm missing some detail about how it is measured.
- The first sentence in the Fig. S5 caption suggests that the parameters shown here were fit in a different method from the K-fold CV method described elsewhere? If true, this seems quite strange

I sign my review

- Charley Wu

References:

Rigoux, L., Stephan, K. E., Friston, K. J., & Daunizeau, J. (2014). Bayesian model selection for group studies—revisited. *Neuroimage*, 84, 971-985.

Wilson, R. C., & Collins, A. G. (2019). Ten simple rules for the computational modeling of behavioral data. *Elife*, 8, e49547.

Wu, C. M., Schulz, E., Garvert, M. M., Meder, B., & Schuck, N. W. (2020). Similarities and differences in spatial and non-spatial cognitive maps. *PLOS Computational Biology*, 16, 1–28.
<https://doi.org/10.1371/journal.pcbi.1008149>

Wu, C. M., Schulz, E., Speekenbrink, M., Nelson, J. D., & Meder, B. (2018). Generalization guides human exploration in vast decision spaces. *Nature human behaviour*, 2(12), 915-924.

R.0. We thank the reviewers for their positive evaluation of our Stage 1 Registered Report and we appreciate the helpful suggestions. We have now addressed all raised concerns, implemented the suggestions and conducted additional simulations and analyses of the pilot data in light of these comments.

In short, we address comments about reward and information being confounded, we detail our computational modeling pipeline and demonstrate that the relevant models (i.e. with or without heuristics) are well distinguishable. We now also acknowledge the diverse nature of impulsiveness and have substantially expanded our statistical analysis section (including the sensitivity power analysis). We would also like to mention that based on a reviewers' comment on a previous manuscript, we have now changed the wording 'tabula-rasa exploration' to 'value-free random exploration', as this is a more precise term. We report all changes in the revised manuscript and trust that it meets the rigorous standards of your journal.

Reviewers' Comments:

Reviewer #1 (Remarks to the Author):

This is an interesting and original research plan based on a generally sound literature review, suitable pilot data and adequate statistical power analysis. The relationship between what the author call "impulsivity" and different facets of their dissection of "exploration" is important and may significantly illuminate the neurocognitive substrates of several neuropsychiatric disorders. The table of hypotheses, possible outcomes and interpretations is in many ways exemplary. The methodology looks straight forward and would be fairly easy to replicate. Whether of course the data obtained by this online methodology in reality conforms to these outcomes is difficult to say and I would have to reserve judgement about its future publishability in Nature Communications, although I am sure the project is valid and worthwhile. I have a few specific substantive points and a more minor query

R1.0: We thank the reviewer for their very positive evaluation. We are pleased to see that this reviewer considers our efforts 'exemplary'. We have now implemented all of the reviewer's suggestions and detail how we address each point below.

Specific points

1.1. Impulsivity itself is a diverse phenomenon as the authors' reference to reference 1 (and other more recent uncited references) makes clear. Thus, the exercise may in the end devolve to finding out which aspect of exploration relates to which aspect of impulsivity. The Barratt Impulsiveness Scale is a rather broad-brush instrument that taps into several constructs and often does not correlate highly with the type of objective discounting measures analogous to what is being deployed by the authors.

R.1.1. We thank the reviewer for raising this important point. Impulsivity is indeed a broad and heterogenous construct (e.g. Evenden, 1999; Caswell et al., 2015; Dalley, Everitt, Robbins, 2011; Dalley & Robbins; 2017) and, as mentioned by the reviewer, the Barratt Impulsiveness Scale (BIS) taps into several aspects of it. We decided to use the BIS as an instrument, because of its wide-spread use in research, because we have previously shown that it relates to important neurocognitive processes (e.g. Ziegler*, Hauser* et al., 2019, Nature Neuroscience), and importantly because its total score captures an overarching impulsivity construct aggregating across different impulsivity sub-domains.

As very little is known about the associations between impulsivity and exploration, we believe that we first need to establish a link between exploration and impulsivity in general. Only subsequently can we refine our analysis and look at the three BIS subscales in more detail (i.e. attentional, motor and nonplanning behaviour; Patton et al., 1995), similarly to previous studies (e.g. Frey et al., 2017). As we have no strong prior hypothesis on which specific subscale will be associated to exploration, we will correct these analyses for multiple comparisons. As these constitute exploratory hypotheses, we will conduct them in our Stage 2 analysis. We have now detailed this further in the methods section of the revised draft. We also mention the heterogeneity of impulsivity in the introduction and provide reference to the relevant literature.

We agree with the reviewer that associations between questionnaire-based measures and task-based behaviours often show moderate correlations (similar to the mentioned temporal discounting correlation with impulsivity; e.g. Dougherty et al., 2002; Reynolds et al., 2006; Mobini et al., 2007; Baumann & Odum, 2012). We believe that this is not a concern per se because these concepts are captured by using entirely different measures (self-report questions vs behavioural probes) and differ in many aspects (e.g. time-scale: reports on last few weeks vs instantaneous behaviour). Critically, our power calculations fully embrace this fact as our calculations are based on low to medium correlations, in accordance with the previous literature (Rollwage et al, 2018; Rouault et al., 2018; Seow & Gillan, 2020). We are thus well equipped to detect associations, and to establish a mechanistic link between impulsivity and exploration. We now clarify this in the revised manuscript.

Introduction: “*Impulsivity is often cast as acting without thinking and is traditionally assessed using self-report questionnaires^{1,2}. It is a broad and heterogenous construct^{1,3-5} whose relevance not only comes from the observation of a substantial variation among a ‘healthy’ population, but also its importance in psychiatry*”

Methods: “*For the 2nd step’s sample size estimation, where the link between exploration and impulsivity is investigated, the correlation coefficient of our previous study using the same task²² was used for our power analysis. In this study, a Pearson correlation of $R=0.26$, $p=0.01$ was observed between an impulsivity measure⁶³ (the Conners ADHD questionnaire) and value-free random exploration in youths. Assuming a similar correlation in adults, our power analysis suggests that a sample of $N=190$ is sufficient to reach 95% power (G*Power suggests a similar sample size of $N=186$). This moderate size correlation factor is in line with previous studies linking BIS-measured impulsivity to behaviour (e.g. with delay discounting⁶⁴⁻⁶⁷). [...]. Considering that impulsivity is a broad heterogenous construct^{1,3-5}, in Stage 2 we will perform an exploratory analysis of the three subdomains of BIS (i.e. attentional, motor and non-planning behaviour⁶⁹) similarly to previous studies⁷⁰. We will investigate whether value-free random exploration is linked to a specific subdomain by looking at the correlations (corrected for multiple comparison) with each of them.*”

1.2. It was at times a little unclear to me whether the authors were really as interested in impulsivity as ADHD which of course includes a broader range of behavioural change. Can this be clarified? I do see that the proposed research is highly relevant to ADHD.

R.1.2. We apologise for our lack of clarity. In this paper, we take a dimensional approach investigating the broad spectrum of impulsive variability in the general population. Impulsivity varies substantially amongst the general population (e.g. Chamorro et al., 2012; Robbins et al., 2012; Ziegler, Hauser et al. 2019), with highest scores for individuals with high mental health problems (Stanford et al., 1994), such as with ADHD (Rodriguez-Jimenez et al., 2006; Winstanley et al., 2006). To exploit the full variability of impulsivity in our sample, we primarily investigate impulsivity in the 1st part of our analysis (step 2.1. in the methods). To this end, we will investigate impulsivity using the BIS total score. Because we believe that these findings may have important implications for ADHD in particular (cf. Dubois, Bowler, et al., 2020), we will look at the association between exploration and ADHD in a more refined and targeted way in the 2nd part (step 2.2.). Our dual approach thus embraces the well-established dimensional perspective on psychiatry, allowing us to further assess whether exploration is linked to impulsivity in general, or only to ADHD-specific symptoms. We have now clarified this, in the Design Table (cf. Table 1) as well as throughout the rest of the manuscript.

Introduction: “*In this study, we want to put the large body of theoretical work to test and exploit the recent advances in exploration to investigate empirically the link between impulsivity and exploration. Importantly, we will investigate impulsivity within a broad spectrum across the general population and as a more specific ADHD-related impulsivity. To provide a clear answer, we will be using a pre-registered, dimensional approach using online testing in a large sample.*”

Methods: “*First, we will look at impulsivity within a broad spectrum, and expect it to be linked with value-free random exploration. For this, we will use the total score on the Barratt Impulsiveness Scale (BIS-11). The BIS is the most commonly administered self-report measure for assessment of impulsiveness⁵³, and has already been used in online studies^{41,42}. [...]. Second, we will look at ADHD symptoms across our sample and expect to find an association with higher ADHD scores being related to increased value-free random exploration. This analysis will extend our previous preliminary findings showing a positive association in youths (9-18 year olds) between ADHD traits (the Conners ADHD questionnaire; Conners, 2008) and value-free random exploration²². It will allow a definitive answer to the hypothesis whether ADHD symptoms are linked to value-free random exploration^{12,21,23}.*”

1.3. Impulsivity is often related to a loss of inhibitory cognitive control and exploration to cognitive flexibility; inhibitory control and cognitive flexibility are generally found to be correlated rather than anti-correlated and so it could be hypothesised in some quarters that impulsivity is related to cognitive inflexibility rather than cognitive flexibility.

R.1.3. We thank the reviewer for raising this interesting point which we had not considered previously. We believe that the link between exploration and cognitive flexibility is not completely straightforward. It is indeed true that in the context of task-switching protocols, switching task is generally referred to as flexibility/exploration while focusing on the task at hand is referred to as stability/exploitation (e.g. Braem et al., 2013). However, such tasks investigate switching in the context of a cognitively challenging and complex task, which requires substantial cognitive resources. In exploration/exploitation tasks, cognitive flexibility is usually assumed to have a different role, namely, to allow the switch between exploitation to exploration in a goal-directed way (e.g. Laureiro-Martinez et al., 2010; Good & Michel, 2013). Importantly, our predictions are not that impulsivity would be related to such complex strategies, such as the complex exploration strategies that we capture in our task. Rather, our predictions are that impulsive subjects rely too much on very simple exploration heuristics that do not require substantial cognitive resources. In line with our ideas, previous studies have found that impulsivity is associated with increased avoidance of mental effort (Patzelt et al., 2019). Given the novelty of our approach, the different exploration strategies have not yet been related to cognitive flexibility, but we would hypothesise that cognitive flexibility is anti-correlated with the simple exploration strategies, as suggested by this reviewer.

As this is an interesting point, we have now decided to additionally assess cognitive flexibility in our online study and have therefore added the Cognitive Flexibility Scale (Martin & Rubin, 1995). As we consider this an exploratory hypothesis, we have not embedded this in the Stage 1 of this process, but label this as a Stage 2 exploratory analysis. We hope that this could shed new light onto the intricate association between impulsivity, exploration, and cognitive flexibility. We have now revised the manuscript to detail this better.

Methods: *“In addition, we will collect further questionnaires to investigate additional psychiatric dimensions (cf. Analysis Plan, step 3). These entail the Liebowitz Social Anxiety Scale⁵⁴ (LSAS), the State-Trait Anxiety Inventory⁵⁵ (STAI-Y2), Intolerance of Uncertainty Scale⁵⁶ (IUS), Obsessive-Compulsive Inventory-Revised⁵⁷ (OCI-R), and Zung's Self-rating Depression Scale⁵⁸ (SDS), in accordance with similar previous approaches⁴¹⁻⁴³, as well as the Cognitive Flexibility Scale⁵⁹ (CFS) and the Autism spectrum Quotient⁶⁰ (AQ-10). [...] Additionally we will explore the link between exploration and autism, as it has overlapping symptoms with ADHD⁷⁶, and between exploration and cognitive flexibility, as it is thought to play a role in the exploration-exploitation trade-off^{77,78}.”*

1.4. One must remember that although significant relationships are often found between these constructs they are often very small (e.g. $r=.1-0.25$) and thus account for rather little of the shared variance. How do the authors consider this?

R.1.4. It is indeed a common finding in such studies that the association strength is relatively small between self-report traits and behaviour in tasks. There are various influences that impact the association strength and give rise to these small-to-medium effect sizes. Firstly, as detailed above, we are investigating associations between constructs that were assessed using entirely different modalities. We assess impulsivity traits using self-report questionnaires, which assess the subject's belief about themselves, reported about a longer time horizon. On the other hand, we assess exploration using a behavioural task, in which subjects make decisions within less than one hour. These measures thus differ across many aspects, from the nature of the report, to the time horizon of the report, and a differential involvement of cognitive skills (e.g. metacognitive insight for self-reports). In addition, there are multiple sources of noise that further impact on the association strength. There is assessment and analysis noise on both the questionnaire and the task. We have put substantial effort in reducing noise on either side. We selected questionnaires that are well

established, have good psychometric properties and a high test-retest reliability. We have also optimised our task to minimise noise influences and to maximise reliability of our task parameters. For example, we have conducted substantial parameter recovery analyses, showing that our parameters are well recoverable and thus provide the best possible indicators of exploration. An additional point is that we investigate natural variability in the general population. This means that we are expecting less variability and thus lower effect sizes than if we would assess e.g. patient groups with greater differences in impulsivity. Lastly, this study is conducted online, which means that the results are likely to be somewhat noisier than if conducted in a laboratory setting. This may further reduce association strength.

Importantly, this is a mechanistic study, establishing whether there is a link between impulsivity and exploration. The goal of the study is not (yet) to use this as a predictor (such as a biomarker). This is well in line with similar previous studies (e.g. Gillan et al., 2016; Rollwage et al., 2018; Seow et al., 2020), and we believe that establishing a link between impulsivity and a specific exploration form is of substantial importance as it allows us to better understand the neurocognitive mechanisms involved in impulsivity. This study will thus pave the way for subsequent, more fine-grained studies that allow to further elaborate this association and to provide stronger associations.

Importantly, we are well aware of the relatively limited expected association strength and we factored this into our power calculations and our analysis plan. We have taken the most conservative association strength that was found in previous studies and calculated the sample size accordingly. This means that we are well equipped to find associations, even if they are of small effect size. We now explain this in more detail in the revised manuscript.

Methods: *“These previous studies have observed correlations from $R=0.15$ (negative association between dogmatism and metacognitive sensitivity⁴⁰) up to correlations of $R=0.25$ (association between confidence and compulsivity⁴²). The relatively small effect sizes can be explained by the higher noise associated with large online samples as well as the precision of behavioural and questionnaire measures. In this study we account for this and consider the study as a first step to establish associations between measures. We conducted thorough effect size and power calculations. The lowest correlation ($R=0.15$) was used to extend our power analysis, which suggests that a sample of $N=580$ is sufficient to reach 95% power (Supplementary Information, Fig. S12; G*Power suggests a similar sample size of $N=571$).”*

1.5. A suggestion is made that OCD patients are less exploratory but the recent Kanen paper cited appears to show reduced stickiness in that situation, i.e. increased 'exploration'. How does that affect your possible theorising?

R.1.5. We thank the reviewer for noticing this and apologise for allowing this slip. Indeed, patients with OCD have been shown to have a reduced stickiness parameter (i.e. exploitation) – including in our own work (Hauser, et al., 2017), suggesting increased exploration in patients. We have now corrected this. Please also note that based on the other reviewers' comments (cf. R.3.5), we have now decided to investigate these associations in Stage 2.

Analysis plan: *“In terms of exploration, studies⁴⁷ have demonstrated increases in OCD patients^{45,75} and depression⁴⁸, as well as abnormalities in anxiety⁴⁹ and other disorders related to avoidance of uncertainty⁴⁶.”*

Reviewer #2 (Remarks to the Author):

I am delighted to review this Stage 1 registered report.

As outlined in the abstract, the authors' intention is 'Capitalising on recent advances in disentangling distinct human exploration strategies, we will assess whether impulsivity, or related psychiatric dimensions, are associated with specific forms of exploration.'

(1) Potential significance if implemented well.

This is a worthwhile endeavor.

Computational psychiatry studies have advanced our understanding of how the component processes of cognition go wrong in psychopathology. Initial studies used only a single trait (e.g. impulsivity) or diagnostic group (e.g. ADHD). More recently, a great advance has been the use of multiple dimensional measures of symptomatology and application of dimension reduction techniques to pull out latent factors of psychopathology and relate them to computational parameters obtained by modelling participants' performance on cognitive tasks, in particular from the decision-making field. There are still only a few examples of this important new approach. Work by Gillan/Daw and Gershman among others have related (i) performance of a task designed to differentiate model-based from model-free decision-making and (ii) performance on an effort-based decision-making task to latent dimensions extracted using factor analysis of items from a range of self-report measures of psychopathology (the factors extracted being compulsive behavior and intrusive thought, anxiety/depression and social withdrawal). To my knowledge this approach has not been applied to relate psychiatric dimensions to exploration/exploitation – hence this proposed study has the potential to make an important addition to the literature.

Implementation concerns

(2) Task. The authors have designed a novel task to tease apart different forms of exploration (in addition to examining preference for exploitation). This task builds on a tradition of exploration/exploitation tasks recently published (e.g. work by Nathaniel Daw, Bob Wilson among others). The authors reference a previous paper in which they first present this task – I found this on bioRxiv (<https://www.biorxiv.org/content/10.1101/2020.02.20.958025v3>) as far as I can tell it is not peer reviewed / published as yet. On each trial, participants are presented with 3 out of 4 possible 'trees' (bandits) and one of two possible horizons (i.e. they will get to choose between the trees once or six times). Three bandits have approximately the same reward value distribution (means and variance matched). One [novel] shows no prior draws (i.e. no information as to bandit value on a given trial), one [high value, standard] shows one prior draw and one [high value, high information] shows three prior draws. The 4th bandit 'low value' has a much lower reward value distribution (the mean is I believe set to be below the minimum of the means of the other 3 bandits) and one prior draw.

R.2.0. We thank the reviewer for their positive review and the helpful suggestions, which we have now implemented in the revised manuscript. We would like to take this reviewer's second comment to briefly comment on the status of the paper which describes and establishes the task used here. As the reviewer correctly noted, this is at the moment available as a preprint. We would like to highlight that this paper has already undergone substantial peer-review and is now undergoing minor, second revisions at a prestigious journal. We have updated our preprint to provide the latest version on BioRxiv, addressing the peer reviewer comments, which are also published there. We are not expecting the reviewer to go through these in detail. Rather, we have incorporated all changes, which were made in the first paper, in this revised draft, such as re-naming 'tabula-rasa exploration' to 'value-free random exploration', and indicate these changes throughout this response to reviewers. We can thus ensure the reviewer that the outlined analyses in this paper are in line with the ones reported in the first paper.

2.1. The authors argue that increased drawing from the high value bandit 'is an evident signature of exploitation (choosing maximal expected value)' – here as far as I can tell they are collapsing the one and three samples shown bandits (that I call 'standard' and 'high information' above). Meanwhile,

novelty-based exploration is indexed by increased selection of the novel bandit at the long horizon and ‘tabula-rasa exploration’ by increased selection of the low-value bandit.

The potential problem I see here is that reward value and information might be confounded. The ‘high value’ bandit also has the highest mean information (either 1 or 3 samples versus 1 or 0 for the other two bandits). In addition, the authors collapse across trials regardless of which 3 bandits are shown. The apparent preference towards one bandit might actually reflect a preference away from the other bandits presented and it is hard to disentangle with this collapsing procedure.

I think the task would be greatly improved by orthogonalizing reward value and information –by inclusion of novel (0 sample) low value and 3 sample low value (i.e. low value, high information) bandits. If the authors now compare low and high value trials they are not confounded by information – so preference towards the high value ones might now more reliably index exploitation versus exploration.

R.2.1. We thank the reviewer for raising this point and apologise if our analyses were not clear. For the simplicity of presentation, we have indeed collapsed the bandits with high certainty (3 prior samples) and low certainty (1 prior sample) in the high-value bandit. We did so because this differentiation is not central to this paper nor to the hypotheses we aim to address with this registered report (as detailed below). However, we present each of these factors independently. When analysing the effect of horizon onto exploration behaviour (Fig S3), we show, independently, the effect of horizon on reward/value and on information. For both, we show significant horizon effects. Based on the reviewer’s suggestion, we have now also further split the data for the high-value bandit. We analysed the frequency of selecting the high-value bandit separately depending whether it had 3 prior samples (certain-standard bandit) or 1 prior sample (standard bandit) associated to it. In line with our previous results, the high-value bandit is selected more in the short (versus long) horizon ($F(1, 60)=71.41, p<.001, \eta^2=.543$). This is the case for both bandit types, irrespective of whether it carried 3 initial samples ($t(61)=-9.092, p<.001, d=1.164, 95\%CI=[-7.23,-4.622]$) or 1 initial sample ($t(61)=-5.825, p<.001, d=.746, 95\%CI=[-4.217,-2.061]$). We also observe a bandit main effect showing that the high-value bandit was overall (i.e. across both horizons) selected more often when 3 (versus 1) initial samples were associated to it (samples main effect: $F(1, 60)=83.248, p<.001, \eta^2=.581$), demonstrating that subjects prefer certainty in their choices. Importantly, in line with our prediction, the horizon effect is stronger in the certain-standard bandit (samples-by-horizon interaction effect: $F(1, 60)=27.767, p<.001, \eta^2=.316$). This means that certainty is discarded more when exploration is beneficial – in line with the predictions of uncertainty-sensitive exploration models (such as our Thompson model).

It is thus important to note that our task is indeed well able to disentangle between information and reward. In particular, the computational models capture both of these aspects, take them into account and let them directly compete with each other. The fact that we found that the uncertainty-informed exploration models show better model fits (cf. revised version of Dubois, Habicht, et al., 2020 BiorXiv, providing additional model comparisons showing the importance of ‘complex’ models), further highlights their importance.

We appreciate the reviewer’s suggestion to introduce further bandits. However, we believe that this would not lead to any improvements, but rather weaken our task’s ability because we would have to substantially reduce the number of trials for the other bandits. For example, a low-value novel bandit would not help us capture better value-free random exploration, because per definition the value of this bandit is unknown to the participant and would therefore be treated equally as any other novel bandit.

Critically, the hypotheses tested in this manuscript are independent from the uncertainty of the bandit. This uncertainty was built in to capture the effects of complex exploration strategies (such as UCB or Thompson). However, our key hypothesis is about value-free random exploration, which is entirely insensitive to the certainty of a bandit. Our goal of associating value-free random exploration and impulsivity necessitates that we can reliably capture value-free random exploration and that all subjects are faced with the same choice options, so to capture meaningful between-subject variability. We believe that we fulfil both of these prerequisites and thus will find meaningful associations (as demonstrated in our pilot data).

Lastly, it is important to highlight that we base our hypotheses and power calculations on data collected with this version of the task, for which we show meaningful preliminary associations. Changing the task would introduce substantial uncertainty and would thus reduce the chances of finding the meaningful associations that we are hoping for.

In the revised manuscript, we explain our experimental manipulations now in more detail. We discuss how we can dissociate information from reward, and we now also mention the results when splitting the high-value bandit by certainty. Given that these additional analyses are not related to our key hypotheses, we added them to the Supplementary Information and will implement them in the Stage 2 report.

Supplementary Information: “Subjects will choose between bandits (i.e., trees) that produced samples (i.e., apples) with varying reward (i.e. size) in two different horizon conditions (Fig. 1). [...]. Each bandit i is from one of four generative groups characterised by different means μ_i and number of initial samples, following the same procedure as other studies²⁴. The size of the apple is determined by its radius (cf. Fig. S1). Manipulating the amount of information subjects have before they make their choice (i.e. initial samples) avoids a potential reward-information confound²⁴. The samples of each bandit will then be sampled from $\mathcal{N}(\mu_i, 0.8)$, truncated to $[2, 10]$, and rounded to the closest integer. The ‘certain standard bandit’ provides three initial samples and its mean μ_{cs} is sampled from a normal distribution: $\mu_{cs} \sim \mathcal{N}(5.5, 1.4)$. The ‘standard bandit’ provides one initial sample and to make sure that its mean μ_s is comparable to μ_{cs} , the trials are split equally between the four following: $\{\mu_s = \mu_{cs} + 1; \mu_s = \mu_{cs} - 1; \mu_s = \mu_{cs} + 2; \mu_s = \mu_{cs} - 2\}$. On each trial the average reward of the certain standard bandit initial samples is compared to the reward of the standard bandit initial sample. The bandit with such higher value is referred to as the ‘high-value’ bandit. The ‘novel’ bandit provides no initial samples and its mean μ_n is comparable to both μ_{cs} and μ_s by splitting the trials equally between the eight following: $\{\mu_n = \mu_{cs} + 1; \mu_n = \mu_{cs} - 1; \mu_n = \mu_{cs} + 2; \mu_n = \mu_{cs} - 2; \mu_n = \mu_s + 1; \mu_n = \mu_s - 1; \mu_n = \mu_s + 2; \mu_n = \mu_s - 2\}$. The ‘low-value’ bandit provides one initial sample which is smaller than all the other bandits’ means on that trial: $\mu_l = \min(\mu_{cs}, \mu_s, \mu_n) - 1$. We will ensure that the initial sample from the low-value bandit is the smallest by resampling from each bandit in the trials were it is not the case. [...] The horizon effect on the frequency of picking the high-value bandit was independent of whether the high-value bandit had 3 prior samples (pilot data: certain-standard bandit: $t(61)=-9.092, p<.001, d=1.164, 95\%CI=[-7.23, -4.622]$) or 1 prior sample (standard bandit: pilot data: $t(61)=-5.825, p<.001, d=.746, 95\%CI=[-4.217, -2.061]$) associated to it (samples main effect: $F(1, 60)=83.248, p<.001, \eta^2=.581$). Interestingly this horizon effect was stronger in the former (samples-by-horizon interaction effect: $F(1, 60)=27.767, p<.001, \eta^2=.316$), in line with predictions from uncertainty-guided exploration strategies.”

2) Modeling

2.2. The authors do not include sections on model comparison, parameter recovery, model identification etc in this registered report. I believe this is important to do as my understanding is that this is the stage at which reviewers are meant to evaluate the modeling. The authors do however state that these sections can be found in their prior paper (the bioRxiv preprint) – I hope it is therefore okay to comment on these sections as presented there.

R.2.2. We apologise for not having presented this data in the original submission. We originally decided to not present them as they were captured in our original paper, and because we had very specific hypotheses that were only tangentially related to these analyses. We appreciate this reviewer’s comments on these sections and are happy to address them. We have now added these sections to the revised manuscript and describe them in more detail. Please also see that we have now changed the model comparison criteria based on this reviewer’s suggestion (cf. R2.3).

2a) Model comparison:

– the authors previously compared a ‘Thompson’ model, a ‘UCB’ model and a hybrid model with or without the addition of a tabula-rasa ϵ -greedy parameter and a novelty bonus η . Within each class, the model with both added parameters performs best. Critically, there is very little difference in performance between the three classes of model at a group level (looking at the model version with both these additional parameters in each case). In a supplementary figure in the bioRxiv preprint, the authors report that 27 participants were best fit by the Thompson class model, 22 by the UCB class model and 10 by the hybrid model. Model performance in each case is evaluated using MLE and cross validation (10fold, conducted by dividing up trial into subsets.)

My concerns here are –

2.3. -Cross validation– participants are highly likely to be learning across trials and performance is hence unlikely to be independent across trials.

R.2.3. We thank the reviewer for raising this point, as we have not previously investigated learning across trials/block.

We believe that using cross-validation for our analyses is not problematic, even if there were some subtle influences between adjacent trials. Cross-validation procedures are problematic if the training and test model predictions are not independent, i.e. if for example the training data (indirectly) uses information from the test data. A classic example would be reward learning tasks, in which values at trial $t+1$ directly depend on trial t , and therefore the data cannot be split. People address this by using distinct (interleaved) sets of stimuli that are either used as training or as test data. As our models assume complete independence between trials, this is not an issue in our analyses.

What this reviewer suggests is that there is some unmodelled link between trials (as instantiated by a form of meta-learning). It is interesting to note that similar processes are likely to take place in any task that uses cross-validation for modelling, from simple psychophysics tasks (where it is known that intra-individual variability in behaviour slowly fluctuates) to complex decision-making tasks as ours. Such limitations would thus apply to the entirety of the field.

However, based on the reviewer’s suggestion we investigated whether there was any evidence of time-dependent changes in any of our measures. We first analysed performance and did not find any evidence of changes across blocks nor trials (cf. Fig. S4). When looking at the different markers for exploration, we did not observe any difference, with the exception of a small effect on novelty learning. There, we indeed found that the frequency of picking the novel bandit increased with block number (block main effect: $F(2.45, 147.22)=22.947$, $p<.001$, $\eta^2=.277$). This was counterbalanced by a decrease of picking the high-value bandit (block main effect: $F(2.64, 158.24)=9.062$, $p<.001$, $\eta^2=.131$). As we think this is an interesting observation, we have now added this analysis to the manuscript (cf. Supplementary Information). No block effect was observed on the low-value bandit frequency (block main effect: $F(2.03, 121.54)=2.219$, $p=.112$, $\eta^2=.036$), our key variable of interest. Please note that with this absence of effect as well as the correlations shown below (R.2.4.) there is no evidence that the hypotheses that we are testing in the current manuscript are affected.

Based on the reviewer’s concern and because we found this small learning effect across blocks, we have now decided to relinquish the cross-validation approach and to use the commonly used Bayesian Information Criterion (BIC) instead. We have now calculated this score for all our models and conducted model comparison using the total score, the best score per subjects and exceedance probabilities (cf. Fig S6). Importantly, all model comparisons reveal the same results as our cross-validation approach, showing that our previously winning model (i.e. Thompson sampling with the ϵ -greedy parameter and the novelty bonus η) is still performing best. We believe that these findings further strengthen our results and fully address the reviewer’s concern. We have now added this analysis to the Supplementary Information.

Methods: “Model comparison will be computed using the commonly used Bayesian Information Criterion (BIC) scores.”

Supplementary Information: “In addition to the horizon condition, when adding block as a within subject-factor in the repeated-measures ANOVA, there was an additional effect of block on the high-value bandit (pilot data: block main effect: $F(2.64, 158.24)=9.062$, $p<.001$, $\eta^2=.131$; Horizon main effect: $F(1, 60)=71.41$, $p<.001$, $\eta^2=.543$; Block-by-horizon interaction effect: $F(3, 180)=1.264$, $p=.288$, $\eta^2=.021$). This was driven by a decreased selection frequency of the high-value bandit with increasing blocks, mainly due to an decrease from block 1 to block 2 (pairwise comparison: block 1 vs block 2: $t(122)=3.261$, $p=.001$, $d=.295$, $95\%CI=[.39, 1.594]$; block 1 vs block 3: $t(122)=3.991$, $p<.001$, $d=.361$, $95\%CI=[.651, 1.931]$; block 1 vs block 4: $t(122)=5.184$, $p<.001$, $d=.469$, $95\%CI=[1.051, 2.35]$; block 2 vs block 3: $t(122)=1.218$, $p=.226$, $d=.11$, $95\%CI=[-0.187, 0.786]$; block 2 vs block 4: $t(122)=[2.511, p=.013]$, $d=.227$, $95\%CI=[.15, 1.268]$; block 3 vs block 4: $t(122)=1.474$, $p=.143$, $d=.133$, $95\%CI=[-0.14, 0.96]$). [...]. In addition to the horizon condition, when adding block as a within subject-factor in the repeated-measures ANOVA, there was no observed effect of block on the low-value bandit (pilot data: block main effect: $F(2.03, 121.54)=2.219$, $p=.112$, $\eta^2=.036$; Horizon main effect: $F(1, 60)=11.872$, $p=.001$, $\eta^2=.165$; Block-by-horizon interaction effect: $F(3, 180)=1.04$, $p=.376$, $\eta^2=.017$). [...]. In addition to the horizon condition, when adding block as a within subject-factor in the repeated-measures ANOVA, there was an additional effect of block on the novelty bandit (pilot data: block main effect: $F(2.45, 147.22)=22.947$, $p<.001$, $\eta^2=.277$; Horizon main effect: $F(1, 60)=73.72$, $p<.001$, $\eta^2=.551$; Block-by-horizon interaction effect: $F(2.68, 160.77)=.644$, $p=.571$, $\eta^2=.011$). This was driven by an increased selection frequency of the novel bandit with increasing blocks (pairwise comparison: block 1 vs block 2: $t(122)=-5.072$, $p<.001$, $d=.459$, $95\%CI=[-2.08, -0.912]$; block 1 vs block 3: $t(122)=-6.427$, $p<.001$, $d=.582$, $95\%CI=[-2.546, -1.347]$; block 1 vs block 4: $t(122)=-8.157$, $p<.001$, $d=.738$, $95\%CI=[-3.178, -1.937]$; block 2 vs block 3: $t(122)=-2.093$, $p=.038$, $d=.19$, $95\%CI=[-0.877, -0.024]$; block 2 vs block 4: $t(122)=-4.001$, $p<.001$, $d=.362$, $95\%CI=[-1.587, -0.536]$; block 3 vs block 4: $t(122)=-2.316$, $p=.022$, $d=.21$, $95\%CI=[-1.133, -0.089]$).

2.4. Modeling learning effects - relatedly, ideally the authors would include parameters in their model to examine influence of trial number and the interaction of this with the other parameters – so addressing whether participants change their preference for different forms of exploration with time on task.

R.2.4. Based on this reviewer’s suggestion and given that we observed a minor change in the novelty bandit, we extended our model comparison with models comprising block-dependent parameters. Concretely, we introduced models that had additional free parameters that captured a block-dependent change either in the novelty bonus (η_B) or in epsilon (ϵ_B). As shown below, adding ϵ_B clearly decreases the model’s performance, further supporting the above behavioural findings that there is no evidence of change in value-free random exploration across the task. Also in line with the above findings, we found a minor effect in the η_B model: This model performed similarly to our winning model when looking at the average score (paired-samples t-test: $t(61)=.084$, $p=.933$, $95\%CI=[-3.746, 4.075]$), but less well than our winning model when looking at the best score per subject (29 versus 19; cf. figure below).

Importantly, none of these additional findings are impacting our main hypothesis. In fact, these analyses show no evidence for a block-dependent effect on the value-free random exploration. To also ensure that the novelty-related effect did not impact our ϵ parameter estimates, we compared the parameter estimates across different models and found that they are almost identical with and without η_B (cf. figure below: (a) Short horizon ϵ ; (b) Long horizon ϵ ; (c) Both ϵ together).

We mention these analyses in the revised Supplementary Information.

Supplementary Information: “Given the observed minor change in the novelty bandit frequency across blocks, we extended our model comparison with a model comprising a block-dependent novelty bonus η_B . This model performed similarly to our winning model when looking at the average score (Thompson sampling+ ϵ + η : BIC: 524.91 (sd: 1.82); Thompson sampling+ ϵ + η + η_B : BIC: 524.75 (sd: 1.84); paired-samples t-test: $t(61)=.084$, $p=.933$, 95%CI=[-3.746, 4.075]), but less well than our winning model when looking at the best score per subject (Thompson sampling+ ϵ + η : Number of subjects: 29; Thompson sampling+ ϵ + η + η_B : Number of subjects: 19). Importantly adding such a parameter did not affect our main parameter of interest ϵ (correlation between ϵ from the Thompson sampling+ ϵ + η model and ϵ from the Thompson sampling+ ϵ + η + η_B model: short horizon: $r=1$, $p<.001$; long horizon: $r=.099$, $p<.001$).”

2.5. It is not possible to tell from the information presented if 27 participants are just slightly better fit by the Thompson model and 22 by the UCB or if there are two clear subsets of participants using different strategies. If the latter, then it would be suboptimal to fit these participants using a model that does not capture how they are performing the task. This is particularly an issue when relating parameter estimates (used to capture aspects of behavioural performance) to individual differences (e.g. psychopathology indices) as it could falsely give the impression that certain individuals are engaging less or more than others in a given form of exploration if the modelling of a key parameter does not capture their performance successfully.

R.2.5. We thank the reviewer for this comment which allows us to clarify our approach and hypotheses. Following the reviewer's suggestion, we have now decided to use the commonly used BIC score for model comparison (cf. R.2.3). The reason why the two complex strategies models (i.e. Thompson sampling and UCB) may performed similarly (for e.g. when using cross-validation) is because these two complex strategies make relatively similar predictions in our task. As the reviewer can see in Fig. S6, this is less pronounced when using BIC scores.

Importantly, the goal of our task and experiment was never to distinguish between these two complex strategies, but to show (and further evaluate) that there are exploration heuristics (i.e. value-free random exploration and novelty exploration) at work in addition to these complex strategies. We did so by showing that models that additionally incorporate these two heuristic parameters performed better than the complex models alone. The sole reason for including multiple complex models was to demonstrate that this effect holds true irrespective of the complex model used.

Our approach could thus be understood similarly to the one of comparing different model families, where we show that the model families with added ϵ and η outperform the other model families. We agree that this was unclear in our original manuscript, and have now rearranged the model comparison figure and detail this in the revised manuscript.

However, for our planned analysis it would be problematic if the ϵ parameters would differ greatly between the best fitting models, or if there would be a systematic bias between the complex models (as suggested by the reviewer). In our previous data (cf. Dubois, Habicht, et al., 2020), we have addressed this issue in great detail and assessed whether our observed drug effects was specific to one model or not. In that data, we did not observe any influence of drug on the best fitting model.

Moreover, we replicated the drug effects across all complex models finding the same effects in the 2nd winning model (UCB+ ϵ + η) and in the 3rd winning model (hybrid+ ϵ + η). These findings further strengthen our approach and demonstrate that model parameters are robust between complex models. Please also see our response R2.7 for additional analyses.

To further address this in the current study, we are planning to use the same approach. In our pilot data, using the BIC total score, the 1st winning model (i.e. Thompson+ ϵ + η) is significantly different than the 2nd winning (i.e. UCB+ ϵ + η). However, if this is not the case during our Stage 2 analysis, we will also link impulsivity to ϵ in this 2nd winning model. Only if we find the same results across both models, will we consider our hypothesis as confirmed. We have now extended our manuscript to clearly state these hypotheses and analysis plans.

Lastly, to further investigate model fitting coherence across subjects, we have now extended our model comparison procedures and performed a Bayesian Model Selection, which takes each participant's relative fit into account (cf. Fig 6). Using this criterion, our previously winning model (i.e. Thompson+ ϵ + η) again performed best by a large margin, further advocating towards this model. This is now reported in the revised manuscript.

Methods: “*Similar to our previous studies^{22,23}, we expect subjects to use a mixture of computationally demanding (i.e. Thompson sampling and/or UCB) and heuristic exploration strategies (i.e. value-free random exploration and novelty exploration) captured by the winning model (pilot data: BIC average score: Thompson+ η + ϵ vs Thompson model: $t(60)=-10.187$, $p<.001$, $95\%CI_M=[-72.866,-48.946]$, $d=1.304$, $95\%CI_ES=[0.967,1.657]$; Fig. S6). Model comparison will be computed using the commonly used Bayesian Information Criterion (BIC) scores. The winning model, i.e. the model with the lowest BIC score, will be used for subsequent analyses. All models which are not significantly different than the 1st winning model will additionally be used for subsequent analysis to demonstrate the generalisability of the effect (similar to previous studies²²).*”

2.6. One way to address this would be to switch to a hierarchical modelling framework and allow for there to be 2 different models adopted by different subgroups of participants – then to compare this against only allowing for 1 model.

R.2.6. We thank the reviewer for this suggestion. However, given that this is not central to our hypothesis and because such an approach was computationally not feasible, we decided not to pursue this analysis. As clearly demonstrated above, our hypotheses are robust to the different complex

models, and we only consider the hypothesis as met if we replicate the findings across different models.

The reason why we refrained from pursuing a hierarchical modelling approach (despite its appeal), was because they are computationally too demanding. As the reviewer is certainly aware of, fitting these models is a quite time-intensive process requiring substantial computational resources. We solve this by running each subject in parallel on our high-performance clusters. This can only be done because each subject's model fitting is independent. A hierarchical model fitting, however would render this approach impossible, because all subjects would need to be processed together with very limited room for parallelised processing. Given all the additional analyses we needed to conduct in the short 8 weeks we were given to revise this paper, this was clearly outside our possibilities. Moreover, the analysis of the planned full sample would render such an approach entirely unfeasible, as this would have to be conducted with almost 600 subjects. Given the above results and our clear hypotheses, we do not think such an approach would be practicable.

2.7. In addition, for novelty and tabula-rasa parameters, it would be useful to correlate the estimates of these from the Thompson versus UCB models across participants to see if they are equivalent. (Even if they are, this still leaves the issue however of how to examine information-based exploration equivalently across participants best fit by the Thompson vs UCB models).

R.2.7. We thank the reviewer for this suggested analysis. We have now assessed the correlation between ϵ values from the model with a Thompson sampling complex strategy and from the model with a UCB complex strategy, in the current study's pilot data. Similarly, we analysed the correlations between η values. Importantly, and in line with the above findings, for ϵ , we found strong correlations (correlation for short horizon ϵ : $r=.77$, $p<.001$; long horizon ϵ : $r=.86$, $p<.001$; both ϵ : $r=.81$, $p<.001$). Similarly we found strong correlations for η (short horizon η : $r=.67$, $p<.001$; long horizon η : $r=.81$, $p<.001$; both η : $r=.79$, $p<.001$). We believe that these findings further strengthen our analysis plans and we have now added all those analyses to the manuscript (cf. Fig. S10 and Fig. S11).

2.8. In the current paper, in figure S4, the authors state that the Thompson model does best, however it is hard to see why they claim this from the figure. Here again, it would be helpful to know how many individuals are best fit by each model and to also see the results once a hierarchical modelling framework is used.

R.2.8. We apologise for not providing this information before. Following the reviewer's suggestion, we have now decided to use BIC scores for model comparison (cf. R.2.3). We have now also extended the model comparison figure (now Fig. S6) with the number of subjects which are best fitted by each model. Importantly, most subjects were best fitted by our previously winning model, further strengthening our results. A Bayesian model selection further confirmed that this model is the clear winner with an exceedance probability of $p=1$.

2b) Model comparison:

2.9. No model identification analysis is provided. This addresses the issue of whether we can even tell the models apart enough for the above comparisons to be meaningful. This is done by simulating data with one model and then fitting it with all three models (UCB & n & e; Thompson & n&e; Hybrid & n&e) and seeing how often the model that was used to simulate the data is recovered (i.e. gives the best fit).

R.2.9. We thank the reviewer for this suggestion which allows us to further strengthen our results. We have now performed a model identification (cf. Fig. S7b). All models have a good recoverability with respect to exploration heuristics, but as expected there is some interplay between the complex strategies of Thompson and UCB. This is because both models make fairly similar predictions in our

task. Importantly, our hypothesis concerns the exploration heuristics, and especially value-free random exploration. The key point is that the models with and without heuristics are identifiable. We have now added these sections to the revised manuscript and describe them in more detail. Please see Figure S7 directly in the manuscript.

Supplementary Information: *“For each model, behaviour was simulated N=100 times with parameter values sampled from the pilot data mean and standard deviation. All models were fitted to this simulated data and BIC scores compared. The percentage of how often (out of the N=100 simulations) each fitted model won was computed (cf. Fig. S7b). Please note that key to our model comparison is to assess the benefit of the two exploration heuristics (novelty exploration, value-free random exploration). We expected a degree of trade-off between the different complex models, as they make relatively similar predictions.”*

2c) Parameter recovery:

2.10. Only 4 parameter values are used for this according to the bioRxiv preprint (despite the figure legend saying 10) – it would be better for values to be drawn randomly across the whole range. In a hierarchical framework, these values could be drawn from population level posterior distributions for each parameter.

R.2.10. We thank the reviewer for this suggestion and apologise for the mistake in the figure legend in the bioRxiv preprint (which had been corrected in the revised manuscript). Based on this reviewer’s suggestion, we have now performed an additional parameter recovery analysis. As suggested, for each parameter, we sampled parameter values from a normal distribution defined by the pilot data parameter means and standard deviations, and used them to simulate behaviour. This was performed N=1000 times. For each simulation, we fitted the model and analysed the correlation between the simulated parameters and the fitted parameters.

Importantly, the results from these analyses are very similar to our previous confusion matrix, which advocates towards the efficacy of our modeling pipeline (for confusion matrix cf. Fig. S7a; for a visualisation of each parameters’ correlation cf. Fig. S8) and the robustness of our analysis. The new analysis is now presented in the revised manuscript.

3) Relating impulsivity to model parameter estimates – multiple comparisons

2.11. The authors describe 3 measures (ADHD symptoms and scores on the BIS -11 and ASRS) they will relate to model parameter estimates. They also state they will perform bivariate correlations and partial correlations controlling for age and IQ. This is 6 tests – have the authors power corrected for performing multiple comparisons? Are the authors uniquely predicting a relationship with tabula-rasa exploration? Using the model based parameter? If they also plan to interrogate the relationship with other model parameters and/or non model-based indices of tabula-rasa exploration (e.g. choice of the low value bandit in the long vs short horizon) then will they multiple comparison correct for this? This also needs detailing.

R.2.11. We thank the reviewer for this question which allows us to further clarify our approach. We intend to only use two questionnaire measures, namely the BIS total score and the ASRS total score (which is an ADHD measure), and link these to our task. These two self-report measures constitute two distinct hypotheses that we will be testing separately. Given that these are clear and distinct hypotheses that we are pre-registering, we do not believe these need to be multiple comparison corrected (see our response to reviewer 1 - R1.2. for a more detailed discussion), although our sample size would be large enough to ensure 95% power even when correcting for it. Additionally, as the reviewer correctly mentions, we will use IQ and age as control variables. These are used as covariates to control for potential confounds. These are not separate hypotheses, but rather control analyses to ensure that the primary hypotheses are not confounded. To our knowledge this is the standard

approach in the field (cf. Gillan et al., 2016; Rouault et al., 2018; Seow et al., 2020), and likewise we only consider our hypotheses confirmed if both of these analyses yield the same results.

Our main analysis indeed concerns the ϵ -greedy parameter. However, we will further illustrate this effect by investigating the low-value bandit choices, which is the behavioural equivalent to the ϵ -greedy parameter. Again, we are taking a conservative approach here and only consider our hypothesis confirmed if we see a similar effect in both measures (cf. Table 1).

We consider any further analysis, such as looking at impulsivity sub-scores or transdiagnostic measures, as exploratory analyses, which will be declared as such and will only be conducted in the Stage 2 process. We have now revised our predictions and the Design Table to make this clear.

4) A broader analysis of psychopathology

2.12. In their table of planned comparisons, the authors seem to suggest they expect to find separate impulsivity and compulsivity factors. On what basis? Are appropriate measures included to facilitate this?

R.2.12. Indeed, supported by previous studies, we expect to find separate impulsivity and compulsivity factors. Previous research has found that these two dimensions only show marginal overlap sharing less than 5% of variance (Ziegler et al., 2019). This is also why previous studies that used factor analyses have found that these items load on different factors (e.g. Gillan et al., 2016; Rouault et al., 2018). We now make this clear in the revised manuscript. Please also note (which is relevant for this and the subsequent questions) that based on reviewer 3's comment (cf. R3.5), we have decided to move the factor analysis from Stage 1 (pre-registration) to Stage 2 (exploratory data analysis). This is because Stage 1 is reserved for directed hypotheses, and because we acknowledge that such a factor analysis is much more exploratory. We hope this reviewer agrees with our decision.

Methods: *"In particular, similar to previous studies^{41-43,72,73}, we expect to retrieve a depression / anxiety dimension, on which depression, social anxiety and anxiety load (SDS, LSAS and STAI-Y2 questionnaires respectively) and a compulsivity dimension, on which OCD and uncertainty intolerance traits load (OCIR and IUS questionnaires respectively). Indeed, previous research has found that impulsivity and compulsivity only shows a modest overlap⁷⁴, which is also why previous studies that used factor analyses have found that these items load on different factors^{41,43}."*

2.13. On a related note, it looks like the authors are not planning to include all the self-report measures used by both Gillan and Gershman. Why is this? It might well change the ability to identify the same latent factors. Have the authors any pilot data on the latent factors they can extract based on the subset of measures they plan to use?

R.2.13. It is indeed true that we will use a different set of questionnaires for this study. We decided to do so because we are primarily interested in impulsivity and thus use questionnaires that better capture this psychiatric domain. It is important to acknowledge that the initial study by Claire Gillan et al. (2016) as well as the subsequent studies were primarily targeted at compulsivity and compulsive-like symptoms. This is why they additionally included questionnaires capturing further compulsive aspects in disorders like alcohol use or eating disorders. Our study, on the other hand primarily focuses on impulsivity, which is why we have added more impulsivity-related questionnaires. Please see the table below for a comprehensive overview of the questionnaires used. As factor analyses indeed depend on the questionnaires entered, we do expect a somewhat different latent factor structure. Unfortunately, we are not aware of any datasets that used these exact questionnaires. However, various previous studies using factor analyses on psychiatric questionnaires (e.g. Polek et al., 2020; St Clair et al, 2017), have consistently revealed different factors for impulsivity-, compulsivity- and anxiety/depression-like factors. We thus expect similar factors. However, given that we have no clear prior data on the factor structure, we have now removed this analysis from Stage 1, labelling it clearly as an exploratory analysis (as mentioned above).

Measure	Questionnaires	Gillan et al., 2016	Rouault et al., 2018	Seow et al., 2020	Current study
ADHD	ASRS	no	no	no	yes
Uncertainty intolerance	IUS	no	no	no	yes
OCD	OCI-R	yes	yes	yes	yes
Impulsivity	BIS-(10 or 11)	yes	yes	yes	yes
Depression	SDS	yes	yes	yes	yes
Social anxiety	LSAS	yes	yes	yes	yes
Trait anxiety	STAI	yes	no	yes	yes
Apathy	AES	yes	yes	yes	no
Alcohol use	AUDIT	yes	yes	yes	no
Eating disorder	EAT-26	yes	yes	yes	no
Generalized anxiety	GAD-7	no	yes	no	no
Schizotypy	SSMS	yes	yes	yes	no
Autism	AQ-10	no	no	no	yes
Cognitive flexibility	CFS	no	no	no	yes

2.14. Do the authors plan to do a hypothesis-free comparison of all latent factor scores (3+?) to all indices of exploration/exploitation (all model parameters? Also non model based indices?) – can they set this out now so it is apparent how many tests are being conducted and whether power analyses have been corrected for this.

R.2.14. We are happy to comment here on our hypotheses. However, as detailed above, they no longer constitute a part of the Stage 1 report. Based on the previous literature, we have well-defined hypotheses about what to expect for this part. Firstly, we expect an impulsivity latent factor to replicate the effects found in the main analyses, i.e. an association with the ϵ -greedy parameter. Second, based on the prior findings of an altered stickiness in OCD patients (cf. response R1.5. for more detail), we expect that compulsivity is linked to an increased value-dependent random exploration. Lastly, we expect an anxiety/depression factor to be linked to increases in exploration (e.g. Blanco et al., 2013) although it is unclear whether it will be a specific exploration strategy. We now mention these hypotheses more concretely in the revised manuscript, but make clear that they are part of Stage 2.

Methods: *“In Stage 2, we will perform a further exploratory step. [...] In terms of exploration, studies⁴⁷ have demonstrated increases in OCD patients^{45,75} and depression⁴⁸, as well as abnormalities in anxiety⁴⁹ and other disorders related to avoidance of uncertainty⁴⁶. Additionally we will explore the link between exploration and autism, as it has overlapping symptoms with ADHD⁷⁶, and between exploration and cognitive flexibility, as it is thought to play a role in the exploration-exploitation trade-off^{77,78}.”*

2.15. Are the authors hoping to be able to claim that a given model parameter is related to one latent factor and not another? If so, given they plan to use oblique rotation (e.g. a correlated factors approach) they should do power calculations taking into account the anticipated degree of correlations between factors.

For this, it would help to have pilot data on the self-report measures, the latent factors extracted and their degree of correlation.

R.2.15. Even though we do not have pilot data on these latent factors (which themselves would require a substantially large sample size), we can draw the association between the expected factors by analogy from existing data. For example, previous research has shown that impulsivity and compulsivity barely overlap using the same questionnaires as used here (e.g. Ziegler et al., 2019; $r \sim .1$). Moreover, across multiple studies using various variants of orthogonal and non-orthogonal

factors, researchers found distinct factors for depression/anxiety, impulsivity, and compulsivity (St Clair et al., 2017; Gillan & Daw, 2016). This suggest strongly that we can expect a relatively low correlation between the latent factors (even though we are planning to use oblique rotation in line with previous studies). Importantly, these previous studies have shown that the results generally benefit from having these latent factors as competing predictors. We thus plan to do the same in Stage 2 and would expect higher effect sizes than the ones used for our power calculations for Stage 1.

Reviewer #3 (Remarks to the Author):

This is a signed review (Chris Chambers, Cardiff University).

I enjoyed reading this Stage 1 Registered Report – it tackles an important question in the area of impulsivity and cognitive control, with implications for theory and the field of computational psychiatry. The study is carefully designed and the manuscript is clearly written.

My main comments are methodological and address the mapping between the hypotheses, statistical tests, and falsifiability. All issues in my review are addressable through revision.

R3.0 We thank Prof Chambers for this very positive and encouraging feedback. We truly appreciate the helpful suggestions regarding the structuring of our registered report. This is our first preregistration, and it is great to learn from an open science expert. In line with this reviewer's suggestion, we have clarified all hypotheses and indeed moved parts of the analysis to Stage 2, as we agree they are exploratory. We believe that our changes now adequately address this reviewer's concerns.

3.1. In reviewing the design table, it was not always clear to me which outcomes would be considered to confirm or disconfirm the specific hypotheses. For example, taking the first hypothesis (that participants will select bandits with lower expected values in the long vs short horizon), the authors appear to propose five t-tests – thus the scientific hypothesis does not map directly on to a statistical hypothesis – and the “interpretation” column does not make clear which tests within this nested structure must reveal evidence of a difference (or not) for the hypothesis to be confirmed or disconfirmed. I also didn't understand the language (used here and elsewhere) “absence of horizon effect may reflect meaningful between-subject variation” – does this imply that the authors consider the hypothesis irrefutable and that any non-significant outcome instead indicates that a true effect is obscured by individual differences? If so (and I accept I may have misunderstood) then I think this is problematic from the point of view of falsifiability. To make this clear, I suggest plotting the predicted results for each of the DVs (or listing them as directional formulae, e.g. $X > Y$) and then making clear which outcomes (or combination of outcomes) must be observed to confirm or disconfirm the overall prediction. The authors may also want to list specific statistical tests as sub-hypotheses (e.g. hyp 1a, 1b) and then explain which sub-hypotheses must be confirmed for the overarching "hypothesis 1" to be confirmed.

R3.1. We thank the reviewer for this comment which helps us improve the quality and precision of our registered report. We have now completely revised our design table to make our hypotheses and their implications much more clear, in accordance with the reviewer's suggestion. Each sub-hypothesis has now been numbered, and the predicted result listed as a directional formula. We have also removed the unclear sentence in the “interpretation” column and have now listed specifically the interpretation given to different outcomes for each sub-hypothesis. For example for the first sub-hypothesis, “Hypothesis” column: 1.1.a: High-value bandit frequency: $SH > LH$, “Interpretation” column: 1.1.a: “No effect: no evidence that our horizon manipulation modulated exploration. Opposite effect: exploitation is increased in the long horizon” (cf. Design Table).

In addition, we have removed the 'individual differences' mention, which was unclear and misleading. Such an analysis would be exploratory and would clearly belong to Stage 2. We believe that the revised Design Table is now much more clearly structured and each hypothesis is now well falsifiable. Please see the revised Design Table directly in the revised manuscript.

3.2. Also on the first hypothesis (and as implied above I would suggest numbering these predictions to make more readily trackable), I didn't understand how this prediction was tested: “In terms of bandit types, they select the low-value and novel bandit more, and the high-value bandit less.” Which test

interrogates this hypothesis? Please ensure a complete and precise correspondence between every prediction and the statistical test(s) that of that prediction.

R.3.2. We apologise for the lack of clarity. This has now been changed as detailed in R.3.1. We now clearly describe which outcome measures capture each sub-hypothesis.

3.3. Further down the design table, turning to the question of whether participants deploy exploration heuristics, the authors write in the interpretation column: “All our previous studies show clear effects, thus an absence of effect is not expected. If we will not find evidence for some of the strategies, we will focus subsequent analyses on the strategies we find evidence for. Moreover, we will explore whether high impulsive subjects have a different winning model from the low impulsive subjects.” This feels a little off balance to me. I can accept that the hypothesis may have a strong prior, but the interpretation column should be free of such baggage and simply make clear what evidence would demonstrate the hypothesis to be supported or unsupported, rather than explaining why that might or might not be (which is best reserved for the Discussion). I would also remove comments about the plans to explore the data given different outcomes – this is a given (and could be reported in the Results as an exploratory post hoc analysis) but in my view doesn’t add any information to inform the precision, precommitment or falsifiability for the confirmatory plans.

R.3.3. We apologise for not being more rigorous on this. We agree that the interpretation column should have a factual interpretation rather than mentioning post-hoc Stage 2 analyses. We have now revised the interpretation section for each hypothesis and detail how the different outcomes will be interpreted. For example, we changed this row (as discussed in R.3.1): “Hypothesis” column: 1.3: Model performance: complex models + $\epsilon + \eta >$ other models, “Interpretation” column: 1.3: “No effect: no evidence that subjects combine complex models with heuristics. Opposite effect: Subjects are not using complex models with both heuristics”. Please see the revised Design Table in the main manuscript.

3.4. Most of the power analyses draw on point effect size estimates from the pilot data. In my view, power analyses should generally not be based solely on the effect size estimate of a pilot study but on the minimal plausible yet theoretically informative value or the lower bound estimate extracted from a set of previous studies (including the pilot). Basing the power analysis on a single point value is inadvisable because the estimate of the effect size from a small pilot study is consistent with a range of values, including effects much smaller than the estimate that may nonetheless be theoretically informative. That said, this issue is mitigated by the fact that the actual sample size (580) will be much higher than needed for most of the hypothesis tests. I would therefore suggest also reporting a sensitivity power analysis for the hypothesis tests that are based on the pilot data, to report what effect size the test has 95% power to detect at the actual N of 580. These effect sizes will be substantially smaller than the observed effect sizes in the pilot study and, e.g. are likely to be below the lower bound 95% CI (see point 7).

R.3.4. We thank the reviewer for this great suggestion. We based our power calculations on the guidelines provided by the Nature journals, stating that “power analysis must be based on the lowest available or meaningful estimate of the effect size”

(<https://www.nature.com/nathumbehav/registeredreports>).

For step 2, we thus extracted the effect sizes from multiple prior studies and calculated the sample size based on the smallest of these effect sizes (leading to the sample size of 580).

For step 1, effect size estimates are less readily available because this is a (relatively) novel task. The reason why we chose to gather online pilot data and use it for our analysis, is because we wanted to base our effects on online data (using the same conditions as in our planned study), which may be more noisy. However, we fully agree that using an effect size from a single study has its limitations, as pointed out by this reviewer. We have thus looked at the power calculations based on our previous data (drug study; Dubois, Habicht et al., 2020), which gave very similar results (cf. table below).

Measure	Effect size		Estimated sample size	
	Pilot data	Drug study	Pilot data	Drug study
High-value bandit	t(60)=8.45, p<.001, d=1.082	t(58)=6.487, p<.001, d=.845	14	21
Low-value bandit	t(60)=-3.446, p=.001, d=.441	t(58)=-4.818, p<.001, d=.627	69	36
Novel bandit	t(60)=-8.586, p<.001, d=1.099	t(58)=-5.498, p<.001, d=.716	13	28
Score 1st SH vs 1st LH	t(60)=6.522, p<.001, d=.835	t(58)=4.741, p<.001, d=.617	21	37
Score 1st SH vs all LH	t(60)=-16.096, p<.001, d=2.06	t(58)=-10.344, p<.001, d=1.347	6	9
ϵ -greedy parameter	t(60)=-3.23, p=.002, d=.413	t(58)=-3.091, p=.003, d=.402	79	83
Novelty bonus η	t(60)=-9.43, p<.001, d=1.207	t(58)=-6.221, p<.001, d=.810	12	22

Additionally, we have now computed 95% Confidence Intervals around the effect size (cf. Table S1) and have performed additional sensitivity power analysis in G*Power. Importantly, for each measure of the pilot data, the lower bounds of the 95% CI are above the minimal detectable effect obtained from the sensitivity analysis. Please also see revised Table S1 with details to all analyses.

Methods: “A sensitivity power analysis (performed in G*Power) predicts that with such a sample size, we will be able to detect an effect size (Minimal Detectable Effect, MDE) of 0.15 with 95%. Importantly, the lower bound of the 95% Confidence Interval of each pilot data measures’ effect size is above this MDE (cf. Table S1), ensuring a detectable effect.”

3.5. The authors will use a number of assessment instruments and then propose to covary for individual scores in step 2 or include them in a factor analysis in step 3. I would like to see a little more precision in the specification of which scores will (and won’t) be included in the factor analysis in step 3. At present, the analysis in step 3 is described relatively vaguely compared to steps 1 and 2. It may be that step 3 is best reserved for Stage 2 exploratory analyses and removed entirely from the Stage 1 manuscript, but I will defer to reviewers who are more expert on FA to offer a view.

R.3.5. We thank the reviewer for this very helpful suggestion. We agree that the factor analysis (step 3) is much more exploratory and we have only limited prior data to make concrete hypotheses. In accordance with this reviewer’s suggestion, we have thus removed step 3 from the initial Stage 1 report (and thus also removed it from the Design Table). This will now be conducted as an exploratory Stage 2 analysis. We believe this is more adequate for this specific analysis and makes our Stage 1 hypotheses and analyses more rigorous. We have decided to keep the step 3 analysis methods in the revised methods section for transparency, but removed it from the Stage 1 parts of the manuscript. We will use all questionnaires listed in the methods section (cf. Table in R.2.13 for a list) for this factor analysis. We now explain this in more detail in the revised manuscript.

Methods: *“In Stage 2, we will perform a further exploratory step. In order to investigate whether there exists a latent trans-diagnostic structure which helps explain exploration differences, we will perform a factor analysis. First, we will use the raw scores from all questionnaire items as variables to reduce their dimensionality similarly as previous studies^{39,41,42}. Factor analysis will be conducted using the factanal() function from the Psych package in R, with an oblique rotation (oblimin). The number of factors will be based on the Cattell’s criterion⁷¹, using the Cattell-Nelson-Gorsuch test (nFactors package in R). Factors will be labelled based on the items which loaded the most strongly in a consensus discussion among the authors.”*

3.6. Will excluded participants be replaced? I would suggest doing so to ensure that the final sample meets the required minimum sample size.

R.3.6. We apologise for the lack of clarity. Indeed, excluded participants will be replaced. This has now been added to the revised methods section.

Methods: *“In Stage 2, subjects that will be excluded due to the above inclusion/exclusion criteria will be replaced prior to data analysis.”*

3.7. In Table S1, could the authors also report the 95% confidence interval around the effect size estimate? I’m not sure what is currently reported in the CI ranges, but a CI around the effect size would be informative.

R.3.7. We thank the reviewer for this suggestion. We have now added 95% confidence interval around the effect size estimate to Table S1.

Reviewer #4 (Remarks to the Author):

I'm very grateful for the opportunity to read and review this pre-registered report. The research questions are clear and engaging, and cover a nice balance between replication and exploration. I applaud the authors for opting to go the route of preregistering the hypotheses and analyses. Below, I provide feedback and suggestions about how to improve the manuscript. I separate my feedback to address the experiment, the models, the analysis, and then some minor comments.

R4.0. We thank Prof Wu for the positive and encouraging feedback, and the many helpful suggestions on this preregistered report. We have now responded to all raised issues and believe we fully address the reviewer's concerns.

Experiment:

4.1. There are several aspects of the task that were not clear to me. First of all, is it the case that participants experienced 200 trials where they made one choice (short horizon) and 200 trials where they made six choices (long horizon)? This would mean that there is substantially more data for the long horizon choices, which may violate some of the statistical assumptions (i.e., your paired t-tests assume equal variance). And since there is no value to exploration in the short horizon trials, why then do you still see some novelty/exploration?

R.4.1. We apologise for not being more clear on these aspects of our task. In line with prior studies using the same horizon manipulation (e.g. Wilson et al., 2014; Sommerville et al., 2016), we only analyse the first choice (draw) of each horizon. This is not only to address the variance issue, raised by this reviewer, but also to prevent any confounding influences of sampled reward (for a detailed discussion of these potential confounding factors, please see Wilson et al., 2014). We have now detailed this in the revised manuscript.

We agree that an optimal agent would show no or little exploration in the short horizon. However, this is not what is commonly observed in human behaviour. In fact, humans tend to still show signs of exploration even in contexts in which exploration is not beneficial (e.g. Findling et al., 2019; Williams & Taylor, 2006). Exploration in the short horizon has indeed also been observed in previous studies using this horizon manipulation, both in our own and in others' work (e.g. Wilson et al., 2014; Sommerville et al., 2016; Dubois et al., 2020). This might be driven by phylogenetically old decision-making biases, as the human brain is optimized to be in environments (such as our world) where outcomes are ever changing and exploration is inherently beneficial. We have now added this information to the revised methods section and are planning to discuss this in more detail in the discussion section at Stage 2.

Methods: *"To induce changes in exploration, we manipulated the number of samples they could draw from a given set of bandits²⁴. This decision horizon varied between two conditions (intermixed trials): they could either perform one draw (short horizon condition) or six draws (long horizon condition). The long horizon promotes exploration as obtained information can subsequently be used²⁴. Although there would be no interest for an optimal agent to explore in the short horizon, humans still show signs of exploration even when it is not beneficial, but to a much lesser extent^{14,50}. In fact exploration in the short horizon has previously been observed in humans^{23,24,51}. [...] In line with previous studies investigating horizon-dependent exploration^{23,24,51}, we only investigate first draw of each horizon in the main analysis. This allows us to compare between horizon conditions preventing biases of collected reward and unequal variance. [...] To formally assess which exploration strategies are being used, we will turn to computational modeling. Similar to the behavioural analysis, only the first draw of each trial will be analysed. [...] To assess the changes in exploration strategy, we will examine the winning model's fitted parameters. Those parameters will be fitted to the first draw of all trials of each subject."*

4.2. Secondly, how are the mean rewards for each tree determined? Do they vary from trial to trial?

What is the range of reward values? Does the global maxima stay fixed across trials? These are important considerations, since participants could use meta learning to determine whether exploring the novel option pays off (e.g., if the pre-samples of the high value tree are already at the max apple size, you already know that the novel tree won't be better). In order to properly measure exploration, there needs to be uncertainty about the reward values. Some previous studies have used random scaling of reward values from round to round (e.g., Wu et al., 2020) to maintain uncertainty about which option has the highest mean reward. It would also be helpful to see the influence of trial and horizon on score, as a diagnostic plot. The communication of reward values to participants was also unclear. Are reward values indicated by the radius, circumference, or the area of the apple? Is the color of the apple reflecting its size (e.g., as a visual aid similar to Wu et al., 2018), or is it indicating which tree it comes from? The color mapping could have some unintended effects, since green is associated with unripe apples.

R.4.2. We thank the reviewer for enquiring about these points, which allows us to further detail our task design. Even though absolute range of the possible rewards was bounded (similar to previous studies, e.g. Wilson et al., 2014), on each trial we sampled a mean reward for the certain-standard bandit from a wide range of possible values to ensure uncertainty and randomly scaled the other mean rewards around it. We followed thereby the same procedure as used in related previous studies (Wilson et al., 2014). Additionally, after being sampled from their respective distributions, the order of the prior samples of the presented bandits was permuted at the beginning of each trial to avoid biases. This has now been thoroughly detailed in the Supplementary Information.

To further assess whether there was a form of meta-learning, we have now also conducted additional analyses of trial/block influences on score. For each participant we performed a regression on the average reward per block as well as a regression on the reward per trial. The obtained slopes were then compared to a null slope. We did not find any evidence that performance changed across time. Indeed, no significant slope was found for the short horizon reward (reward per block: paired-samples t-test: $t(60)=1.036$, $p=.304$, $d=.133$, 95% CI=[-.0106,.0333]; reward per trial: $t(60)=.396$, $p=.694$, $d=.051$, 95% CI=[-.0001,.0002]), for the long horizons' first sample reward (reward per block ($t(60)=-.286$, $p=.776$, $d=-0.037$, 95% CI=[-.0249,.0187]; reward per trial: $t(60)=1.023$, $p=.31$, $d=.131$, 95% CI=[-.0001,.0002]), nor for the long horizons' average reward (reward per block ($t(60)=1.139$, $p=.259$, $d=.146$, 95% CI=[-.0066,.024]; reward per trial: $t(60)=-.502$, $p=.618$, $d=-.064$, 95% CI=[-.0002,.0001]). This suggest that there is no evidence that meta-learning would have helped to improve performance throughout the task. This has now been added to the Supplementary Information (Fig. S4).

Interestingly, although there was no influence of block on reward, there was a mild increase in novelty bandit frequency with increasing block numbers (cf. our response R.2.3), which was counterbalanced by a mild decrease in high-value bandit frequency. However, these effects did not prevail in the model comparison and we report these analyses now in the Supplementary Information. Please note (and see a detailed discussion in R.2.3) that these effects did not impact on our variables of interest (especially the ϵ parameter).

The reward is given by linearly increasing the radius of the apple. For our previous study (cf. Dubois, Habicht, et al. 2020), we tested various versions with different apple size ranges and based on these pilots we opted for a range of 9 different apple sizes, as our pilots showed that they were easily distinguishable (even in children as young as 8 years of age; Dubois, Bowler, et al., 2020). We plot these below for the reviewer's visual inspection. Moreover, the apples are presented (and remained) next to each other on the screen in the 'crate', so that apple sizes are directly comparable. We have now added this information and the below figure (cf. Fig. S1) to the revised manuscript.

The colour of the apple indicates which tree it was sampled from (cf. Fig. 1), and to make unwanted associations with the colour code unlikely, colours are randomly shuffled on each trial (8 sets of 3

different colours) in the task. Additionally, to assess subjects' understanding of apple sizes, we conduct extensive training prior to the main experiment, in which subjects have to categorise different apple sizes. This has now been added to the methods.

Supplementary Information: “ [...] Each bandit i is from one of four generative groups characterised by different means μ_i and number of initial samples, following the same procedure as other studies²⁴. The size of the apple is determined by its radius (cf. Fig. S1). Manipulating the amount of information subjects have before they make their choice (i.e. initial samples) avoids a potential reward-information confound²⁴. The samples of each bandit will then be sampled from $\mathcal{N}(\mu_i, 0.8)$, truncated to $[2, 10]$, and rounded to the closest integer. The ‘certain standard bandit’ provides three initial samples and its mean μ_{CS} is sampled from a normal distribution: $\mu_{CS} \sim \mathcal{N}(5.5, 1.4)$. The ‘standard bandit’ provides one initial sample and to make sure that its mean μ_S is comparable to μ_{CS} , the trials are split equally between the four following: $\{\mu_S = \mu_{CS} + 1; \mu_S = \mu_{CS} - 1; \mu_S = \mu_{CS} + 2; \mu_S = \mu_{CS} - 2\}$. On each trial the average reward of the certain standard bandit initial samples is compared to the reward of the standard bandit initial sample. The bandit with such higher value is referred to as the ‘high-value’ bandit. The ‘novel’ bandit provides no initial samples and its mean μ_N is comparable to both μ_{CS} and μ_S by splitting the trials equally between the eight following: $\{\mu_N = \mu_{CS} + 1; \mu_N = \mu_{CS} - 1; \mu_N = \mu_{CS} + 2; \mu_N = \mu_{CS} - 2; \mu_N = \mu_S + 1; \mu_N = \mu_S - 1; \mu_N = \mu_S + 2; \mu_N = \mu_S - 2\}$. The ‘low-value’ bandit provides one initial sample which is smaller than all the other bandits’ means on that trial: $\mu_L = \min(\mu_{CS}, \mu_S, \mu_N) - 1$. We will ensure that the initial sample from the low-value bandit is the smallest by resampling from each bandit in the trials were it is not the case. Even though the absolute range of reward is set, randomly scaling each reward mean around the certain standard bandits’ reward mean allows to maintain uncertainty about the overall average reward on each trial similarly to previous studies^{24,27}. At the beginning of each trial, the prior samples of the presented bandits are sampled from their respective distributions. Their order is then permuted to avoid biases.”

4.3. Lastly, how long do you expect the experiment to take? Based on (200 trials * 1 choice in short horizon cond) + (200 trials * 6 choices in long horizon cond) = 1400 choices given by participants, averaging at 3000 ms per choice (Fig. S1b), that would be 70 minutes just for response times in the main task, without any instructions. It's also not clear how long the questionnaires will take, but I worry about decision fatigue and drop out. This is likely to disproportionately affect ADHD participants, who by definition have difficulties with sustained attention. This may lead to a self-selection bias and a majority healthy control population in your final sample. And although I feel there are already too many questionnaires, I wonder if you would want to replace one with an autism scale, since that is the closest to ADHD, and provide valuable differentiation. Also, you may want to ask participants if they are taking medication, because that may mask ADHD-like behavior.

R.4.3. We apologise for not providing this information beforehand. We have now added a figure (Fig. S2a) with the total duration of the task in the pilot data, which took on average 45 min. The reason why the task takes much less time than estimated by this reviewer is because the reaction time in the first trial (former Fig. S1b and current Fig. S2c) is a much slower than in the subsequent trials in the long horizon (cf. figure below). This is because the reaction time for the first draw includes studying and comparing all prior samples, whereas for all subsequent choices, only one new sample is being added.

Long horizon reaction times

We indeed agree that it is important to be mindful of the time such experiments take online. We have thus designed the task to minimise fatigue effects. We split the task in 4 short blocks of approximately 10 minutes. Subjects are free to take breaks between blocks which can be long as they want, so to help them to re-focus. Moreover, we deliberately collect task data before questionnaire data, to ensure that they play the task when they are still ‘fresh’ and alert. Additionally, in the pilot data, we observe no evidence for any fatigue effects (i.e. block effects) on the low-value bandit, our key outcome measure (pilot data: block main effect: $F(2.03, 121.54)=2.219$, $p=.112$, $\eta^2=.036$; Horizon main effect: $F(1, 60)=11.872$, $p=.001$, $\eta^2=.165$; Block-by-horizon interaction effect: $F(3, 180)=1.04$, $p=.376$, $\eta^2=.017$). This analysis has now also been added to the Supplementary Information.

We thank the reviewer for raising this interesting point regarding autism. We would like to highlight that ADHD can indeed be seen as a general risk factor for psychopathology and many different psychiatric disorders, not just for autism. However, based on the reviewer’s suggestion, we have decided to add a short, additional autism scale (the Autism spectrum Quotient; AQ-10; Allison et al., 2012), which we will investigate the association in an exploratory analysis in Stage 2. As it is only 10 items, it will not make the experiment noticeably longer.

Regarding medication, we have previously not found any evidence that medication would substantially impact the found associations (Dubois, Bowler, et al, 2020). Likewise, all prior online studies (to our knowledge) have not controlled for medication consumption. However, because we agree that it would be interesting to assess any medication effects, we have decided to collect medication status in the full sample.

Methods: “In addition, we will collect further questionnaires to investigate additional psychiatric dimensions (cf. Analysis Plan, step 3). These entail the Liebowitz Social Anxiety Scale⁵⁴ (LSAS), the State-Trait Anxiety Inventory⁵⁵ (STAI-Y2), Intolerance of Uncertainty Scale⁵⁶ (IUS), Obsessive-Compulsive Inventory-Revised⁵⁷ (OCI-R), and Zung’s Self-rating Depression Scale⁵⁸ (SDS), in accordance with similar previous approaches^{41–43}, as well as the Cognitive Flexibility Scale⁵⁹ (CFS) and the Autism spectrum Quotient⁶⁰ (AQ-10). [...]. Additionally we will explore the link between exploration and autism, as it has overlapping symptoms with ADHD⁷⁶, and between exploration and cognitive flexibility, as it is thought to play a role in the exploration-exploitation trade-off^{77,78}.”

Supplementary Information: “In addition to the horizon condition, when adding block as a within subject-factor in the repeated-measures ANOVA, there was no observed effect of block on the low-value bandit (pilot data: block main effect: $F(2.03, 121.54)=2.219$, $p=.112$, $\eta^2=.036$; Horizon main effect: $F(1, 60)=11.872$, $p=.001$, $\eta^2=.165$; Block-by-horizon interaction effect: $F(3, 180)=1.04$, $p=.376$, $\eta^2=.017$).”

Models:

4.4. As the authors themselves admit, what was supplied in the supplementary information is not sufficient for understanding/reproducing the models. I hope that will be addressed in a later version. The simple inclusion of the description that their models are time-invariant Kalman filters would have been helpful, instead of directing the reader to a different preprint by the authors, which largely contains the same textual description of the models. The concept "tabula rasa" exploration should also be explained in the introduction of the current report, since it involves important theoretical commitments.

R.4.4. We apologise for not detailing this information beforehand. We have now added a detailed explanation of the terms (please note that 'tabula-rasa' is now termed 'value-free' random exploration) in the introduction, as well as a detailed description of the models in the Supplementary Information. We now also explicitly mention the use of Kalman filters, which we forgot to highlight in the original study. Please see the revised manuscript for the model descriptions.

Introduction: *"Such 'value-free' random exploration ignores all available information (i.e., expectation and uncertainty or choices) and thus forgoes any costly computation, in contrast to more refined 'value-based' random exploration which adds stochasticity during choice value computation or directed exploration which biases choice towards information gain^{24,25}."*

Supplementary Information: *"At each time point t , in which a sample m , of one of the bandits is presented, the expected mean Q and precision $\tau = \frac{1}{\sigma^2}$ of each bandit are updated as follows: [...]. Those update rules are equivalent to using a Kalman filter⁷² in stationary bandits."*

4.5. In the "Subjects explore using heuristics" section, a t-test is reported, but it's not clear what is being compared. Is this the winning model compared against the next closest model? Fig. S4 suggests that the model results are rather inconclusive, and doesn't seem to match the test statistics. Also, if you are comparing likelihoods (bounded between [0,1]), the normality assumptions of the t-test may not be met. You would do better to use a t-test against the negative log likelihoods, or better yet, to use a Bayesian model selection framework and compute the protected exceedance probability (Rigoux et al., 2014).

R.4.5. We apologise for the lack of clarity regarding this analysis. Importantly, following the suggestion of reviewer 2, we have now decided to use BIC for model comparison (cf. R.2.3). In the "Subjects explore using heuristics", we now report the t-test comparing the BIC score of the winning model (the Thompson+ ϵ + η model) and the same complex model with no heuristics (the Thompson model here). Please note that BIC is more likely to comply with normality assumptions, comparable to the negative log likelihoods suggested but this reviewer. We now also report normality tests (Shapiro normality assumption) and Wilcoxon signed rank tests with their effect sizes for completion (cf. Table S1). As this reviewer can see, we obtain identical effects when using tests that do not rely on any normality assumptions. We have now decided to use non-parametric tests whenever the normality assumptions will be violated, as detailed in response R.4.7.

Additionally, we thank the reviewer for the suggested Bayesian model selection framework, which we have now added to the model comparison and report the exceedance probabilities of our models (cf. Fig. S6). The results of the model comparison are now very conclusive with the Thompson+ ϵ + η winning irrespective of whether comparing the average BIC, the number of subjects best fitted by each model or the exceedance probabilities (cf. Fig. S6).

4.6. Along these lines, I was a bit surprised to learn that your UCB model has as many as 4 different forms of exploration. The softmax tau, the information bonus gamma, the tabula rasa exploration epsilon, and the novelty bonus eta. From eyeballing Fig. S4, this model seems to perform quite similarly as your winning model. While the introduction is quite committed to interpreting the model

results in terms of novelty vs. tabula rasa exploration, I'm not sure the pilot results justify the exclusion of these other two forms of exploration (information bonus and softmax temperature). I think the ideal method would be to run model recovery and parameter recovery on your pilot data (see Wilson & Collins, 2019) to find out if your experiment can in fact distinguish between these models and whether these parameters are in fact capturing distinct and interpretable phenomenon. This would substantially improve the interpretability of your model results.

R.4.6. We thank the reviewer for these suggested analyses and the question regarding the model comparison, which allows us to further detail our approach.

Firstly, we would like to highlight that our key hypothesis is that the value-free random exploration parameter is associated with impulsivity. This is because of the previous work (including our own preliminary findings) that support this specific hypothesis. Importantly, our model comparison clearly demonstrates that the presence of the ϵ parameter improves model fits, independent of whether the complex model is UCB or Thompson. We have now clarified this in the revised manuscript.

This brings us to better explain our model comparison strategy. Please see our responses R.2.5 for a more detailed discussion of the topic. Importantly, the key purpose of our model comparison was to demonstrate the presence of the two exploration heuristics (i.e. via the novelty bonus η and the ϵ -greedy parameter). As previous studies have suggested the presence of several different complex exploration strategies (i.e. Thompson and UCB), we added both of these strategies, to show that the exploration heuristics are present, irrespective of the complex strategy. One could see this as a comparison of different model families. Again, our model comparison clearly favoured the presence of novelty and value-free random exploration heuristics.

Nevertheless, because we believe our results should be robust across different complex models, we evaluated our results even further and investigated whether (i) our previously found drug effects on ϵ generalises across different complex strategies (i.e. whether they remained significant in the UCB model), and (ii) how strongly the ϵ parameter correlated between different complex models. Indeed, both these analyses clearly support our findings, showing that epsilon in both models are very similar and that the previous effects were replicated across models. Please see our responses R.2.5 and R.2.7 for a more detailed discussion and our explanation how we address this in the current study.

Based on another reviewer's suggestion, we have now also adjusted our model selection procedure (cf. R.2.3), which confirms the winning model being the same as previously, and in addition showing that this model now wins clearly in terms of model performance, head counts, and exceedance probability (cf. R.2.8).

Following the reviewer's suggestion, we have now also performed an additional parameter recovery analysis for our winning model. The results clearly show that the parameters are well recoverable and thus our pipeline interpretable (for confusion matrix cf. Fig. S7; for a visualisation of each parameters' correlation cf. Fig. S8). Please also see our response R.2.10 for further discussion.

Lastly, we have now also performed a model identification analysis. As expected, the different model variants have a good recoverability. As detailed above, we expected some trade-offs between the complex strategy models (i.e. Thompson vs UCB), because they make fairly similar predictions in our tasks. Indeed, we find some interplay between Thompson or UCB, but not between models with or without the exploration heuristics. Please see our response R.2.9 for a more detailed discussion of the model recovery analysis and our (adjusted) analysis plans.

All these analyses have now been added to revised manuscript. Please see our responses R.2.3-R.2.9 for the corresponding manuscript excerpts.

Analysis:

4.7. The analysis plan seems detailed, and both the hypotheses and interpretations of alternative outcomes seem well thought-out. I would suggest however, that t-tests might not be the ideal mode of analysis for all of these research questions. While the 3 separate steps of analysis make a lot of sense, there are potential interactions between variables that would be missed. I believe a mixed effects regression framework would be superior in many ways. For instance, you could see if your ADHD/impulsivity variables interact with the exploration and horizon length variables from step 1. A mixed effects approach would also be beneficial in step 3 because you could treat participants as

random intercepts to account for individual biases in the survey results. Additionally, when comparing parameter estimates, a Wilcoxon signed-rank test would provide a rank-based alternative to a paired t-test, which wouldn't require assumptions of normality that might be violated by your bounded parameter estimation procedure. Lastly, I was wondering how you intend to avoid overfitting or surprising correlations in your factor analysis? With a sample size of 580 and a large number of survey questions, some type of statistical correction seems crucial.

R.4.7. We thank the reviewer for raising these analysis-related points. For our analyses, we decided to use the simplest analyses possible, which allow clear and straightforward interpretations. For example, in the step 1 horizon effects, we expect horizon effects without taking any individual differences into account. This is why we opted for simple comparisons of means (i.e. t-tests). We believe that for this step, such an approach is most appropriate. However, we agree that t-tests are only valid when the Normality assumptions are fulfilled. We thus now report both t-tests and Wilcoxon signed-rank tests as well as the normality tests of our pilot data (cf. Table S1). As the reviewer can see, those yield identical results. For the analysis of the entire sample, we will again use the Shapiro test to assess normality and then use the adequate test.

For step 2 (individual differences), we again use the most simple approach we could think of, which is (partial) correlations. To our understanding, mixed-effects analyses are not applicable here as there is only one data point (e.g. model parameter estimate) per subject. However, we can indeed additionally use more refined regression models, such as repeated-measures ANOVAs with within factor horizon and a between subjects variable [impulsivity/ADHD-symptoms] to assess these effects further. We have now added this to the Design Table.

Lastly, we would like to highlight that based on reviewer 3's comment (cf. R.3.5), we have decided to move the factor analysis from Stage 1 (pre-registration) to Stage 2 (exploratory data analysis). Importantly, factor analysis is solely done on the questionnaire data, whereas our research question is targeted at the link to the exploration variables. To avoid overfitting, we will correct for the number of comparisons using a Bonferroni correction. By doing so, we follow the same protocol as previous studies, which have successfully overcome these potential limitations (e.g. Gillan et al., 2016; Seow & Gillan, 2020; Rouault et al., 2018). We have now detailed all these analyses in the revised manuscript.

Methods: "A summary of the statistics performed on all measures on the pilot data can be found in the Supplementary Information, Table S1. Importantly, Wilcoxon signed-rank tests will be used instead of paired t-tests if the Shapiro normality assumption is violated. [...] Additionally, we will also perform repeated-measures ANOVAs with within factor horizon and a between subjects variable [impulsivity/ADHD-symptoms] to assess these effects further. [...] First, we expect that our two impulsivity questionnaires (BIS and ASRS) to primarily load onto 1 factor, which will be at least as much associated with value-free random exploration as the impulsivity/ADHD questionnaires alone (cf. above). In addition to the hypothesized increase in value-free random exploration, we will investigate using multiple comparison whether impulsivity correlates with other forms of exploration (e.g. complex strategies). As a second step, we will investigate whether other factors correlate with exploration. We will control for multiple comparisons using Bonferroni correction."

Minor comments:

4.8. There are parts where the writing is fragmented or grammatically confusing. I know this isn't the final paper, but I believe this journal has rather high standards, and any future submission would benefit from more proofreading and editing

R.4.8. We apologise for allowing such slips and have now thoroughly proofread the manuscript.

4.9. Heuristic strategies require <relatively> less computation. I would avoid using the absolute term "computationally light"

R.4.9. We thank the reviewer for pointing this out. We indeed wanted to express that these strategies are less demanding relative to the complex strategies. This has now been changed and clarified throughout the manuscript.

Introduction: *“In particular, one can distinguish between sophisticated and complex exploration strategies, such as upper confidence bound²³ (UCB) which take the expectation as well as the uncertainty of all possible choice options into account, and heuristic strategies which require relatively less computation.”*

4.10. What is an "ordinary t-test"? Do you mean unpaired?

R.4.10. We thank the reviewer for noticing this imprecise wording. We meant paired t-tests and this has now been changed.

Methods: *“Concretely, for t-tests, n simulated subjects were sampled from each group normal distribution: $N(m1, std1)$ and $N(m2, std2)$ and significance was assessed using paired t-test on those 2 data sets.”*

4.11. Please report 95% CI in future plots, since your large sample size will make SEM meaningless

R.4.11. We thank the reviewer and agree with this suggestion. We now report 95% Confidence Interval for all statistics in the manuscript and will display these instead of SEM in the Figures of Stage 2.

4.12. Why the 1500ms RT exclusion criteria? Maybe I missed something, but the rationale wasn't clear. Also, the average 3000 ms RT seems really long, but perhaps I'm missing some detail about how it is measured.

R.4.12. The 1500ms RT exclusion criterion is just an arbitrary value we chose based on our pilot data (cf. Fig. S2) and our previous study (cf. Dubois et al., 2020). We chose this value because subjects rarely make a choice faster than that. Indeed, this criterion would have not excluded anyone in our lab-based studies and excluded only one subject in the pilot data which would have already been excluded for low performance. We are, however, happy to remove this exclusion criteria or change it if there is an alternative suggestion.

It is important to note that an RT of 3000ms is relative to trial onset. This is when the subjects are shown all prior samples. Therefore, they need to first investigate these prior samples and then decide which tree to choose. We thus do not think that 3000ms for the first choice was exceptionally long. Importantly, as detailed above, the subsequent choices in the long horizon are substantially shorter (cf. Figure in R.4.3.). We now explain this in more detail in the revised manuscript and in Figure S2.

Methods: *“To ensure data quality, subjects will be excluded according to the following criteria: data is incomplete, the mean score (i.e., apple size) is lower than 5.5 indicating subjects were performing at chance level⁴¹ (cf. Fig. S2b), the first draw mean reaction time was lower than 1500ms (based on our pilot data and previous study²³) indicating subjects were not according much thought to their choice (cf. Fig. S2c) [...]”*

4.13. The first sentence in the Fig. S5 caption suggests that the parameters shown here were fit in a different method from the K-fold CV method described elsewhere? If true, this seems quite strange

R.4.13. We apologise for not being clear about our method. Model fitting in cross-validation is done only on a part of the data at a time, while for parameter estimation, model fitting is done on the whole

data. Indeed, cross-validation is a way to test the models' ability to predict new data and therefore avoid overfitting. To do this, the data is split in K folds (i.e. partitions). The model is then fitted to K-1 folds and tested on the remaining fold. This will be done K times (so that each single fold is once a test fold). The model which has the highest average likelihood of left-over data is the best fitting model. Once the best model has been designated, the model needs to be fitted to the whole data in order to find the best parameter estimation, without ignoring any data.

However, this does no longer apply here as we have now changed the model comparison criteria based on this reviewer's suggestion (cf. R2.3). It is thus now fitted in the same step, rendering this point no longer relevant.

I sign my review
- Charley Wu

References:

Rigoux, L., Stephan, K. E., Friston, K. J., & Daunizeau, J. (2014). Bayesian model selection for group studies—revisited. *Neuroimage*, 84, 971-985.

Wilson, R. C., & Collins, A. G. (2019). Ten simple rules for the computational modeling of behavioral data. *Elife*, 8, e49547.

Wu, C. M., Schulz, E., Garvert, M. M., Meder, B., & Schuck, N. W. (2020). Similarities and differences in spatial and non-spatial cognitive maps. *PLOS Computational Biology*, 16, 1–28. <https://doi.org/10.1371/journal.pcbi.1008149>

Wu, C. M., Schulz, E., Speekenbrink, M., Nelson, J. D., & Meder, B. (2018). Generalization guides human exploration in vast decision spaces. *Nature human behaviour*, 2(12), 915-924.

Reviewers' Comments:

Reviewer #1 (Remarks to the Author):

Thanks for your well reasoned responses to my comments

Reviewer #2 (Remarks to the Author):

The authors have done a good job of addressing many of my concerns. I thank them for this. I have two major concerns that are still outstanding.

1) The first relates to the task design and the issue of balancing outcome value and information (samples). I see that reviewer 4 also had concerns about this. I do not feel that the authors thoroughly addressed either my or reviewer 4's concerns here. As one specific: did the authors provide a figure showing the influence of trial and horizon on score, as a diagnostic plot as requested by reviewer 4? I apologise if so, I did not find it. I concur with reviewer 4's sentiment that: "These are important considerations, since participants could use meta learning to determine whether exploring the novel option pays off (e.g., if the pre-samples of the high value tree are already at the max apple size, you already know that the novel tree won't be better). In order to properly measure exploration, there needs to be uncertainty about the reward values. Some previous studies have used random scaling of reward values from round to round (e.g., Wu et al., 2020) to maintain uncertainty about which option has the highest mean reward."

The authors' inclusion of block is helpful but does not fully address the issues raised. I still believe that we need to know if better balancing bandit value across bandit types changes behavior on this task. The authors declined to address this as it would involve task redesign and repiloting and re-establishing effects. I do appreciate that but it seems that the advantages of pre-registration come with the expectation that it is not 'too late' to ask for method issues to be addressed. Given this is an online task, I don't think it should take too much for the authors to pilot a version with alternate bandit values or potentially just a change to consistency in bandit colors (see below for more details), give both to the same participants in separate sessions, and see if estimates of parameters (ϵ, η) pertaining to value-free and novelty based exploration remain constant across both or not. If there is no meta-learning and no issue with balancing of bandit value on the parameters of interest then there should be good 'alternate version' reliability of parameter estimates across task versions.

A related issue here concerns clarity of task design. The additional information about task design provided by the authors is invaluable – this needs to be in the main manuscript. The task design is complicated and I still have questions about it – please can you clarify the below and also provide details addressing this in the manuscript, ideally with detailed information regarding the value of initial samples, the first draw and later draws (for the long horizon) as well as bandit means and distributions for each of the four bandit types. Further specific requests are given below.

To go into the weeds: in the current version of the task my understanding is that the bandits are as follows:

There are 4 types and on any given trial, 3 of the 4 bandit types are presented, each of these 3 bandits is a different color. There are 8 sets of colors.

Qu 1: Is each type of bandit shown in every possible combination equally often? Can any bandit type be any color? Is there any consistency?

Bandit types

A) Certain standard 3 samples, value : $\mu_{cs} \sim N(5.5, 1.4)$.

B) Standard. 1 sample, value : μ_s . mean μ_s is comparable to μ_{cs} , the trials are split equally between the four following: $\{\mu_s = \mu_{cs} + 1; \mu_s = \mu_{cs} - 1; \mu_s = \mu_{cs} + 2; \mu_s = \mu_{cs} - 2\}$.

'On each trial the average reward of the certain standard bandit initial samples is compared to the reward of the standard bandit initial sample. The bandit with such higher value is referred to as the 'high-value' bandit'

C) Low value 1 sample, low value: $\mu_l = \min(\mu_{cs}, \mu_s, \mu_n) - 1$.

D) Novel (no samples), value : $\mu_n = \mu_s$

Qu 2: Can I check, is μ here the mean for the urn or the mean of the initial samples? As mentioned above we need information on both the bandit mean and range of values and on the mean and range of initial sample values (and also details of first draw and later draws for the long horizon),

Qu 3: Is it constrained such that there are equal number of trials where the standard is the high value bandit and where the certain standard is the high value bandit? Is it also the case that there are trials where only one of these two bandit types is presented? Does this strategy mean that when only either the standard or certain standard is presented it can end up being the 'high-value' bandit with a lower value than it would on trials where both these bandit types are present? Please can the authors break down the values of the high value bandit according to whether it is the standard or certain standard and as a function of the combination of bandits presented. Ideally a table showing bandit type (standard, certain standard, novel, low value) and value and range of both initial samples (where there are any), 1st draw and later draws (for long horizon) as a function of whether the S and CS are co-presented or not would be useful.

Qu 4: The authors state that 'We will ensure that the initial sample from the low-value bandit is the smallest by resampling from each bandit in the trials were it is not the case. [...]'. It would be good to also understand how this procedure influences sample values across trials. Can the authors also please give the values for each bandit initial samples as a function of whether the low value bandit is one of the bandits included in a given set of 3.

Putting to one-side the potential influence of the sampling strategies highlighted in Qus 3 and 4 above, I am concerned that there are imbalances in the design (as raised in my prior review) that could impact participants' heuristics in a manner that might be difficult to ascertain. The authors state that bandit colors are varied across trials: "The trees' positions (left, middle or right) as well as their colour (8 sets of 3 different colours) were shuffled between trials." This will mean the salient initial information participants have is the number of samples. Unless I'm missing something (apologies if so) this translates as follows:

$\mu_{3\text{samples}} = \mu_{0\text{sample}} = \mu_{cs}$

$\mu_{1\text{sample}} = .5(\mu_s + \mu_l) = .5(\mu_{cs} + (\min(\mu_{cs}, \mu_s, \mu_n) - 1))$. In practice, this value is going to be lower still due to the constraints put on the initial sample of μ_l .

Hence, a value-based heuristic would be to pick the 3 sample bandit when initial values are high, otherwise to pick the 0 sample bandit and to de-prioritize the 1 sample bandits. It is unclear how this would impact the parameter estimates (maybe ϵ_B will be underestimated and η_B overestimated?). It does however seem troubling, which is why I would like, at a minimum, to be reassured there is no difference in parameter estimates for the key parameters between this version of the task and one where there is no mean difference in value as a function of samples. As I suggested previously this could be achieved by also having 'low value' 0 sample and 3 sample bandits. An alternative which doesn't involve increased numbers of trials would be to just use 4 color trees and to keep the color of each bandit constant across trials so the standard bandit and low value bandits are clearly distinguishable (I believe the potential for meta-learning that this raises would be offset by reduction in the likelihood that participants simply lump standard and low value bandits together). If the authors chose this latter option, I would still suggest piloting it to ensure the effects established to date do not change.

1b. Following on the issues outlined above, participants might well treat the standard and certain standard bandits very differently. The choice to collapse these in the results section is, in my opinion, hence suboptimal. Could the authors not simply provide results for the 4 bandit types? Certain standard, standard, novel, low value?

2. Multiple comparisons.

The authors have done a commendable job in clarifying their hypotheses and proposed tests in different stages of the study. I would ask that they provide a little further detail still.

Specifically, I agree that there is no need for multiple comparison if the authors are only deeming a hypothesis proven if two separate measures (e.g. impulsivity and ADHD symptom, scores) both show a relationship to a given parameter of interest. Please can they clearly specify those tests where this is the case. Where this is not the case and multiple comparisons are required, please can the authors provide sample size / power calculations taking this into account.

For the Stage 2 exploratory analyses (e.g. using autism, cognitive flexibility measures, latent compulsivity and anxiety/depression factor scores etc.) it is still important that the tests to be explored (i.e. which measures related to which parameters) are listed at this point so the correct multiple comparisons can be determined. An alternate would be to take any interesting findings and replicate in a separate sample. Please can the authors clarify which of these approaches they plan to take and provide full test and multiple comparison details if the former.

Reviewer #3 (Remarks to the Author):

The authors have responded thoroughly to my comments and I am happy to recommend Stage 1 IPA. Best of luck with the study!

Reviewer #4 (Remarks to the Author):

Thank you for the great amount of effort that has been put into addressing the reviews. I only now see that I'm one of four reviewers, who all seemed quite interested in the study and generated a large volume of feedback. I hope that the sheer quantity of comments from myself and the other reviewers does not dissuade you from pre-registrations in the future. Even for a journal of this status, it's clear that the amount of work that has gone into the revision is quite substantial. I applaud the authors for the thoroughness of responses and believe that the manuscript in its current form meets the requirements for provisional acceptance.

My comments have largely been addressed, although I would also like to provide a couple of comments about some results that were unclear. I'm not sure if this is the correct moment to raise them, but perhaps addressing them now will prevent additional work at a later stage. Thus, these comments should not be taken as requirements before provisional acceptance, but as suggestions for how to improve the interpretability of the results when analyzing the full data set.

1. I still have confusion about the model comparison in Fig. S6. It's a bit unclear what the UCB and hybrid models refer to. Do UCB and the hybrid always use gamma (uncertainty bonus) and beta (softmax inverse temperature)? Specifically, I'm wondering if the latter variants of UCB and the hybrid (with epsilon and eta) still include gamma and beta? Specifically, inverse temperature (beta) and value-free exploration (epsilon) are potentially interchangeable. I wonder if the obtained results are dependent on the fact that the UCB comparison always includes inverse temperature in addition to epsilon. If the key hypothesis "is that the value-free random exploration parameter is associated with impulsivity", then perhaps it would also be worthwhile to include a comparison to UCB without beta (i.e., beta fixed to 1) and then also add epsilon or epsilon+eta variants. Otherwise, your UCB results could be interpreted as being unnecessarily handicapped. As I mentioned in my previous comments (sec 4.6 of the rebuttal letter), you are instantiating as many as 4 different exploration strategies, and the current model comparison isn't quite sufficient to rule out all alternative hypothesis in competition with your key hypothesis.

2. The model recovery simulations (Fig. s7b) suggest there are some potential mis-identification issues, where data generated by the UCB+epsilon+eta model is also quite likely to be recovered by the winning thompson+epsilon+eta model, but not vice versa. It's difficult to read the exact percentage of this model mimicry from the figure, due to the existing color legend (please consider a more contrastive palette in the future; see <https://www.nature.com/articles/s41467-020-19160-7>), but it looks non-negligible. If you were to calculate the inversion matrix $p(\text{sim}|\text{recovered})$ (c.f., Wilson &

Collins, 2019 box 5 Fig. 1), it seems that your results in favor of thompson+epsilon+eta could also reflect a decent probability of the underlying data actually resulting from the competing UCB+epsilon+eta model. Intuitively, this means that participants could be behaving like the UCB+epsilon+eta model, but your task and model-comparison procedure attributes it to the Thompson+epsilon+eta model. This could perhaps be something worth mentioning in the general discussion of the final paper as a potential limitation, if similar results are also obtained in the full sample.

3. I also agree with R2 in 2.10 of the rebuttal letter that for parameter recovery "it would be better for values to be drawn randomly across the whole range". The choice of using mean and stdev from participant estimates (R2.10) seem poorly suited for the bounded parameter ranges you specify.

Minor points:

- Typo: "The same was will be for w as we assume that..."
- parameter range for gamma and beta are not given
- if w was not estimated, how was it determined?
- Fig. S5 should use a binomial confidence interval. It's also unclear what the p-values correspond to
- Novelty bonus is sometimes described as η_B in contrast to another parameter η . But I thought η was the novelty bonus?

R.0.1: We thank the editor and the reviewers for their positive evaluation of our revision and the encouraging comments. We have now conducted more analyses, incorporated the requested missing methodological details and responded in detail all raised concerns below, including additional models in the model comparisons a suggested by Reviewer 4. As can be seen below, these additional analyses further corroborate our initial findings and plans.

As discussed with the editor directly, we believe that Reviewer 2's suggested additional pilot studies may be based on a misunderstanding and we believe conducting these would not change anything because the raised concerns do not affect any of the registered hypotheses. We now explain this in detail in the responses below. However, if the reviewer and editor insist that an additional pilot study is quintessential and if they could instruct what exactly they would like to see tested and how this is related to our registered hypotheses, we will happily conduct this additional pilot study.

Reviewers' Comments:

Reviewer #1 (Remarks to the Author):

Q1.0: Thanks for your well reasoned responses to my comments.

R.1.0: We thank this reviewer for helping us improve the paper and for acknowledging that we have addressed their concerns.

Reviewer #2 (Remarks to the Author):

Q.2.0. The authors have done a good job of addressing many of my concerns. I thank them for this. I have two major concerns that are still outstanding.

R.2.0. We appreciate that this reviewer is acknowledging our substantial efforts in the previous round of revisions. We have clarified and further detailed the remaining points that this reviewer raised, which we believe are mainly born out of a misunderstanding. In particular, we believe that we can convincingly show that meta-learning does not affect our registered hypotheses, which evolve around value-free random exploration, and which we can very well identify and isolate. However, it may be possible that we were unable to fully appreciate the worries this reviewer expressed. If the reviewer does not acknowledge our below efforts to resolve the issues, we would appreciate concrete guidance. We would also like to point out that all other reviewers (esp. Reviewer 4) were highly appreciative of our revisions and clearly state that we addressed the points they raised.

Q.2.1. The first relates to the task design and the issue of balancing outcome value and information (samples). I see that reviewer 4 also had concerns about this. I do not feel that the authors thoroughly addressed either my or reviewer 4's concerns here. As one specific: did the authors provide a figure showing the influence of trial and horizon on score, as a diagnostic plot as requested by reviewer 4? I apologise if so, I did not find it. I concur with reviewer 4's sentiment that: "These are important considerations, since participants could use meta learning to determine whether exploring the novel option pays off (e.g., if the pre-samples of the high value tree are already at the max apple size, you already know that the novel tree won't be better). In order to properly measure exploration, there needs to be uncertainty about the reward values. Some previous studies have used random scaling of reward values from round to round (e.g., Wu et al., 2020) to maintain uncertainty about which option has the highest mean reward."

R.2.1. We apologise if our previous responses were not clear enough in the previous revision. The requested information on the influence of block and trial were provided in Figure S4 in the previous version (figure of block effect and statistics of block and trial effects in the legend), and Reviewer 4 acknowledged that our responses fully addressed his queries. For completion, we have now added 3 subpanels to Figure S4 to visualize the effect of trial on reward.

Legend of Figure S4: "For each participant, we performed a regression on the average reward per block as well as a regression on the reward per trial. Slopes were compared to a vector of null slopes. When analysing the reward per block: (a) No significant slope was found for the short horizon reward ($t(60)=1.036$, $p=.304$, $d=.133$, $95\%CI=[-.0106,.0333]$), (b) the long horizons' first sample reward ($t(60)=-.286$, $p=.776$, $d=-.037$, $95\%CI=[-.0249,.0187]$) nor in (c) the long horizon's average reward ($t(60)=1.139$, $p=.259$, $d=.146$, $95\%CI=[-.0066,.024]$). Similarly, when analysing the reward per trial: (d) No significant slope was found for the short horizon reward ($t(60)=.342$, $p=.734$, $d=.044$, $95\%CI=[-.0004,.0005]$), (e) the long horizons' first sample reward ($t(60)=-.498$, $p=.62$, $d=-.064$, $95\%CI=[-.0005,.0003]$) nor in (f) the long horizon's average reward ($t(60)=1.683$, $p=.098$, $d=.215$, $95\%CI=[0,.0006]$). There is therefore no evidence that performance changed across time."

Q.2.2. The authors' inclusion of block is helpful but does not fully address the issues raised. I still believe that we need to know if better balancing bandit value across bandit types changes behavior on this task. The authors declined to address this as it would involve task redesign and repiloting and re-establishing effects. I do appreciate that but it seems that the advantages of pre-registration

come with the expectation that it is not 'too late' to ask for method issues to be addressed. Given this is an online task, I don't think it should take too much for the authors to pilot a version with alternate bandit values or potentially just a change to consistency in bandit colors (see below for more details), give both to the same participants in separate sessions, and see if estimates of parameters (ϵ, η) pertaining to value-free and novelty based exploration remain constant across both or not. If there is no meta-learning and no issue with balancing of bandit value on the parameters of interest then there should be good 'alternate version' reliability of parameter estimates across task versions.

R.2.2. We thank the reviewer for acknowledging that our analysis per block was helpful, and hope that the additional analysis of frequencies per trial (cf. Supplementary Information) and our new Figure S4 with an analysis of score per trial supports our point even further (cf. R.2.1). We would also like to clarify that there is no consistency in colour. The colour was randomly assigned to every bandit at every trial and does thus not carry any valuable information (colour: bandit main effect: $F(2.34, 53.91) = 0, p = 1, \eta^2 = 0$). This means that there is no specific colour associated with specific bandit-types which could facilitate meta-learning as the reviewer seems to suggest (please see response R.2.4. for more details).

Additionally, there seems to also be some confusion about the bandit types: the standard and the low-value bandit are clearly distinguishable (cf. R.2.8. for details). Importantly, meta-learning is irrelevant for the research question addressed here (i.e. value-free random exploration in impulsivity) and does not affect our behavioural measures and model parameter of interest. Please see R.2.8. for a detailed answer to this matter, in which we provide all the evidence demonstrating this. We therefore believe that the suggested additional pilot study will not bring any new information to our research question.

Q.2.3. A related issue here concerns clarity of task design. The additional information about task design provided by the authors is invaluable – this needs to be in the main manuscript. The task design is complicated and I still have questions about it – please can you clarify the below and also provide details addressing this in the manuscript, ideally with detailed information regarding the value of initial samples, the first draw and later draws (for the long horizon) as well as bandit means and distributions for each of the four bandit types. Further specific requests are given below.

R.2.3. We thank the reviewer for pointing out the omissions below. We have now moved the additional task description from the Supplementary Information to the main manuscript, as requested. We have also now better detailed the information about bandit means and distributions. Detailed information about the value of initial samples, first draw and later draws can be found in the newly added Table S2 (cf. R.2.5. and R.2.6. for details).

Methods: *"We constructed the reward and information of each bandit to be able to assess the contributions of different exploration strategies that have previously been put forward²³⁻²⁵. Each bandit is from one of four generative groups characterised by different means and number of initial samples, following the same procedure as other studies²⁴. The size of the apple is determined by its radius (cf. Fig. S1). Manipulating the amount of information subjects have before they make their choice (i.e. initial samples) avoids a potential reward-information confound²⁴. The samples of each bandit will then be sampled from a normal distribution with a fixed sampling variance ($\sigma^2 = 0.8$), truncated to $[2, 10]$, and rounded to the closest integer. Each mean was sampled from (overall, 1.4), with an "overall mean" overall specific to each bandit type. The overall mean was computed similarly to previous studies²⁴. On each trial we set the overall mean for one of the bandits, the 'certain-standard bandit', to be either 4.5 or 6.5. We determine the overall mean of the 'standard bandit' by adding a number sampled uniformly from $[-2, -1, +1, +2]$ to the*

certain-standard bandit overall mean. Similarly, we determine the overall mean of the ‘novel’ bandit by adding a number sampled uniformly from [-2, -1, +1, +2] to either the certain-standard bandit overall mean or the standard bandit overall mean. By doing this, we make sure that the means of those 3 bandits are comparable. This results in the means of the standard bandit and novel bandit spanning a slightly larger range compared to the certain-standard bandit means (cf. Table S2). To make sure that the ‘low-value’ bandit mean was always the smallest, its overall mean is computed by subtracting 1 to the minimum of the above-mentioned average means. Bandits also carry different amounts of information: The certain-standard bandit provides 3 initial samples, the standard bandit provides 1 initial sample, the novel bandit does not provide any initial samples and the low-value bandit provides 1 initial sample. Even though the absolute range of reward is set, randomly scaling each reward mean around the certain-standard bandits’ reward mean allows to maintain uncertainty about the overall average reward on each trial similarly to previous studies^{24,27}. On each trial, the average value of the certain-standard bandit initial samples is compared to the value of the standard bandit initial sample. The bandit with such higher value is referred to as the (expected) ‘high-value’ bandit. For detailed comparison between those average rewards cf. Table S3. At the beginning of each trial, the prior samples of the presented bandits are sampled from their respective distributions. We will ensure that the initial sample from the low-value bandit is the smallest by resampling from this bandit in the trials where it is not the case. For detailed information about the value of initial samples, first draw and later draws cf. Table S2. The order of all initial samples is then permuted to avoid biases.”

Q.2.4. To go into the weeds: in the current version of the task my understanding is that the bandits are as follows:

There are 4 types and on any given trial, 3 of the 4 bandit types are presented, each of these 3 bandits is a different color. There are 8 sets of colors.

Is each type of bandit shown in every possible combination equally often? Can any bandit type be any color? Is there any consistency?

R.2.4. We apologise for not being clear about the presented colours. We would like to highlight that the colour (intentionally) did not carry any information beyond any given trial (trial here refers to an interaction with one specific plantation, which is 1 sample in the short and 6 samples in the long horizon). We randomly assigned a colour for each of the three presented trees. Concretely, for each trial a random number between 1 and 8 is drawn using Math.random() function in javascript and rounded to the closest integer using Math.round(). Each one of the 8 sets has a fixed background (i.e. 3 bandits and the farmer boy Tommy; no previous samples on the wooden crate). On each trial, each bandit is associated to a tree using the function randperm() in Matlab. In summary, each bandit type can be any colour and there is no consistency as this is randomised.

For completion, we checked the frequency of the colours in the pilot data. For each of the 24 different colour combinations (8^3 , 8 sets of 3 different bandit colours), we computed the occurrence of each bandit ($N=4$). We averaged these across participants. We then performed an ANOVA with the subject identifier: colour and the within factor: bandit. As expected, there was no evidence of a difference in bandit occurrence for each colour (bandit main effect: $F(2.34, 53.91)=0$, $p=1$, $\eta^2=0$), meaning that each type of bandit is shown in every combination equally often. This has now been added to the Supplementary Information.

We would like to highlight here that we had no intention to study meta-learning in this task, but purely to assess value-free random exploration and its relationship with impulsivity/ADHD. We therefore did not intend that subjects learn across trials or assign any value/expectation to the colours. The participants were also not informed about the existence of four different tree categories, or that there were eight sets of 3 different colours in total. If we would have wanted to study meta-learning, we would have opted for a different task and study design.

Supplementary Information: “When performing an ANOVA with the subject identifier colour (8 sets of 3 different colors, $8^3 = 24$ different colors) and the within factor bandit there was no evidence of a difference in bandit occurrence for each colour (bandit main effect: $F(2.34, 53.91)=0$, $p=1$, $pes=0$), meaning that each type of bandit is shown in every combination equally often.”

Q.2.5. Bandit types

A) Certain-standard 3 samples, value : $\mu_{cs} \sim N(5.5, 1.4)$.

B) Standard. 1 sample, value : μ_s . mean μ_s is comparable to μ_{cs} , the trials are split equally between the four following: $\{\mu_s = \mu_{cs} + 1; \mu_s = \mu_{cs} - 1; \mu_s = \mu_{cs} + 2; \mu_s = \mu_{cs} - 2\}$.

‘On each trial the average reward of the certain-standard bandit initial samples is compared to the reward of the standard bandit initial sample. The bandit with such higher value is referred to as the ‘high-value’ bandit’

C) Low value 1 sample, low value: $\mu_l = \min(\mu_{cs}, \mu_s, \mu_n) - 1$.

D) Novel (no samples), value : $\mu_n =: \mu_s$

Can I check, is μ here the mean for the urn or the mean of the initial samples? As mentioned above we need information on both the bandit mean and range of values and on the mean and range of initial sample values (and also details of first draw and later draws for the long horizon)

R.2.5. We apologise for not being clear on this, which has now been clarified in the manuscript. The μ is the mean of the Gaussian distribution from which the samples were drawn. Each bandit provides samples according to the normal distribution: $N(\mu, 0.8)$. The samples are then rounded to the closest integer so that each apple takes one of 9 different sizes (absolute range: $[2, 10]$). By doing so, we closely followed the procedures used in previous exploration tasks (Wilson et al., 2014; Dubois et al., 2021). As suggested by the reviewer, we have now added a table (cf. Table S2) with each bandit’s value mean, standard deviation and range for the initial samples, short horizon 1st draw, long horizon 1st draw and long horizon later draws split by bandit combination condition.

Methods : “Manipulating the amount of information subjects have before they make their choice (i.e. initial samples) avoids a potential reward-information confound²⁴. The samples of each bandit will then be sampled from a normal distribution with a fixed sampling variance $.W(i, 0.8)$, truncated to $[2, 10]$, and rounded to the closest integer. Each mean i was sampled from $.W(i\&\#, 1.4)$, with an “average mean” $i\&\#$ specific to each bandit. The average mean was computed similarly to previous studies²⁴: On each trial we set the average mean for one of the bandits, the ‘certain-standard bandit’, to be either 4.5 or 6.5. We determine the average mean of the ‘standard bandit’ by adding the difference sampled uniformly from $[-2, -1, +1, +2]$ to the certain-standard bandit average mean. Similarly, we determine the average mean of the ‘novel’ bandit by adding the difference sampled uniformly from $[-2, -1, +1, +2]$ to either the certain-standard bandit average mean or the standard bandit average mean. By doing this, we make sure that the means of those 3 bandits are comparable. This results in the means of the standard bandit and novel bandit spanning a slightly larger range compared to the certain-standard bandit means (cf. Table S2). To make sure that the ‘low-value’ bandit mean was always the smallest, it’s average mean is computed by subtracting 1 to the minimum of the above-mentioned average means. Bandits also carry different amounts of information: The certain-standard bandit provides 3 initial samples, the standard bandit provides 1 initial sample, the novel bandit does not provide any initial samples and the low-value bandit provides 1 initial sample. Even though the absolute range of reward is set, randomly scaling each reward mean around the certain-standard bandits’ reward mean allows to maintain uncertainty about the overall average reward on each trial similarly to previous studies^{24,27}. On each trial, the average value of the certain-standard bandit initial samples is compared to the value of the standard

bandit initial sample. The bandit with such higher value is referred to as the (expected) 'high-value' bandit. For detailed comparison between those average rewards cf. Table S3. At the beginning of each trial, the prior samples of the presented bandits are sampled from their respective distributions. We will ensure that the initial sample from the low-value bandit is the smallest by resampling from this bandit in the trials where it is not the case. For detailed information about the value of initial samples, first draw and later draws cf. Table S2."

Q.2.6. Is it constrained such that there are equal number of trials where the standard is the high value bandit and where the certain-standard is the high value bandit? Is it also the case that there are trials where only one of these two bandit types is presented? Does this strategy mean that when only either the standard or certain-standard is presented it can end up being the 'high-value' bandit with a lower value than it would on trials where both these bandit types are present? Please can the authors break down the values of the high value bandit according to whether it is the standard or certain-standard and as a function of the combination of bandits presented. Ideally a table showing bandit type (standard, certain-standard, novel, low value) and value and range of both initial samples (where there are any), 1st draw and later draws (for long horizon) as a function of whether the S and CS are co-presented or not would be useful.

R.2.6. Yes, the reviewer is correct in his questions. Either of these bandits could be the higher valued bandit, and this with similar probability (when both were present with the novel bandit: $t(60)=0.543$, $p=0.589$, $95\%CI=[-3.22, 1.85]$; and with the low-value bandit: $t(60)=0.475$, $p=0.637$, $95\%CI=[-1.58, 2.56]$). This is detailed in the newly added Table S3. There were trials in which both of them were present together, and trials where only one of them was shown. We have clarified this in the revised manuscript.

Concretely, each trial belongs to one of four bandit combination conditions, depending which three bandits are presented: [Certain-standard, standard, novel], [Certain-standard, standard, low-value], [Certain-standard, novel, low-value] or [Standard, novel, low-value]. In the [Certain-standard, standard, novel] and in the [Certain-standard, standard, low-value] the high-value bandit can either be the standard or the certain-standard bandit, depending which one has the highest mean. As the bandit means are sampled from distributions with the same means, the number of time that one is bigger than the other is the same. For completion we have computed these frequencies (cf. Table S3) and indeed show that there is no difference in the number of times that the mean of the certain-standard bandit is higher than the mean of the standard bandit ($(\mu > \mu)$) and the number of times that the mean of the standard bandit is higher than the mean of the certain-standard bandit ($\mu^* > \mu + *$). This is the case in both the [Certain-standard, standard, novel] condition ($t(60)=-0.543$, $p=0.589$, $95\%CI=[-3.22, 1.85]$) and in the [Certain-standard, standard, low-value] condition ($t(60)=0.475$, $p=0.637$, $95\%CI=[-1.58, 2.56]$).

As mentioned above (cf. R.2.3) we have now added a table (cf. Table S2) with each bandit's value mean, standard deviation and range for the initial samples, short horizon 1st draw, long horizon 1st draw and long horizon later draws split by bandit combination condition. We also look at the certain-standard bandit and the standard bandit separately according to whether it is the high-value bandit or not. The initial samples' mean of the certain-standard bandit when it is the high-value bandit does not differ depending which bandits are presented with it (initial samples certain-standard bandit mean: [Certain-standard, standard, novel] vs [Certain-standard, novel, low-value]: $t(60)=0.26$, $p=0.796$, $95\%CI=[-0.132, 0.172]$; [Certain-standard, standard, low-value] vs [Certain-standard, novel, low-value]: $t(60)=0.514$, $p=0.609$, $95\%CI=[-0.101, 0.172]$). The initial samples mean of the standard bandit when it is the high-value bandit is indeed larger when presented with the certain-standard bandit compared to when it is presented alone (initial samples standard bandit mean: [Certain-standard, standard, novel] vs [Standard, novel, low-value]: $t(60)=15.6$, $p<.001$, $95\%CI=[1.09, 1.41]$; [Certain-standard, standard, low-value] vs [Standard, novel, low-value] : $t(60)=16.3$, $p<.001$,

95%CI=[1.13,1.44]). This is a result of the way that the bandit means were generated, which was done in the same way as previous studies (cf. Wilson et al., 2014), and which we have now detailed in the manuscript. Each bandit mean μ was sampled from $.7T(\mu, 1.4)$, with an “average mean” μ specific to each bandit type. In particular, the average mean of the standard bandit was computed by taking the average mean of the certain-standard bandit and adding a number sampled uniformly from [-2, -1, +1, +2]. By doing this, we make sure that the means of those 3 bandits are comparable, but this results in the means of the standard bandit spanning a larger range compared to the certain-standard bandit means. Because of this larger range, the average of the standard bandit means larger than the certain-standard bandit means ($\mu_{+*} > \mu^*$; depending the condition: 6.95(0.49) and 6.98(0.39); Table S2) is larger than the mean of the standard bandit means when the certain-standard bandit is not present (5.7(0.34); Table S2). In other words, as suggested by the reviewer, when only the standard bandit is presented it can end up being the ‘high-value’ bandit with a lower value than it would on trials where it is co-presented with the certain-standard bandit. We have now detailed this in the revised manuscript.

Importantly, even though there is a slight difference when splitting by ‘high-bandit’, there is no difference in the apple sizes of standard and certain-standard bandits across all trials (Table S2). Moreover, the number of trials in which either is the larger bandit does not differ (Table S3). This means that for the subjects both are giving the same apple sizes on average, and we find no evidence that subjects would learn to differentiate between them (see R.2.8).

Crucially, none of these analyses affects the low value bandit, which is what we are interested in in this registered report.

Methods: “Each mean was sampled from $.7T(\mu, 1.4)$, with an “overall mean” μ specific to each bandit type. The overall mean was computed similarly to previous studies²⁴: On each trial we set the overall mean for one of the bandits, the ‘certain-standard bandit’, to be either 4.5 or 6.5. We determine the overall mean of the ‘standard bandit’ by adding a number sampled uniformly from [-2, -1, +1, +2] to the certain-standard bandit overall mean. Similarly, we determine the overall mean of the ‘novel’ bandit by adding a number sampled uniformly from [-2, -1, +1, +2] to either the certain-standard bandit overall mean or the standard bandit overall mean. By doing this, we make sure that the means of those 3 bandits are comparable. This results in the means of the standard bandit and novel bandit spanning a slightly larger range compared to the certain-standard bandit means (cf. Table S2). To make sure that the ‘low-value’ bandit mean was always the smallest, its overall mean is computed by subtracting 1 to the minimum of the above-mentioned average means.”

Q.2.7. The authors state that ‘We will ensure that the initial sample from the low-value bandit is the smallest by resampling from each bandit in the trials where it is not the case. [...]’. It would be good to also understand how this procedure influences sample values across trials. Can the authors also please give the values for each bandit initial samples as a function of whether the low value bandit is one of the bandits included in a given set of 3.

R.2.7. Apologies for the imprecise wording. We are not resampling from each bandit but only from the low-value bandit if its sample was larger than another bandits’ initial sample (by chance). We have now clarified this in the methods. Additionally the values of initial samples for each different condition can now be found in Table S2.

Methods: “We will ensure that the initial sample from the low-value bandit is the smallest by resampling from this bandit in the trials where it is not the case. For detailed information about the value of initial samples, first draw and later draws cf. Table S2.”

Q.2.8. Putting to one-side the potential influence of the sampling strategies highlighted in Qus 3

(now Q.3.5) and 4 (now Q.3.6) above, I am concerned that there are imbalances in the design (as raised in my prior review) that could impact participants' heuristics in a manner that might be difficult to ascertain. The authors state that bandit colors are varied across trials: "The trees' positions (left, middle or right) as well as their colour (8 sets of 3 different colours) were shuffled between trials." This will mean the salient initial information participants have is the number of samples. Unless I'm missing something (apologies if so) this translates as follows:

$$\mu_{3\text{samples}} = \mu_{0\text{sample}} = \mu_{cs}$$

$$\mu_{1\text{sample}} = .5(\mu_s + \mu_l) = .5(\mu_{cs} + (\min(\mu_{cs}, \mu_s, \mu_n) - 1)).$$
 In practice, this value is going to be lower still due to the constraints put on the initial sample of μ_l .

Hence, a value-based heuristic would be to pick the 3 sample bandit when initial values are high, otherwise to pick the 0 sample bandit and to de-prioritize the 1 sample bandits. It is unclear how this would impact the parameter estimates (maybe ϵ_B will be underestimated and η_B overestimated?). It does however seem troubling, which is why I would like, at a minimum, to be reassured there is no difference in parameter estimates for the key parameters between this version of the task and one where there is no mean difference in value as a function of samples. As I suggested previously this could be achieved by also having 'low value' 0 sample and 3 sample bandits. An alternative which doesn't involve increased numbers of trials would be to just use 4 color trees and to keep the color of each bandit constant across trials so the standard bandit and low value bandits are clearly distinguishable (I believe the potential for meta-learning that this raises would be offset by reduction in the likelihood that participants simply lump standard and low value bandits together). If the authors chose this latter option, I would still suggest piloting it to ensure the effects established to date do not change.

R.2.8. We thank the reviewer for suggesting that subjects could have used a sophisticated strategy and guide their behaviour based on the number of initial samples.

First, we would like to highlight that the colour did not carry any information across trials because this was randomly assigned to the bandits. We did so because we are not interested in meta-learning, and neither of our hypotheses is related to meta-learning. We therefore cannot see any benefit for fixing the colours for each bandit, as this may indeed introduce meta-learning, which is not the focus of this paper.

If we understand correctly, then the reviewer suggests that subjects learn over time that the 3-sample bandits lead (on average) to a better outcome than the 1-sample bandits ('standard and low value bandits together'). We believe this is a highly unlikely strategy and we can find no evidence for such a tendency. If this were true, the subjects would over time be preferring the certain standard over the standard bandit. We find no evidence for this being the case, neither in the short horizon (linear regression between [certain-standard bandit frequency minus standard bandit frequency] and [trial]: slope vs null slope: $t(60)=0.516$, $p=.608$) nor in the long horizon ($t(60)=-0.794$, $p=.43$).

Moreover, we also find no evidence for choosing any 1-sample bandit less compared to the 3-sample bandit over time, neither in the short horizon (linear regression between [3-sample bandit frequency minus 1-sample bandit frequency] and [trial]: slope vs null slope: $t(60)=0.85$, $p=.399$), nor in the long horizon ($t(60)=-0.326$, $p=.746$). This clearly speaks against the reviewer's hypothesis. We believe that the reason why subjects did not 'lump' them together was because the low-value bandit's initial sample is clearly smaller than the other samples, which makes it unlikely that they conflate all 1-sample bandits. Those analyses have now been added to the Supplementary Information. Most importantly, we would like to highlight that neither of these analyses and concerns affects our registered hypothesis. In this study, we are interested in studying association of the low-value bandit and impulsivity/ADHD. As detailed in the previous set of revisions, we find no evidence for meta-learning or any changes in the utilisation of the low-value bandit over time. We previously showed (cf. R.2.3 in the first revision) that the low-value bandit frequency does not change over blocks, and our computational modelling further highlighted this by showing that adding a time-changing

epsilon-greedy parameter did worsen our model fits. Similar to our additional analysis presented above (cf. R.2.1), we have now additionally investigated whether there is any trial effect on the low-value bandit (in the previous revisions, we used block number which is slightly less fine-grained). As expected, we do not find any significant effect of trial number on the low-value bandit picking, neither in the short horizon (linear regression slope vs null slope: $t(60)=-1.71$, $p=.092$) nor in the long horizon ($t(60)=-1.47$, $p=.147$). This further corroborates our argument that there is only minimal (if any) meta-learning in the task, and this does not affect the task measures we are interested in. We therefore believe that our design is entirely valid, even without conducting additional pilot studies. The trial analysis for individual bandits can now be found in the Supplementary Information.

Supplementary Information: “ [...] when analysing the frequency of picking the low-value bandit per trial, no evidence of changes across time were observed, neither in the short horizon (linear regression slope vs null slope: $t(60)=-1.71$, $p=.092$) nor in the long horizon ($t(60)=-1.47$, $p=.147$). [...]. We found no evidence that subjects prefer the certain-standard bandit over the standard bandit, neither in the short horizon (linear regression between [certain-standard bandit frequency minus standard bandit frequency] and [trial]: slope vs null slope: $t(60)=0.516$, $p=.608$) nor in the long horizon ($t(60)=-0.794$, $p=.43$). Moreover, we also find no evidence for choosing any 1-sample bandit (standard or low-value bandit) less compared to the 3-sample bandit over time, neither in the short horizon (linear regression between [3-sample bandit frequency minus 1-sample bandit frequency] and [trial]: slope vs null slope: $t(60)=0.85$, $p=.399$), nor in the long horizon ($t(60)=-0.326$, $p=.746$).”

Q.2.9. Following on the issues outlined above, participants might well treat the standard and certain-standard bandits very differently. The choice to collapse these in the results section is, in my opinion, hence suboptimal. Could the authors not simply provide results for the 4 bandit types? Certain-standard, standard, novel, low value?

R.2.9. We thank the reviewer for this suggestion, which we indeed used in the very first versions when we developed this task. However, we opted for the current version, because it allows us to better illustrate exploration and exploitation. Concretely, by identifying the high-value option in each trial we are identifying the exploitation option and therefore provide a ‘cleaner’ exploration, and horizon effect. This is also how we have presented our findings in our previous study (cf. Dubois et al., 2021, *eLife*).

We agree that providing information about the certain-standard and standard bandit is still important and we are happy to provide this in the context of this paper. We have now added this as an additional figure (Fig. S12) which we will keep the Stage 2 report, even in the main manuscript if the reviewer so wishes.

Supplementary Information: “The frequency of picking the certain-standard bandit was increased in the long versus short horizon (pilot data: $t(60)=9.825$, $p<.001$, $95\%CI_M=[5.653,8.544]$, $d=-1.258$, $95\%CI_ES=[-1.605,-0.926]$; Fig. S12a). Similarly, the frequency of picking the standard bandit was increased in the long horizon (pilot data: $t(60)=5.489$, $p<.001$, $95\%CI_M=[1.881,4.037]$, $d=-0.703$, $95\%CI_ES=[-0.989,-0.423]$; Fig. S12b).”

Q.2.10. Multiple comparisons.

The authors have done a commendable job in clarifying their hypotheses and proposed tests in different stages of the study. I would ask that they provide a little further detail still. Specifically, I agree that there is no need for multiple comparison if the authors are only deeming a hypothesis proven if two separate measures (e.g. impulsivity and ADHD symptom, scores) both show a relationship to a given parameter of interest. Please can they clearly specify those tests where this

is the case. Where this is not the case and multiple comparisons are required, please can the authors provide sample size / power calculations taking this into account.

R.2.10. We thank the reviewer for acknowledging our efforts in clarifying our hypothesis and tests. We agree that the multiple comparisons have not been clearly pointed out. We have now revised the Methods section to provide these additional details.

Methods: *“For Step 2’s sample size estimation, where the link between exploration and impulsivity is investigated, the correlation coefficient of our previous study using the same task²² was used for our power analysis. In this study, a Pearson correlation of $R=0.26$, $p=0.01$ was observed between an impulsivity measure⁶² (the Conners ADHD questionnaire) and value-free random exploration in youths. Assuming a similar correlation in adults, our power analysis suggests that a sample of $N=190$ is sufficient to reach 95% power (G*Power suggests a similar sample size of $N=186$). This moderate size correlation factor is in line with previous studies linking BIS-measured impulsivity to behaviour (e.g. with delay discounting⁶³⁻⁶⁶). Similarly, for Step 2’s Stage 2 exploratory analysis in which we will look at the correlation between value-free random exploration and the three subdomains of BIS, G*Power analyses suggest that a sample size of $N=228$ is sufficient to reach a 95% power at a significance corrected for multiple comparisons using Bonferroni correction. (...).*

We will look at the correlation between the BIS total score and the low-value bandit frequency, and between the BIS total score and the ϵ -greedy parameter (cf. Table 1). (...). Considering that impulsivity is a broad heterogeneous construct^{1,3-5}, in Stage 2 we will perform an exploratory analysis of the three subdomains of BIS (i.e. attentional, motor and non-planning behaviour⁶⁹) similarly to previous studies⁷⁰. We will investigate whether value-free random exploration is linked to a specific subdomain by looking at the correlations with each of them. Specifically, we will look at the correlation (corrected for multiple comparisons using Bonferroni correction) between the low-value bandit frequency and the BIS subdomains: attentional, motor and non-planning, as well as the correlation (corrected for multiple comparisons using Bonferroni correction) between the ϵ -greedy parameter and the BIS subdomains: attentional, motor and non-planning. (...). We will look at the correlation between the ASRS total score and the low-value bandit frequency, and between the ASRS total score and the ϵ -greedy parameter (cf. Table 1). (...). In Stage 2 we will additionally perform an exploratory analysis of the sub-scales of the ASRS (i.e. inattention, hyperactivity-impulsivity). Specifically, we will look at the correlation (corrected for multiple comparisons using Bonferroni correction) between the low-value bandit frequency and the ASRS sub-scales: inattention and hyperactivity-impulsivity, as well as the correlation (corrected for multiple comparisons using Bonferroni correction) between the ϵ -greedy parameter and the ASRS sub-scales: inattention and hyperactivity-impulsivity.”

Q.2.11. For the Stage 2 exploratory analyses (e.g. using autism, cognitive flexibility measures, latent compulsivity and anxiety/depression factor scores etc.) it is still important that the tests to be explored (i.e. which measures related to which parameters) are listed at this point so the correct multiple comparisons can be determined. An alternate would be to take any interesting findings and replicate in a separate sample. Please can the authors clarify which of these approaches they plan to take and provide full test and multiple comparison details if the former.

R.2.11. In line with the previous studies that uses the same approach (cf. Gillan et al., 2016; Rouault et al., 2018; Seow et al., 2020), we expect to obtain a 3 factor solution, comprising an impulsivity dimension, a depression / anxiety dimension and a compulsivity dimension. As mentioned before, we have several concrete hypotheses for these dimensions. For example, previous studies have demonstrated increases in exploration in OCD patients (cf. Kanen et al., 2019; Hauser et al., 2017), but it is not clear which exploration strategy is concerned. We will therefore look at the correlation

(corrected for multiple comparisons using Bonferroni correction) between the compulsivity dimension and each exploration free parameter (depending the model). Similarly, studies have demonstrated abnormality in exploration in patients with depression (cf. Blanco et al., 2013), anxiety (cf. Browning et al., 2015) and other avoidance of uncertainty disorders (cf. Addicott et al., 2017). However, here as well, different exploration strategies have not been tested. We will therefore look at the correlation (corrected for multiple comparisons using Bonferroni correction) between the depression/anxiety dimension and each exploration free parameter (depending the model).

Based on the suggestions by the reviewers in the previous round of reviews, we have also added a measure for cognitive flexibility and autism, which we are addressing separate questions. As suggested by Reviewer 4 (cf. R.4.3. in the previous revision), because autism has overlapping symptoms with ADHD (cf. Mayes et al., 2012), we will also look at the correlation between the autism scale, AQ-10 total score and value-free random exploration. Depending on the association between our impulsivity measure and the autism score, we may conduct further analysis using partial correlations. Lastly, as suggested by Reviewer 1 (cf. R.1.3. in the previous revision), because cognitive flexibility is thought to play a role in the exploration-exploitation trade-off (cf. Good et al., 2013; Laureiro-Martinez et al., 2010), we will look at the correlation between the cognitive flexibility scale (CFS) and value-free random exploration. If there are meaningful associations, we may further explore associations using mediation analyses.

We have now also revised the Stage 2 section of the manuscript to point out these different analyses, and how we correct for multiple comparisons. We hope the reviewer agrees that this is now sufficiently clear.

Methods: "As a second step, we will investigate whether exploration correlates with other factors. In particular, similar to previous studies^{41-43,74,75}, we expect to retrieve a depression / anxiety dimension, on which depression, social anxiety and anxiety load (SDS, LSAS and STAI-Y2 questionnaires respectively) and a compulsivity dimension, on which OCD and uncertainty intolerance traits load (OCIR and IUS questionnaires respectively. Indeed, previous research has found that impulsivity and compulsivity only shows a modest overlap⁷⁶, which is also why previous studies that used factor analyses have found that these items load on different factors^{41,43}. Previous studies have demonstrated increases in exploration in OCD patients^{45,77}, but it is not clear which exploration strategy is concerned. We will therefore look at the correlation (corrected for multiple comparisons using Bonferroni correction) between the compulsivity dimension and each exploration free parameter (depending the model). Similarly, studies have demonstrated abnormality in exploration in patients with depression⁴⁸, anxiety⁴⁹ and other disorders related to avoidance of uncertainty⁴⁶. However, here as well, different exploration strategies have not been tested. We will therefore look at the correlation (corrected for multiple comparisons using Bonferroni correction) between the depression/anxiety dimension and each exploration free parameter (depending the model). We will also investigate two separate questions. We will look at the correlation between the autism scale, AQ-10 total score and value-free random exploration, as autism has overlapping symptoms with ADHD⁷⁸. Depending on the association between the autism score and our impulsivity measure, we may conduct further analysis using partial correlations. We will also look at the correlation between the cognitive flexibility scale (CFS) and value-free random exploration, as cognitive flexibility is thought to play a role in the exploration-exploitation trade-off^{79,80}."

Reviewer #3 (Remarks to the Author):

Q3.0: The authors have responded thoroughly to my comments and I am happy to recommend Stage 1 IPA. Best of luck with the study!

R.3.0: We thank the reviewer for their help feedback in the first round of reviews and for their positive evaluation of our response.

Reviewer #4 (Remarks to the Author):

Thank you for the great amount of effort that has been put into addressing the reviews. I only now see that I'm one of four reviewers, who all seemed quite interested in the study and generated a large volume of feedback. I hope that the sheer quantity of comments from myself and the other reviewers does not dissuade you from pre-registrations in the future. Even for a journal of this status, it's clear that the amount of work that has gone into the revision is quite substantial. I applaud the authors for the thoroughness of responses and believe that the manuscript in its current form meets the requirements for provisional acceptance.

R.4.0: We thank this reviewer for this very positive endorsement of our revisions and for acknowledging our efforts. We are grateful for the reviewers' previous feedback that helped us strengthen our paper. We also thank the reviewer for the additional comments and for pointing out the lack of clarity and missing information for the points below. We have now addressed all of them as detailed below.

My comments have largely been addressed, although I would also like to provide a couple of comments about some results that were unclear. I'm not sure if this is the correct moment to raise them, but perhaps addressing them now will prevent additional work at a later stage. Thus, these comments should not be taken as requirements before provisional acceptance, but as suggestions for how to improve the interpretability of the results when analyzing the full data set.

Q.4.1. I still have confusion about the model comparison in Fig. S6. It's a bit unclear what the UCB and hybrid models refer to. Do UCB and the hybrid always use gamma (uncertainty bonus) and beta (softmax inverse temperature)? Specifically, I'm wondering if the latter variants of UCB and the hybrid (with epsilon and eta) still include gamma and beta? Specifically, inverse temperature (beta) and value-free exploration (epsilon) are potentially interchangeable. I wonder if the obtained results are dependent on the fact that the UCB comparison always includes inverse temperature in addition to epsilon. If the key hypothesis "is that the value-free random exploration parameter is associated with impulsivity", then perhaps it would also be worthwhile to include a comparison to UCB without beta (i.e., beta fixed to 1) and then also add epsilon or epsilon+eta variants. Otherwise, your UCB results could be interpreted as being unnecessarily handicapped. As I mentioned in my previous comments (sec 4.6 of the rebuttal letter), you are instantiating as many as 4 different exploration strategies, and the current model comparison isn't quite sufficient to rule out all alternative hypothesis in competition with your key hypothesis.

R.4.1. We thank you for pointing out, and apologise for the lack of clarity regarding each model's parameter configuration. We have now added a table (cf. Table S4) which clearly summarises all parameters.

In our previous analyses, all 'UCB' models (as correctly interpreted by this reviewer) included all parameters, i.e., they were equipped with the beta parameter. Based on this reviewer's suggestion, we have now looked at the suggested additional models (cf. Fig. S6). Importantly, our previous overall winning model is still the winning model, confirming the validity of our model selection. Interestingly (and as predicted by this reviewer), a free beta parameter generally improves model fits, but in the most complex model the model with a fixed inverse temperature parameter (=1) indeed has a lower BIC value. We have added this to the paper, and now mention that we will investigate both models in our analysis.

Methods: "[...] we will also look at the UCB models with fixed inverse temperature parameter (=1). This leads to a total of 16 models (see the labels on the x-axis in Fig. S6)."

Q.4.2. The model recovery simulations (Fig. S7b) suggest there are some potential mis-identification issues, where data generated by the UCB+epsilon+eta model is also quite likely to be recovered by the winning thompson+epsilon+eta model, but not vice versa. It's difficult to read the exact percentage of this model mimicry from the figure, due to the existing color legend (please consider a more contrastive palette in the future; see <https://www.nature.com/articles/s41467-020-19160-7>), but it looks non-negligible. If you were to calculate the inversion matrix $p(\text{sim} | \text{recovered})$ (c.f., Wilson & Collins, 2019 box 5 Fig. 1), it seems that your results in favor of thompson+epsilon+eta could also reflect a decent probability of the underlying data actually resulting from the competing UCB+epsilon+eta model. Intuitively, this means that participants could be behaving like the UCB+epsilon+eta model, but your task and model-comparison procedure attributes it to the Thompson+epsilon+eta model. This could perhaps be something worth mentioning in the general discussion of the final paper as a potential limitation, if similar results are also obtained in the full sample.

R.4.2. We thank the reviewer for suggesting a more contrastive palette (cf. Thyng et al., 2016; Cramer et al., 2020) which we have now incorporated. We also thank the reviewer for raising this point, which we had not explicitly mentioned. Indeed, we believe that the confusion matrix results support the notion of the full-UCB model being overly complex and thus punished. It is thus likely that the conservative BIC punished the high complexity of the UCB model, supporting the simpler (but equally versatile) full-Thompson model. We agree this is an interesting point and we now mention it in Figure 7 legend and we will mention it in the discussion in the Stage 2 report.

Methods: *"The 'matter' gallery of the 'cmocean' colourmap was used of the figures^{69,70}"*

Figure S7 legend: *"The lower recovery of the full-UCB model is likely to reflect the conservative nature of the BIC which punishes its high complexity to the advantage of the simpler (but equally versatile) full-Thompson model."*

Q.4.3. I also agree with R2 in 2.10 of the rebuttal letter that for parameter recovery "it would be better for values to be drawn randomly across the whole range". The choice of using mean and stdev from participant estimates (R2.10) seem poorly suited for the bounded parameter ranges you specify.

R.4.3. We thank the reviewer for this suggestion. We originally chose to use (truncated) Gaussian distributions to sample the parameters for this analysis, because the parameter distributions of the participants' data did not look too dissimilar. However, we agree this may be suboptimal. We have thus now performed an additional, third, parameter recovery analysis (cf. Fig. S13), where - instead of sampling from the distribution defined by the mean and variance of the fitted parameter values (cf. Fig. S7-8) - we sample directly from the originally fitted parameter values. This analysis yielded very similar results as our previous two parameter recovery analyses, further strengthening our results. We now present all of them in the revised registered report.

Minor points:

Q.4.4. Typo: "The same was will be for as we assume that..."

R.4.4. Thank you, this has now been corrected.

Supplementary Information: *“The prior mean θ_0 will be fitted to both horizons together as we do not expect the belief of how good a bandit is to depend on the horizon. The same holds for β , as we assume that the arbitration between the UCB model and the Thompson model does not depend on the horizon.”*

Q.4.5. Parameter range for gamma and beta are not given

R.4.5. We apologise for this omission, this has now been added.

Supplementary Information: *“The parameters will be permitted to vary within the following bounds: $\theta_0 = [0.01, 6]$, $\beta = [1, 10]$, $c = [0, 0.5]$, $\gamma = [0, 5]$, $\delta = [0, 0.5]$, $\eta = [1.4, 5]$, $\lambda = [0, 1]$.”*

Q.4.6. If w was not estimated, how was it determined?

R.4.6. The parameter w is indeed a free-parameter that will be estimated. We apologise if this was not clearly formulated.

Supplementary Information. *“All the parameters besides θ_0 and β will be free to vary as a function of the horizon as they capture different exploration forms: directed exploration (information bonus ; UCB model), novelty exploration (novelty bonus), random exploration (inverse temperature ; UCB model), uncertainty-directed exploration (prior variance θ_0 ; Thompson model) and value-free random exploration (c -greedy parameter). The prior mean θ_0 will be fitted to both horizons together as we do not expect the belief of how good a bandit is to depend on the horizon. The same holds for β as we assume that the arbitration between the UCB model and the Thompson model does not depend on the horizon.”*

Q.4.7. Fig. S5 should use a binomial confidence interval. It's also unclear what the p-values correspond to

R.4.7. We thank the reviewer for noticing this, we have added the correct figure references in the main manuscript. We have also added the binomial confidence interval on Fig. S5 and have done the same for the certain-standard bandit and standard bandit frequencies (cf. Fig. S12).

Methods: *“This will be reflected on the frequency of picking the high-value bandit, which we predict to be decreased in the long horizon (pilot data: $t(60)=8.45$, $p<.001$, $95\%CI_M=[6.92,11.211]$, $d=-1.082$, $95\%CI_ES=[-1.407,-0.769]$; Fig. S5), and similarly on the frequency of picking the low-value bandit, which we predict to be increased in the long horizon (pilot data: $t(60)=-3.446$, $p=.001$, $95\%CI_M=[-1.568,-0.416]$, $d=0.441$, $95\%CI_ES=[0.178,0.708]$; Fig. S5). [...]. This will be largely reflected on the frequency of the novel bandit, which we predict to be increased in the long horizon (pilot data $t(60)=-8.586$, $p<.001$, $95\%CI_M=[-11.178,-6.954]$, $d=1.099$, $95\%CI_ES=[0.784,1.427]$; Fig. S5).”*

Q.4.8. Novelty bonus is sometimes described as η_B in contrast to another parameter η . But I thought η was the novelty bonus?

R.4.8. We apologise for the confusion. Eta is indeed the novelty bonus, and eta_B is how we refer to the block-dependent quantity of the novelty bonus in our analysis per block. We have clarified what this term is.

Supplementary Information: *“Given the observed minor change in the novelty bandit frequency across blocks and trials (cf. above), we extended our model comparison with a model comprising a block-dependent quantity of the novelty bonus, which we named γ .”*

Reviewers' Comments:

Reviewer #2 (Remarks to the Author):

I am happy to confirm that I feel that the Stage 1 submission is now suitable for in-principle acceptance.

I note that the authors were incorrect in believing there was a misunderstanding. I did not think the colors were assigned to given urn types - indeed I was worried that this would lead to the most salient information being number of samples potentially resulting in differential treatment of urns with 3 versus 1 initial sample. I hence suggested changes to the design that might address this. However, the authors have now provided additional analyses comparing choice of urn as a function of initial samples provided and comparing choice of the standard versus certain standard urn and addressing whether these choices vary across time. The results of these analyses are compelling and together with the other requested details provided fully address my outstanding concerns. I thank the authors for their patience and willingness to provide this information.

Reviewer #4 (Remarks to the Author):

The revisions and further clarification in the response letter have substantially improved the manuscript. I will again recommend it for provisional acceptance.

In particular, I'm quite happy with the additional validation of the models. I'm glad that the authors agree the interpretation of the model recovery in Fig. S7b may be a potential issue. But I would be satisfied if they could include the inversion matrix I had previously requested (c.f., Wilson & Collins, 2019 box 5 Fig. 1), and mention it in the discussion, in the round 2 report. The inversion matrix would help clarify the extent to which the thompson sampling model winning in the model comparison, in fact provides evidence that people behave like said model, as opposed to the thompson sampling model mimicking the UCB model. I would also expect that this model recovery analysis includes the fixed beta UCB model shown in Fig. S6. It's fine if there are some limitations to which these models can be distinguished, but I would recommend that these limitations be clearly laid out in the discussion of the final paper, and that alternative hypotheses (i.e., models) be adequately investigated in case they lead to different interpretations of the results.

I also wanted to clarify that although the statement in R.2.1 of the rebuttal letter "Reviewer 4 acknowledged that our responses fully addressed his queries" wasn't exactly true (I said "my comments [had] largely been addressed" in the previous response, at risk of sounding pedantic), the latest revisions have now addressed them. Like R2, I was previously still somewhat concerned about the same aspects of the task design. But the updated methods clarifies that the mean of the novel bandit was actually sampled from a uniform distribution. This combined with the lack of meta-learning as shown in Figure 4 helps to put these worries to rest. I also appreciate that the task has been published online, which gave me a much better understanding of the dynamics of the task.

I look forward to reading the 2nd round report and I'm grateful to the authors for pre-registering their experiment and for the thoroughness of the previous revisions.

I sign my review,
Charley Wu

Review Completed Study (Stage 2)

Reviewers' Comments:

Reviewer #1 (Remarks to the Author):

This study is an impressive follow-up to the pre-registered proposal that was reviewed and approved previously. The authors have used a new paradigm to explore various forms of 'exploration' and a hypothesised relationship to the dimensional trait of impulsivity. This is combined with a sophisticated computational approach following online testing of a large number of healthy volunteers who also complete a number of relevant self-report questionnaire scales, including for ADHD, anxiety and depression, also subjected to principal component analysis.

In relation to Stage 2 ms evaluation, I confirm that the authors have kept to the same protocol and experimental procedures as laid out in Stage 1. They have included the outcomes of all the pre-registered analyses. The analyses are presented exhaustively in the main text and in 32 pages of supplementary information.

The main results are clear. Their test paradigm is again validated as a means of directing various exploration strategies. The only form of exploration linked to impulsivity (tested according to the Barratt Scale) is 'random exploration'. This is related most significantly to the Barratt motor impulsivity sub-scale. The significant relationship is not especially high ($r < 0.2$) but significant at $p < 0.001$. There were no relationships with other strategies based on uncertainty driven Thompson sampling or 'novelty exploration'.

In further exploratory (i.e. not pre-registered) analyses the authors show a significant relationship between 'novelty exploration' and anxiety/depression scores. This analysis appears sound, adds to the ms and is appropriately caveated.

The authors have delivered the precise analyses that were approved previously to show an hypothesised, but previously untested, relationship between impulsivity and exploration, with possible relevance to ADHD. This novel result certainly deserves publication. They also, more tentatively show a degree of dissociation with another dimension, of anxiety/depression that further suggests possible applicability to neuropsychiatric disorders. They perhaps do need to consider possible underlying mechanism(s) for these findings- other than linking them with central noradrenergic functioning.

Specific comments

1. Title. Is this fully justified? I do not believe any "role" for exploration has been demonstrated in impulsivity, only an association or relationship. The implication of the title is that individual differences in exploration contribute to impulsivity but this has not been shown here; only a correlation between the two.

2. Sample. I could not find any details on the $n=580$ participants in terms of gender, age and other demographics, even in the Supplementary Material. Were age and IQ considered as co-variates?

3. The task is ingenious, though complex. Understanding it does rather depend on reference to other papers. Can its description be improved for the non-expert reader?

4. The main result, that impulsivity is linked to random choice responding, sounds as though it could simply be the product of impaired general cognitive control (or even IQ- 'g'). In the Friedman and Miyake (2006) scheme, this could also explain why it is not related to cognitive flexibility per se. The authors might wish to consider their findings in terms of generally deficient fronto-executive function (in addition to speculation regarding noradrenaline, the effects of which are likely mediated by prefrontal cortex modulation) .

5. It wasn't clear to me why and how enhanced novelty preference would necessarily be associated with increased levels of anxiety/depression. Intuitively, the opposite relationship might perhaps have

been expected?

6. It should perhaps be acknowledged as a limitation of the study that the clinical significance of these interesting results needs to be tested more directly in clinically diagnosed patients.

Reviewer #2 (Remarks to the Author):

The authors have carefully tested the hypotheses as laid out in their stage 1 manuscript. They clearly differentiate between planned analyses and post-hoc enquiries. Their results support their a-priori hypothesis regarding an association between impulsivity and random exploration. The finding that this association is even stronger for a latent dimension of impulsivity (obtained by conducting a factor analysis across item-level scores from multiple measures) than for individual measures is particularly nice as is the finding that the association holds both for the model parameter (ϵ -greedy) designed to capture the influence of value-free random exploration and when using frequency of choice of the low-cost option as a proxy for value-free random exploration.

Adherence to proposed methods.

The one place where the authors deviate from their registered plans is in relation to sample size. I am not surprised that the initially pre-registered sample size was insufficient. The authors should specify whether they waited until the final sample was obtained to run the planned analyses or conducted initial analyses when the pre-registered sample size was achieved.

It is also hard to follow how many subjects were used in different analyses due to the authors' omission of degrees of freedom when reporting statistics – it would be useful if the authors could include these in any revision of the manuscript. In particular, did the authors use the full $N=580$ sample for the group level analyses as well as the trait / individual difference analyses reported in this manuscript? (I believe the authors had originally planned to use a smaller sample for the former than for the latter).

Please can the authors specify if anonymized data will be made available to readers, alongside code for the models used.

Additional comments

At a group-level the different forms of exploration are all elevated when the horizon is long, in line with the authors' hypotheses. The authors also point out that the first choice in the long horizon produces less points than the single choice in the short horizon but that mean performance in the long horizon condition is better than that in the short. This does suggest, as they argue, that exploration is being used to the benefit of performance. It would be helpful here if the authors could test for direct links between engagement in a given exploration strategy in the long condition and performance, using either the model parameters or/and the frequency of choices of the different options. In particular, is increased engagement in random exploration linked to better performance in the long horizon condition? Or is it that engagement in any one of the three exploration strategies improves performance, but the one selected is less important? This would be useful to know.

The authors note that impulsivity increases value-free random exploration across horizons. It would be useful if the authors could address this in the context of whether value-free exploration is beneficial to performance in the long horizon relative to the short horizon and whether impulsivity is linked to poorer or better performance in one or both conditions.

As the authors recognize, their exploratory finding that elevated scores on their latent anxiety/depression factor is linked to increased novelty-based exploration seems rather surprising given the well-established link between anxiety and depression and intolerance of uncertainty. It would be nice if the authors could devote a little more space to discussing this finding.

It would also be helpful if the authors can address whether scores on the anxiety-depression factor and engagement in novelty-based exploration are linked to better or worse performance in each

horizon condition.

The STAI can be broken down into both anxiety items and depression (predominantly anhedonia-related) items. It would be helpful to know whether it is primarily the latter that are loading on the anxiety/depression factor.

Minor:

Supplementary Figure 4 does not seem to be referenced in the main text (it jumps from Fig S3 to Fig S5)

Discussion – would we expect motor impulsivity in particular to be linked to noradrenergic function?

Reviewer #3 (Remarks to the Author):

This is a signed review (Chris Chambers, Cardiff University). I was a reviewer for the Stage 1 manuscript and I'm happy to see the Stage 2 completed successfully.

On a positive note, the study appears to have been undertaken closely in line with the protocol, with only some minor deviations that are transparently flagged.

However, I would recommend some significant revisions to the reporting structure of the results. First, in my view, to ensure clarity for readers, the study design table should be returned to the main text and a column added to the right of the table indicating the outcome of each specific hypothesis. Second, the results section should be revised to align its structure exactly with the questions that are asked in the question column of the design table, with clear and precise references to which test outcomes are diagnostic about which specific numbered hypothesis. A nice way to do this that I have seen in other Stage 2 RRs is provide a simplified answer to each question that is then expanded to reveal the outcome of each hypothesis test. e.g. "Are exploitation and exploration horizon-modulated?" "Yes. [expand]" etc. This isn't just a stylistic suggestion -- it is intrinsic to the transparency of the RR format that the narrative reporting of confirmatory results is tightly aligned with the confirmatory design structure.

I should also emphasise that in making this recommendation I am not at all implying that the authors are deliberately obscuring the transparency of their reporting. I can see how the authors could arrive at the conclusion that the current approach reads nicely (and it does). But ensuring tight links between confirmatory plans and confirmatory outcomes is the most important priority for RRs. Ambiguity in this area is not uncommon in Stage 2 RRs due to unfamiliarity with the requirements of the format.

Reviewer #4 (Remarks to the Author):

I am very grateful to have been part of the review process for this pre-registered manuscript, and greatly enjoyed reading the 2nd stage report with the full dataset.

Given the pre-registered nature of this manuscript, I will focus my comments on the clarity and interpretation of the results, of which I have several suggestions. In addition, I have a few analysis-related comments, which are not new but reiterate previous feedback I had provided in early stages of the review process. I'm not sure if the ship has sailed to provide feedback of this nature, but I will provide them regardless, while also acknowledging the fact that I had previously approved the analysis plan. I also have no objection to this paper getting published, but would like to first see my comments addressed in a revision.

1. The clarity of writing and accuracy in representing the current state of theories in the field seems

somewhat wanting, especially for the standards of this journal. Perhaps I should have provided more detailed feedback during the previous review stages, but I had assumed they would be improved upon during the 2nd stage of submission. I highlight some main issues here, while I include additional comments in the “Minor comments” and “Grammatical issues” sections below

- i. The distinction between complex and heuristic exploration strategies lacks clarity. By some accounts, all of the different strategies can be considered heuristic. Thompson sampling is just a probability matching heuristic, while UCB is widely considered a heuristic as well. In the discussion, this distinction is clarified based on whether means and uncertainty estimates need to be represented, which would be helpful if included in the introduction. There is a remaining leap of logic however, to relate representing means and uncertainties with additional computational costs. This could be provided by reviewing previous work using experiments implementing external limitations on cognitive resources through working memory (Cogliati Dezza et al., 2019) or decision-time manipulations (Wu et al., 2021), which have found a reduction in uncertainty-directed exploration, but increases in random exploration. Connecting with this literature could improve the theoretical development of the main argument
- ii. The argumentation structure of the paragraph beginning at the end of page 4 and ending on page 5 seemed a bit all over the place. This is a key section of the text that justifies the interpretation of the results and main contributions of the manuscript, which I expect could be improved quite substantially
- iii. Perhaps I am misunderstanding something, but effect sizes of $\eta^2 = 0.03$ is considered a small effect. A fair and balanced interpretation of these results should also qualify the relative weakness of this relationship. I also felt that later exploratory analyses relating model parameters to survey scores and latent factors could benefit from a quantification of the variance explained. The latent factor loadings could also benefit from a quantification of how much the “impulsivity-related questionnaires (BIS and ASRS) primarily loaded onto one factor (labelled as impulsivity factor)”.
- iv. The “anxious-depression factor correlat[ing] positively with the novelty bonus” seems like an important result, which could also be featured more prominently in the introduction. But certainly with the qualification that it wasn’t one of the initially pre-registered hypotheses
- v. There are many grammatical issues throughout the manuscript that don’t reflect a completely polished and publishable text. I’ve included a non-exhaustive list in the “Grammatical issues” section below.

2. I have some concerns about the strength of some claims made in the paper (e.g., “it allows a definite answer to the hypothesis whether ADHD symptoms are linked to value-free random exploration”). While I consider the methods to be quite rigorous, there are still some notable limits on the generalizability of the results due to the task design. In particular, the use of uniform sample variance across options may limit the extent to which more complex exploration strategies (e.g., UCB and Thompson sampling, which require representations of uncertainty) are beneficial relative to simpler “heuristics”. While it is true that the horizon manipulation influences uncertainty in one specific fashion, other recent studies have employed more direct and orthogonal manipulations of reward expectations and sample variance (Gershman, 2018; Tomov et al., 2020; Wu et al., 2021), which have been shown to influence the trade-off between random and directed exploration. The task is also somewhat limited in only having 3 different options, whereas previous work with larger decision spaces has found a strong advantage of UCB over thompson sampling in predicting participant behavior (Wu et al., 2018). Thus, the current results indicating a lack of relationship between impulsivity and uncertainty-directed exploration strategies could be a result of a floor effect, where uniform sample variance across options and only having 3 options disincentivizes more complex exploration strategies. I’m not saying there is any evidence of this alternative hypothesis in the current data, but the authors should acknowledge the limitations of generalizing from a single paradigm. I would be satisfied if the authors could water down some of these absolute/definitive claims, and mention some potential limitations of their task in the discussion.

3. Model mimicry. I am grateful that the authors have now included the inversion matrix in Supplementary Figure 6, which I had previously requested in my 2nd and 3rd round of reviews. These results confirm my suspicion that the winning model (Thompson + epsilon + eta) has substantial mimicry of the UCB + epsilon + eta model. In the best case, when the Thompson + epsilon + eta is

the best fitting model, there is only 63% probability that it was the same simulating model. A large (but unreported) probability is that the UCB + epsilon + eta model was in fact the generating model. This suggests that the winning model overlaps with and is a strong mimic of behavior generated from the UCB + epsilon + eta model. A bit of cocktail napkin math (assuming $P(\text{sim}=\text{UCB}|\text{fit}=\text{thompson}) = 1 - .63 = .37$) modifying the # of participants best fit by the elements of the inversion matrix shows that there could be a similar number of participants relying on each model in the population (Thompson + epsilon + eta: $271 * .63 = 171$; UCB + epsilon + eta : $131 * 1.37 = 179$). This represents an important limitation of the results, and should certainly be mentioned in the discussion. If I had been able to catch this earlier, I may have recommended additional comparisons of the UCB + epsilon + eta model in the subsequent dimensional analyses (e.g., checking the correlation of eta and impulsivity from the UCB model).

4. I found a lot of the explorative survey/factor analyses difficult to interpret and keep track of. Perhaps one of the SI figures or tables could be moved into the main text to summarize the findings, but that still might only be a partial solution. To reiterate my suggestion from the 1st round of reviews, I still believe that a mixed effects regression framework would provide a more robust statistical framework, but I'm also now realizing that it would provide the additional advantage of substantially improving the readability and interpretability of your dimensional analyses. Rather than reporting endless correlations, you could holistically run a single regression for each of the ADHD/impulsivity variables, directly modeling confounds such as age and IQ. For instance, $\text{BISscore} \sim \text{model parameters} * \text{horizon} + \text{age} + \text{IQ} + (1|\text{id})$ to use an LMER or BRMS formulation of a model. Then, you could simply plot the regression coefficients and see which predictors don't overlap with zero, accounting for the influence of all predictors simultaneously. I think it's fair to point out that I didn't reiterate this suggestion of using a mixed effects regression framework in subsequent reviews and that I approved of the analysis plan during both the 2nd and 3rd rounds of the review process. But in the current stage 2 manuscript, there is a new problem of a sprawling and confusing overflow of correlation analyses, which provides a new reason for wanting to revisit my previous suggestion of a mixed effects regression framework. I recognize that requesting new analyses at the 2nd stage might not be in the spirit of a pre-registered report, so I can be quite flexible here, so long as the authors find a solution to make these analyses more readable.

Minor comments

- ..."exploration itself is not a homogenous concept" could benefit from references
- More precisely, UCB is one algorithmic implementation of uncertainty-directed exploration, the latter of which is the strategy.
- "Even though such mechanisms are suboptimal," All exploration strategies are generally suboptimal, since optional solutions to the explore-exploit problem are usually only available in limited cases (e.g., infinite time horizons)
- The claim that value-free exploration is cheap is quite reasonable, but alternative explanations exist. For instance, having a more stochastic policy has been related to increased computational costs (Gershman, 2020) required to represent the policy. Previous work has also related more cost-effective decision-making with increased repetition of previous choices (Wu et al., 2021). Perhaps this is worth mentioning in the discussion
- The description of the task could be more descriptive. Describing it in terms of a 3-armed bandit influenced by the Wilson horizon task would be helpful. Also "we used our previously validated Maggie's Farm task" doesn't seem accurate, since isn't this paper the validation of that task?
- "Manipulating the number of prior samples and reward allows to capture complex exploration strategies" it's one method, but has limitations. Also this sentence is grammatically incorrect
- It's not always clear which variables are being compared in the Wilcoxon tests
- "Our findings thus match our preregistered hypotheses of an increase in different exploration forms in the long horizon" This statement could be clarified, in terms of which exploration forms. I'm also not sure if "exploration forms" is the appropriate technical term here
- "we found that the first reward (i.e., apple size) obtained in the long horizon [...] was indeed lower than the reward obtained on the first draw in the short horizon". Is this true in general or idiosyncratic of the task? I mean this in terms of whether the horizon-related changes in exploration would always

induce higher rewards, or if it is specific to the reward structure of the task

- "we examined the winning model's fitted parameters" Please name the model for clarity
 - "which supports the notion that the long horizon facilitates all exploration strategies". This is too general of a claim, since the set of potential exploration strategies is potentially infinite and not all tested in this study
 - "After having established that the general task behaviour was aligned with our previous findings". It's not very helpful to state that the task aligned with the author's previous findings, since it doesn't inform the reader of whether they are also consistent with other findings in the field, or whether the previous work by the authors is an exception
 - It's unclear how age and IQ were controlled for in the correlation results. If there is a section in the supplementary materials already, then you should point the reader to the section in the main text
 - "we found a main effect of impulsivity on the low-value bandit..." what is the DV?
 - I wonder if the "uncertainty-related distress factor" is related to learning, for instance, correlated with the Kalman filter parameters.
- "This means that value-free random exploration is being used in a skilful and goal-directed approach to drive exploration, when it is useful." Perhaps a better way to state this is that participants adapt to the demands of the task
- "We deployed a multi-armed bandit task which we have recently developed" Perhaps a more informative description of the task is as a 3-armed variant of the Wilson Horizon task.

Grammatical issues

- There are many mistakes confusing "which" and "that", and omitting or incorrectly using determinants such as "the" or "a". This doesn't hurt the interpretability of results, but also doesn't reflect a polished manuscript.
- "Whether and if so, which exploration mechanism " ?
- "exploit the recent advances in [theories of?] exploration to investigate empirically"
- "To provide a clear answer, we use a pre-registered..." this isn't exactly what a pre-registration is for
- "We thus advance on a method..." which method?
- "For this, we used the total score on the Barratt Impulsiveness Scale ... " this sentence is too long
- "it needs to be highlighted that value-free random exploration is used by all participants in a goal directed manner by increasing its utilization in the long horizon"
- "Heuristics captured similarly in 2nd best model" This is a section title in the SI that didn't really make sense

I again sign my review
Charley Wu

References

- Cogliati Dezza, I., Cleeremans, A., & Alexander, W. (2019). Should we control? The interplay between cognitive control and information integration in the resolution of the exploration-exploitation dilemma. *Journal of Experimental Psychology. General*, 148(6), 977–993.
- Gershman, S. J. (2018). Deconstructing the human algorithms for exploration. *Cognition*, 173, 34–42.
- Gershman, S. J. (2020). Origin of perseveration in the trade-off between reward and complexity. *Cognition*, 204, 104394.
- Tomov, M. S., Truong, V. Q., Hundia, R. A., & Gershman, S. J. (2020). Dissociable neural correlates of uncertainty underlie different exploration strategies. *Nature Communications*, 11(1), 2371.
- Wu, C. M., Schulz, E., Pleskac, T. J., & Speekenbrink, M. (2021). Time to explore: Adaptation of exploration under time pressure. <https://doi.org/10.31234/osf.io/dsw7q>
- Wu, C. M., Schulz, E., Speekenbrink, M., Nelson, J. D., & Meder, B. (2018). Generalization guides human exploration in vast decision spaces. *Nature Human Behaviour*, 2(12), 915–924.

REVIEWER COMMENTS

Reviewer #1 (Remarks to the Author):

Q.1.0. This study is an impressive follow-up to the pre-registered proposal that was reviewed and approved previously. The authors have used a new paradigm to explore various forms of 'exploration' and a hypothesised relationship to the dimensional trait of impulsivity. This is combined with a sophisticated computational approach following online testing of a large number of healthy volunteers who also complete a number of relevant self-report questionnaire scales, including for ADHD, anxiety and depression, also subjected to principal component analysis.

In relation to Stage 2 ms evaluation, I confirm that the authors have kept to the same protocol and experimental procedures as laid out in Stage 1. They have included the outcomes of all the pre-registered analyses. The analyses are presented exhaustively in the main text and in 32 pages of supplementary information.

The main results are clear. Their test paradigm is again validated as a means of directing various exploration strategies. The only form of exploration linked to impulsivity (tested according to the Barratt Scale) is 'random exploration'. This is related most significantly to the Barratt motor impulsivity sub-scale. The significant relationship is not especially high ($r < 0.2$) but significant at $p < 0.001$. There were no relationships with other strategies based on uncertainty driven Thompson sampling or 'novelty exploration'.

In further exploratory (i.e. not pre-registered) analyses the authors show a significant relationship between 'novelty exploration' and anxiety/depression scores. This analysis appears sound, adds to the ms and is appropriately caveated.

The authors have delivered the precise analyses that were approved previously to show an hypothesised, but previously untested, relationship between impulsivity and exploration, with possible relevance to ADHD. This novel result certainly deserves publication. They also, more tentatively show a degree of dissociation with another dimension, of anxiety/depression that further suggests possible applicability to neuropsychiatric disorders. They perhaps do need to consider possible underlying mechanism(s) for these findings- other than linking them with central noradrenergic functioning.

R.1.0: We thank the reviewer for this very positive evaluation of our Stage 2 report and for acknowledging our substantial work during the analysis. We also thank the reviewer for the detailed suggestions below and the comment regarding possible underlying mechanisms. We have now implemented all in this revised manuscript.

Specific comments

Q.1.1. Title. Is this fully justified? I do not believe any "role" for exploration has been demonstrated in impulsivity, only an association or relationship. The implication of the title is that individual differences in exploration contribute to impulsivity but this has not been shown here; only a correlation between the two.

R.1.1: We thank the reviewer for this suggestion. We did not change the title from our initial submission, but we agree that this title is not reflecting our results. We have thus changed this to 'Value-free random exploration is linked to impulsivity'.

Q.1.2. Sample. I could not find any details on the $n=580$ participants in terms of gender, age and other demographics, even in the Supplementary Material. Were age and IQ considered as co-variates?

R.1.2: We apologise for this omission and thank the reviewer for pointing this out. We have added a table with those details (Supplementary Table 1). Additionally, following reviewer's 3 suggestion (cf. Q.3.1.), we have moved the Design Table back to the main manuscript, where we detail that we controlled for age and IQ in our partial correlations.

Results: '[...] we found a significant association between the BIS total score and the ϵ -greedy parameter ($r=0.171$, $p < .001$, Figure 3a; controlling for age and IQ: $r=0.117$, $p=.005$; Hypothesis 2.1. in Design Table; cf. Methods for details and Supplementary Table 1 for demographics) [...]'

Q.1.3. The task is ingenious, though complex. Understanding it does rather depend on reference to other papers. Can its description be improved for the non-expert reader?

R.1.3: The Method section can unfortunately not be altered anymore at this stage, but we have added a 'real-life example' in the results where we go through the task again. We hope this will further clarify the task.

Results: 'Bandits carried either a lot, some or no prior information (i.e., 3, 1 or 0 initial samples; Figure 1d) and had either a standard or a low reward mean (Figure 1c). In effect, there were 4 different types of bandits: the certain-standard bandit (standard mean, 3 initial samples), the standard bandit (standard mean, 1 initial sample), the novel bandit (standard mean, 0 initial samples) and the low-value bandit (low mean). A real-life example would be having to choose between four different ice-cream flavours in some Italian city: chocolate, which you have enjoyed 3 times in the past, Toblerone, which you have enjoyed once in the past, hibiscus, which you have never tried, and spinach, which you have disliked once in the past. The decision horizon (cf. below), represents how often you will come back to this exact same ice-cream shop (e.g. the number of vacation days left, assuming you have once ice-cream per day). On each trial, 3 out of those 4 bandit types were used. In the analysis, the bandit with the highest mean reward of prior samples (either 1 or 3) is referred to as the 'high-value bandit'. [...]. To promote and assess exploration, we manipulated the number of choices per trial (i.e., decision horizon; Figure 1b), similarly to the Horizon Task²⁴. Subjects could perform either one draw, encouraging exploitation (short horizon condition), or six draws, encouraging more substantial explorative behaviour (long horizon condition) as in the latter condition, the newly gained information could subsequently be exploited. Going back the ice-cream example, knowing that you will come back to the same place many times will push you to explore different flavours (i.e. other than chocolate), as it could help guide your future choices.'

Q.1.4. The main result, that impulsivity is linked to random choice responding, sounds as though it could simply be the product of impaired general cognitive control (or even IQ- 'g'). In the Friedman and Miyake (2006) scheme, this could also explain why it is not related to cognitive flexibility per se. The authors might wish to consider their findings in terms of generally deficient fronto-executive function (in addition to speculation regarding noradrenaline, the effects of which are likely mediated by prefrontal cortex modulation).

R.1.4: We thank the reviewer for raising this point. As we were concerned about the overlap with general cognitive function, i.e. intelligence, we conducted a short IQ test with all subjects, and used this as a covariate in our analysis. This was already part of our initially submitted pre-registration. Given that our effects hold also in the light of this, we do not believe that this is simply due to an impaired general cognitive control. However, we appreciate that what we have identified might align well with some of the aspects generally termed as executive functions. In the Friedman and Miyake, we believe exploration may be most related to set shifting, which was not found to be linked with IQ. We now speculate about this in more detail in the discussion

Discussion: 'Impulsivity is a crucial construct across both general and clinical populations, but the links to specific computational mechanisms are still far from clear⁴. Based on previous theoretical^{6,12,14-17} and some experimental work^{21,23}, exploration is believed to be increased in impulsivity¹⁴ and especially in ADHD. Here, we extend these previous studies by identifying that it is value-free random exploration specifically which is increased, whilst other forms of exploration were not found to be robustly linked. This form of exploration is the computationally least demanding as it simply ignores all existing information. This is well aligned with a notion of impulsivity as 'acting without thinking', which is also captured in the motor impulsivity scale of the BIS. The latter showed a much closer association with value-free exploration than the other attentional and non-planning impulsivity BIS subscores, which capture the inability to concentrate or a lack of forethought. This form of exploration was not linked to cognitive flexibility (cf. Supplementary Information), supporting the idea that cognitive flexibility and planning abilities might be aspects of a different neurocognitive construct. We believe that this form of exploration may be more closely related to mechanisms often subsumed as set shifting executive functions⁵⁴, which also involve to flexibly shift between options. Interestingly, similar to our findings, this form of executive function does not relate closely to general cognitive functioning (i.e. IQ).

From our results, it remains unclear which brain processes exactly mediate value-free random exploration. Interestingly, we have previously found that this form of exploration is modulated by noradrenaline functioning²³, a neurotransmitter which plays an important role in impulsivity-related disorders such as ADHD^{5,7,12,17,34-38}, which could be a potential mechanism. Previous findings that linked noradrenaline functioning to what is traditionally seen as motor impulsivity support this notion⁵⁵. This form of exploration may be also related to brain circuits generally seen to be linked to noradrenaline functioning, and which are known to be relevant in set shifting (for a detailed discussion of noradrenaline and executive functions, see Chamberlain & Robbins⁵⁵). In particular, anterior cingulate cortex would be a candidate as it is heavily innervated by noradrenaline³³, and impairments in this region are linked to impaired set shifting⁵⁶. In addition, fronto-striatal loops including orbito-frontal and dorso-lateral prefrontal cortex may also be involved, as they have often been found to be related to set shifting^{6,57}. However, the precise neural processes underlying value-free random exploration needs to be examined in more detail.'

Q.1.5. It wasn't clear to me why and how enhanced novelty preference would necessarily be associated with increased levels of anxiety/depression. Intuitively, the opposite relationship might perhaps have been expected?

R.1.5: We appreciate the reviewer's surprise, we also did not expect this association to be present so clearly. However, in the context of exploration, previous studies have already linked anxiety to an increased exploration (Aberg et al., 2021; Bennett et al., 2021). This is because it is believed that anxiety/depression is associated to an uncertainty aversion (e.g. McEvoy & Mahoney 2011), and exploration essentially is a way to decrease long-term uncertainty by learning more about the immediate environment. The reason why this factor is specifically linked to novelty exploration, we believe is because this is a relatively cognitively in-expensive strategy that seems to be well deployable also under increased stress. We have now extended our discussion on this topic to detail our interpretation.

Discussion: 'Our findings suggest that those with increased anxiety-depression traits deployed the novelty-related exploration heuristic more eagerly. This is aligned with previous findings showing increased exploration in anxious subjects^{66,67}. It is believed that this is because exploration aids in overcoming long-term uncertainty, and an uncertainty aversion is commonly reported in anxiety⁶⁸. Targeting novelty in exploration might be a way to save cognitive resources as one does not need to compute expected values and uncertainties of the other options, but instead can be simply guided by what has not been encountered before. This strategy thus seems deployable even under increased stress and anxiety. Even though we have rigorously controlled for multiple comparisons, we believe an independent replication of this somewhat unexpected result would be desirable.'

Q.1.6. It should perhaps be acknowledged as a limitation of the study that the clinical significance of these interesting results needs to be tested more directly in clinically diagnosed patients.

R.1.6: We agree with the reviewer and have now added this to the revised discussion.

Discussion: 'In this registered report we demonstrate that transdiagnostic impulsivity is associated with value-free random exploration. By pre-registering and peer-reviewing our specific hypotheses using a previously-validated task^{22,23} and a well-defined dimensional approach^{39-41,60}, we were able to robustly demonstrate this specific association. Our results help understand the adaptivity of impulsivity, and, through the high prevalence of impulsivity, are important for the understanding of behaviour in the general and in clinical populations. However, future studies should investigate the validity of those effects in clinically diagnosed patient populations.'

Reviewer #2 (Remarks to the Author):

Q.2.0. The authors have carefully tested the hypotheses as laid out in their stage 1 manuscript. They clearly differentiate between planned analyses and post-hoc enquiries. Their results support their a-priori hypothesis regarding an association between impulsivity and random exploration. The finding that this association is even stronger for a latent dimension of impulsivity (obtained by conducting a factor analysis across item-level scores from multiple measures) than for individual measures is particularly nice as is the finding that the association holds both for the model parameter (ϵ -greedy) designed to capture the influence of value-free random exploration and when using frequency of choice of the low-cost option as a proxy for value-free random exploration.

R.2.0: We thank the reviewer for this positive evaluation of the structure of our report as well as our findings. We address the remaining comments below and clarify the issue raised regarding the sample size estimation.

Adherence to proposed methods.

Q.2.1. The one place where the authors deviate from their registered plans is in relation to sample size. I am not surprised that the initially pre-registered sample size was insufficient. The authors should specify whether they waited until the final sample was obtained to run the planned analyses or conducted initial analyses when the pre-registered sample size was achieved.

It is also hard to follow how many subjects were used in different analyses due to the authors' omission of degrees of freedom when reporting statistics – it would be useful if the authors could include these in any revision of the manuscript. In particular, did the authors use the full N=580 sample for the group level analyses as well as the trait / individual difference analyses reported in this manuscript? (I believe the authors had originally planned to use a smaller sample for the former than for the latter).

R.2.1: We apologise for what we believe to be a misunderstanding regarding the sample size. The Methods section can unfortunately not be altered at this stage, but we mentioned that: 'Taking all steps together, to reach at least 95% power across all measures, we collected a total sample of N=580 subjects.' This sample was used for all analysis, in accordance with the journals' registered report guidelines, which state that 'power analysis must be based on the lowest available or meaningful estimate of the effect size'. We thus believe that we completely adhered to the policy for registered reports, and we did not deviate from our initial plans and our initial power calculations.

However, we would like to apologise for not providing the degrees of freedom in all analysis. This has now been added in the manuscript (cf. Results) and in the Supplementary Material (cf. Supplementary Tables 6-11). Additionally, we have moved the initial design table back to the main manuscript, which further clarifies the overall sample size. We hope this further clarifies our approach and the adherence with the policy.

Q.2.2. Please can the authors specify if anonymized data will be made available to readers, alongside code for the models used.

R.2.2: All data and code is publicly available. We have clarified this in the 'data availability' and 'code availability' sections.

Data availability: *'Raw data (anonymized), concatenated data for figures, as well as a log file containing data collection dates can be found on: <https://github.com/MaqDub/Mfweb-data> (pilot data on: https://github.com/MaqDub/Mfweb-pilot_data).'*

Code availability: *'Code for power simulations, computational modeling and data analysis can be found on: https://github.com/MaqDub/MFweb-data_analysis.'*

Additional comments

Q.2.3. At a group-level the different forms of exploration are all elevated when the horizon is long, in line with the authors' hypotheses. The authors also point out that the first choice in the long horizon produces less points than the single choice in the short horizon but that mean performance in the long horizon condition is better than that in the short. This does suggest, as they argue, that exploration is being used to the benefit of performance. It would be helpful here if the authors could test for direct links between engagement in a given exploration strategy in the long condition and performance, using either the model parameters or/and the frequency of choices of the different options. In particular, is increased engagement in random exploration linked to better performance in the long horizon condition? Or is it that engagement in any one of the three exploration strategies improves performance, but the one selected is less important? This would be useful to know.

R.2.3: We thank the reviewer for this suggestion. We believe that this analysis is somewhat outside of the scope of this paper, which is focused on the links between exploration and impulsivity. We have previously shown (Dubois et al, 2020; 2021) that increased exploration indeed leads to improved performance in the long horizon. However, we decided to carry out this exploratory analysis in our current data set. We have thus split the data by their first choice. We indeed find that the higher outcome (when looking at the final choice relative to the best initial draw) following exploration was irrespective of the exploration strategy used (low-value bandit or novel bandit). As this is outside of the scope of our paper, we now report this in the revised supplement.

Results: *'To investigate whether the improved performance following exploration (cf. Supplementary Figure 3c) was specific to an exploration strategy, we split the data by their first choice (i.e. High-value bandit, novel bandit or low-value bandit; cf. Supplementary Figure 3d). The higher outcome (in the long run) following exploration was irrespective of the exploration strategy (i.e., novel or low-value bandit).'*

Supplementary Figure 3d:

Q.2.4. The authors note that impulsivity increases value-free random exploration across horizons. It would be useful if the authors could address this in the context of whether value-free exploration is beneficial to performance in the long horizon relative to the short horizon and whether impulsivity is linked to poorer or better performance in one or both conditions.

R.2.4: As detailed above, we do find that value-free exploration is beneficial to performance in the long horizon. We had no clear expectations about how impulsivity would be related to actual performance, as this is also heavily task-dependent and has possibly only vague resemblance of the exploration impact in real life. When exploring the performance link to impulsivity, we find a weak link in the short horizon, meaning that more impulsive subjects (as measured by the impulsiveness factor) are linked to a lower average reward ($R=-0.129$, $p=.004$). However, this effect does not remain when controlling for our common covariates ($R=-0.076$, $p=.096$). In the long horizon condition, we observe the same even weaker pattern ($R=-0.113$, $p=.013$; controlling for covariates: $R=-0.076$, $p=.095$). Given that we did not intend to investigate this association in our pre-registered report and the weak association, we decided to not implement these findings in the revised manuscript.

Q.2.5. As the authors recognize, their exploratory finding that elevated scores on their latent anxiety/depression factor is linked to increased novelty-based exploration seems rather surprising given the well-established link between anxiety and depression and intolerance of uncertainty. It would be nice if the authors could devote a little more space to discussing this finding.

R.2.5: We agree with the reviewer that our finding may seem a bit surprising at first, but our finding in fact replicates an earlier study that found similar effects (Aberg et al., 2021). We believe this is linked to resolving long-term uncertainty through exploration. Please see our response to R.1.5, which addresses this exact point in detail.

Q.2.6. It would also be helpful if the authors can address whether scores on the anxiety-depression factor and engagement in novelty-based exploration are linked to better or worse performance in each horizon condition.

R.2.6: Similarly to above, we believe that this is rather outside the scope of this registered report. However, when exploring this link, we do not find any association with anxiety/depression (factor 1) in the short horizon ($R=0.063$, $p=.164$). When looking at the long horizon, we find a very weak link ($R=-0.113$, $p=.013$) that did not survive when controlling for covariates ($R=0.075$, $p=.098$).

Q.2.7. The STAI can be broken down into both anxiety items and depression (predominantly anhedonia-related) items. It would be helpful to know whether it is primarily the latter that are loading on the anxiety/depression factor.

R.2.7: We apologise, but to the best of our knowledge (and after searching the literature), we were unable to find such a dissociation. However, because we believe there might be considerable interest in the link between the factors and the specific items, we have now added all items and their specific factor loadings to the revised supplement. This way, we believe, the readers can resolve their specific questions individually, and this may also provide a helpful basis to use this analysis for subsequent studies.

Minor:

Q.2.8. Supplementary Figure 4 does not seem to be referenced in the main text (it jumps from Fig S3 to Fig S5)

R.2.8: We apologise for this omission and have now added a reference to it in the results section.

Results: *'For an analysis of the score per trial and per block cf. Supplementary Figure 4.'*

Q.2.9. Discussion – would we expect motor impulsivity in particular to be linked to noradrenergic function?

R.2.9: We thank the reviewer for this interesting point that we did not address enough in the previous discussion. It is important to note that the term motor impulsivity is somewhat misleading as it is mostly linked to a lack of reflection. We have now extended our discussion about possible neurobiological mechanisms and speculate about the links to more general executive functions. In this context, we now also discuss this point in some more detail.

Discussion: *'This is well aligned with a notion of impulsivity as 'acting without thinking', which is also captured in the motor impulsivity scale of the BIS. The latter showed a much closer association with value-free exploration than the other attentional and non-planning impulsivity BIS subscores, which capture the inability to concentrate or a lack of forethought. This form of exploration was not linked to cognitive flexibility (cf. Supplementary Information), supporting the idea that cognitive flexibility and planning inability might be aspects of a different neurocognitive construct. We believe that this form of exploration may be more closely related to mechanisms often subsumed as set shifting executive functions⁵⁴, which also involve to flexibly shift between options. Interestingly, similar to our findings, this form of executive function does not relate closely to general cognitive functioning (i.e. IQ).*

From our results, it remains unclear which brain processes exactly mediate value-free random exploration. Interestingly, we have previously found that this form of exploration is modulated by noradrenaline functioning²³, a neurotransmitter which plays an important role in impulsivity-related disorders such as ADHD^{6,7,12,17,34-38}, which could be a potential mechanism. Previous findings that linked noradrenaline functioning to what is traditionally seen as motor impulsivity support this notion⁵⁵. This form of exploration may be also related to brain circuits generally seen to be linked to noradrenaline functioning, and which are known to be relevant in set shifting (for a detailed discussion of noradrenaline and executive functions, see Chamberlain & Robbins⁵⁵). In particular, anterior cingulate cortex would be a candidate as it is heavily innervated by noradrenaline³³, and impairments in this region are linked to impaired set shifting⁵⁶. In addition, fronto-striatal loops including orbito-frontal and dorso-lateral prefrontal cortex may also be involved, as they have often been found to be related to set shifting^{6,57}. However, the precise neural processes underlying value-free random exploration needs to be examined in more detail.'

Reviewer #3 (Remarks to the Author):

Q.3.0. This is a signed review (Chris Chambers, Cardiff University). I was a reviewer for the Stage 1 manuscript and I'm happy to see the Stage 2 completed successfully.

On a positive note, the study appears to have been undertaken closely in line with the protocol, with only some minor deviations that are transparently flagged.

R.3.0: We thank the reviewer for this positive evaluation and for acknowledging our efforts in successfully completing Stage 2.

Q.3.1. However, I would recommend some significant revisions to the reporting structure of the results. First, in my view, to ensure clarity for readers, the study design table should be returned to the main text and a column added to the right of the table indicating the outcome of each specific hypothesis.

R.3.1: We thank the reviewer for this great suggestion, which we agree will be very helpful. We have now moved the design table back to the main text (in the Results section) and have added the suggested outcome column.

Q.3.2. Second, the results section should be revised to align its structure exactly with the questions that are asked in the question column of the design table, with clear and precise references to which test outcomes are diagnostic about which specific numbered hypothesis. A nice way to do this that I have seen in other Stage 2 RRs is provide a simplified answer to each question that is then expanded to reveal the outcome of each hypothesis test. e.g. "Are exploitation and exploration horizon-modulated?" "Yes. [expand]" etc. This isn't just a stylistic suggestion -- it is intrinsic to the transparency of the RR format that the narrative reporting of confirmatory results is tightly aligned with the confirmatory design structure.

I should also emphasise that in making this recommendation I am not at all implying that the authors are deliberately obscuring the transparency of their reporting. I can see how the authors could arrive at the conclusion that the current approach reads nicely (and it does). But ensuring tight links between confirmatory plans and confirmatory outcomes is the most important priority for RRs. Ambiguity in this area is not uncommon in Stage 2 RRs due to unfamiliarity with the requirements of the format.

R.3.2: We thank the reviewer for this additional useful suggestion and apologise for the lack of clarity in the previous version of this manuscript. We indeed were unclear about how to best structure the results, and we are grateful for this guidance. We have now reorganized each result section according to their order in the design table. We have renamed the sections accordingly so that the hypothesis/question and the outcome are clearly presented and subsequently elaborated. We have also added more information after each statistic/result specifying which specific hypothesis it was tested. Please see the revised manuscripts for the detailed changes and adaptations.

Reviewer #4 (Remarks to the Author):

Q.4.0. I am very grateful to have been part of the review process for this pre-registered manuscript, and greatly enjoyed reading the 2nd stage report with the full dataset.

Given the pre-registered nature of this manuscript, I will focus my comments on the clarity and interpretation of the results, of which I have several suggestions. In addition, I have a few analysis-related comments, which are not new but reiterate previous feedback I had provided in early stages of the review process. I'm not sure if the ship has sailed to provide feedback of this nature, but I will provide them regardless, while also acknowledging the fact that I had previously approved the analysis plan. I also have no objection to this paper getting published, but would like to first see my comments addressed in a revision.

The clarity of writing and accuracy in representing the current state of theories in the field seems somewhat wanting, especially for the standards of this journal. Perhaps I should have provided more detailed feedback during the previous review stages, but I had assumed they would be improved upon during the 2nd stage of submission. I highlight some main issues here, while I include additional comments in the "Minor comments" and "Grammatical issues" sections below.

R.4.0: We thank the reviewer for their very useful comments, in the previous rounds as well as in this round, which are helping us to improve our manuscript. We have tried to implement the reviewer's comments and additional comments, wherever we think it is justified with respect to the relative restrictive rules for pre-registered reports. We are also grateful for the grammatical suggestions, which may have also arisen due to different writing styles and cultural backgrounds.

Q.4.1.1. The distinction between complex and heuristic exploration strategies lacks clarity. By some accounts, all of the different strategies can be considered heuristic. Thompson sampling is just a probability matching heuristic, while UCB is widely considered a heuristic as well. In the discussion, this distinction is clarified based on whether means and uncertainty estimates need to be represented, which would be helpful if included in the introduction. There is a remaining leap of logic however, to relate representing means and uncertainties with additional computational costs. This could be provided by reviewing previous work using experiments implementing external limitations on cognitive resources through working memory (Cogliati Dezza et al., 2019) or decision-time manipulations (Wu et al., 2021), which have found a reduction in uncertainty-directed exploration, but increases in random exploration. Connecting with this literature could improve the theoretical development of the main argument.

R.4.1.1: We thank the reviewer for those suggestions. Unfortunately, as this is a registered report, we are no longer allowed to alter the introduction or methods. However, we believe these are important contributions, and we have discussed them in detail in previous papers. To address them in this current manuscript, we have now extended our discussion to address these points.

Discussion: *'Given that value-free random exploration ignores all prior information, it begs the question why humans use this strategy in exploration. Interestingly, inducing randomness or noise has often been shown to benefit a system both in living species and in machines, supporting the importance of such strategies^{12,58-62}. Here, the main benefit of value-free random exploration is that it does not require demanding computations, allowing exploration even with restrained neural resources⁶³ or a limited ability/willingness to engage with mentally effortful computations⁴³. Exploring in a seemingly random way can be beneficial, either at an individual or a group level, in many different contexts. For example, in the case of an absence of prior knowledge⁴⁴, increased stochasticity can help to speed up learning. Additionally, in a case of imprecise or even inaccurate prior knowledge, random exploration ignores such erroneous priors and prevents them from penalizing future decision-making. Introducing stochasticity can also be beneficial in the case of dynamic environments e.g. where values can change drastically and thus agents should not rely solely on their expectations⁶². Our findings of such exploration heuristics are also well aligned with recent findings showing that limiting cognitive resources impacts the use of exploration strategies⁶⁴, and shifts in exploration strategies can be induced by applying constraints such as time pressure⁶⁵.*

Q.4.1.2. The argumentation structure of the paragraph beginning at the end of page 4 and ending on page 5 seemed a bit all over the place. This is a key section of the text that justifies the interpretation of the results and main contributions of the manuscript, which I expect could be improved quite substantially

R.4.1.2: We apologise if this paragraph appears a somewhat unclear. This most likely arose because this wording was fixed in the Stage 1 submission, and we were only able to change the tense. As we agree with this reviewer, we have now taken the liberty to make minor stylistic changes to this paragraph. We did not change any of the meaning, but simply improved readability.

Introduction: *"In this study, we put the large body of theoretical work to test and exploited the recent advances in the exploration literature to empirically investigate the link between impulsivity and exploration. Here, we investigated*

impulsivity as a broad spectrum across the general population and also with respect to a more specific ADHD-related impulsivity. We used a pre-registered, dimensional approach via large sample online testing to provide a clear answer. We advanced on a method that has recently proven the most promising for detecting meaningful mechanisms underlying psychiatric symptoms³⁹⁻⁴². We ran a big data dimensional study using our recently developed exploration task²³, which was designed to disentangle the exploration strategies that have been put forward in the literature, and which allowed us to provide an answer to whether exploration behaviour is linked to impulsivity. To determine not only whether, but also which, exploration mechanism predicts impulsivity, we made use of computational modeling. Supported by previous findings that impulsivity is associated to increased avoidance of mental effort⁴³ and that ADHD is associated to increased value-free random exploration²², we tested our hypothesis that it is specifically value-free random exploration (captured by our model parameter ϵ) which correlates with impulsivity measures (cf. Design Table), therefore determining which of these mechanisms is impaired in impulsive subjects. In addition, our data allowed us to explore how exploration impairments may be linked to other psychiatric domains (e.g. to OCD and other avoidance of uncertainty disorders^{44,45}; to depression, anxiety and anhedonia⁴⁶⁻⁴⁸) using data-driven methods.”

Q.4.1.3. Perhaps I am misunderstanding something, but effect sizes of $\eta^2 = 0.03$ is considered a small effect. A fair and balanced interpretation of these results should also qualify the relative weakness of this relationship. I also felt that later exploratory analyses relating model parameters to survey scores and latent factors could benefit from a quantification of the variance explained. The latent factor loadings could also benefit from a quantification of how much the “impulsivity-related questionnaires (BIS and ASRS) primarily loaded onto one factor (labelled as impulsivity factor)”.

R.4.1.3: We would like to highlight that the effects we observe are well within the effect sizes that are traditionally observed in such online studies, and align well with what we used as effect sizes to power this study (e.g. Gillan et al., 2016; Rollwage et al., 2018). We believe we are being most transparent about the effect size and the magnitude of the effect throughout all our analyses. We provide the effect size measures (as mentioned by the reviewer here), and we transparently show both the correlation coefficient as well as plot the actual data points. This is, we believe, more transparent than other studies that only report beta weights and bar plots that do not allow for an understanding of the association (also see the comment below). Regarding the factor loadings, we have now added several tables (cf. Supplementary Table 12-20) that list all questionnaire items and their respective factor loadings. This should make it evident how the items are linked to the factors, and also allow for further use in subsequent studies.

Q.4.1.4. The “anxious-depression factor correlat[ing] positively with the novelty bonus” seems like an important result, which could also be featured more prominently in the introduction. But certainly with the qualification that it wasn’t one of the initially pre-registered hypotheses

R.4.1.4: As mentioned above, the introduction cannot be altered at this stage, especially not to incorporate exploratory findings like these. However, we agree with reviewer that this is a potentially important finding, and we discuss it more thoroughly in the revised discussion. Please also see our responses R.1.5. and R.2.5. for more details on the matter. We also revised the results to make clear that this is an exploratory finding.

Discussion: ‘Our findings suggest that those with increased anxiety-depression traits deployed the novelty-related exploration heuristic more eagerly. This is aligned with previous findings showing increased exploration in anxious subjects^{66,67}. It is believed that this is because exploration aids in overcoming long-term uncertainty, and an uncertainty aversion is commonly reported in anxiety⁶⁸. Targeting novelty in exploration might be a way to save cognitive resources as one does not need to compute expected values and uncertainties of the other options, but instead can be simply guided by what has not been encountered before. This strategy thus seems deployable even under increased stress and anxiety. Even though we have rigorously controlled for multiple comparisons, we believe an independent replication of this somewhat unexpected result would be desirable.’

Q.4.1.5. There are many grammatical issues throughout the manuscript that don’t reflect a completely polished and publishable text. I’ve included a non-exhaustive list in the “Grammatical issues” section below.

R.4.1.5: We thank the reviewer for pointing these out, and we have thoroughly proof-read the final manuscript version.

Q.4.2. I have some concerns about the strength of some claims made in the paper (e.g., “it allows a definite answer to the hypothesis whether ADHD symptoms are linked to value-free random exploration”). While I consider the methods to be quite rigorous, there are still some notable limits on the generalizability of the results due to the task design. In particular, the use of uniform sample variance across options may limit the extent to which more complex exploration strategies (e.g., UCB and Thompson sampling, which require representations of uncertainty) are beneficial relative to simpler “heuristics”.

While it is true that the horizon manipulation influences uncertainty in one specific fashion, other recent studies have employed more direct and orthogonal manipulations of reward expectations and sample variance (Gershman, 2018; Tomov et al., 2020; Wu et al., 2021), which have been shown to influence the trade-off between random and directed exploration. The task is also somewhat limited in only having 3 different options, whereas previous work with larger decision spaces has found a strong advantage of UCB over Thompson sampling in predicting participant behavior (Wu et al., 2018). Thus, the current results indicating a lack of relationship between impulsivity and uncertainty-directed exploration strategies could be a result of a floor effect, where uniform sample variance across options and only having 3 options disincentivizes more complex exploration strategies. I'm not saying there is any evidence of this alternative hypothesis in the current data, but the authors should acknowledge the limitations of generalizing from a single paradigm. I would be satisfied if the authors could water down some of these absolute/definitive claims, and mention some potential limitations of their task in the discussion.

R.4.2: We would like to highlight that the cited sentence stems from the initial introduction and was intended to reflect the link between these measures in our task. In the revision of this paragraph (see R.4.1.2), we have now toned this down. We also would like to highlight that the complex exploration strategies are performing well in this task and we see clear evidence for a mixture between complex and more simple strategies. In fact, we even find evidence for value-free random exploration in some of the openly accessible data stemming from the papers this authors cites here (Rani Moran, personal communication). It should also be highlighted that by manipulating the prior samples we do directly speak to the complex strategies and we cannot see any evidence for a floor effect as suggested by this reviewer. However, as discussed in previous revisions and in the paper, we do agree that both complex strategies make somewhat similar predictions in this task. However, we also replicate our findings when using the UCB model, which is highly reassuring and shows the robustness of our findings (see Response R.4.3). Having said this, we agree that different tasks may show different results with respect to more complex models. We now mention this – as suggested by this reviewer – in the revised discussion section.

Discussion: *“We did not find any direct association between the trans-diagnostic factors and our complex exploration strategy (here: Thompson sampling). It needs to be noted that our task was optimised to detect the exploration heuristics. As a consequence, the complex exploration strategies make relatively similar predictions (cf. Supplementary Material). It is thus possible that in other tasks (e.g. by varying the generative bandit variance^{25,65,69}; or larger decision spaces⁷⁰), the coexistence of Thompson and UCB exploration is clearer and may be more directly linked to one of the trans-diagnostic dimensions.”*

Q.4.3. Model mimicry. I am grateful that the authors have now included the inversion matrix in Supplementary Figure 6, which I had previously requested in my 2nd and 3rd round of reviews. These results confirm my suspicion that the winning model (Thompson + epsilon + eta) has substantial mimicry of the UCB + epsilon + eta model. In the best case, when the Thompson + epsilon + eta is the best fitting model, there is only 63% probability that it was the same simulating model. A large (but unreported) probability is that the UCB + epsilon + eta model was in fact the generating model. This suggests that the winning model overlaps with and is a strong mimic of behavior generated from the UCB + epsilon + eta model. A bit of cocktail napkin math (assuming $P(\text{sim=UCB} | \text{fit=thompson}) = 1 - .63 = .37$) modifying the # of participants best fit by the elements of the inversion matrix shows that there could be a similar number of participants relying on each model in the population (Thompson + epsilon + eta: $271 * .63 = 171$; UCB + epsilon + eta: $131 * 1.37 = 179$). This represents an important limitation of the results, and should certainly be mentioned in the discussion. If I had been able to catch this earlier, I may have recommended additional comparisons of the UCB + epsilon + eta model in the subsequent dimensional analyses (e.g., checking the correlation of eta and impulsivity from the UCB model).

R.4.3: We thank the reviewer for highlighting this point. As we have pointed out before, the UCB and Thompson sampling models make relatively similar predictions in our task, which is why we simply refer to them as complex strategies across the manuscript. As mentioned before, the heuristic strategies (i.e. novelty, ϵ -greedy) yield highly similar effects when combining them either with Thompson or UCB (cf. Supplementary Figure 8). To assure this reviewer, we have now additionally analysed the link between impulsivity and value-free random exploration in the 2nd winning model (i.e. UCB-based model) – as implied here by the reviewer. As expected, we again find very similar associations. This clearly shows that our results are robust and not dependent on the complex strategy chosen. We have now added these findings to the Exploratory Analyses (Results section), and we mention the similar predictions of the models in the discussion section.

Discussion: *‘However, this is unlikely to impact the impulsivity findings presented here, as we find them irrespective of the complex strategy we are using in our computational models (cf. Supplementary Material).’*

Results: *“Value-free random exploration (captured by the ϵ -greedy parameter) was similar in the 1st winning model (Thompson+ ϵ + η) and in the 2nd winning model (UCB+ ϵ + η), both in the short horizon (Pearson correlation: $r=0.87$, $p<.001$;*

Supplementary Figure 8a) and in the long horizon ($r=0.85$, $p<.001$; Supplementary Figure 8b). Similarly, novelty exploration (captured by the novelty bonus η) was similar across both models, both in the short horizon ($r=0.71$, $p<.001$; Supplementary Figure 9a) and in the long horizon ($r=0.72$, $p<.001$; Supplementary Figure 9b).

Similar to the 1st winning model, we observed an association between value-free random exploration (i.e., ϵ -greedy parameter) as captured by 2nd winning model, and impulsivity. We observed a significant association between ϵ and the BIS total score ($R=0.155$, $p<.001$; controlling for age and IQ: $R=0.101$, $p=.015$), between ϵ and the ASRS total score ($R=0.167$, $p<.001$; controlling for age and IQ: $R=0.126$, $p=.006$), between ϵ and the impulsivity factor (cf. Figure 4; $R=0.250$, $p<.001$; controlling for age and IQ: $R=0.196$, $p<.001$). Additionally, we observed an association between novelty exploration (i.e., novelty bonus η) and the anxious-depression factor ($R=0.124$, $p=.003$; controlling for age and IQ: $R=0.101$, $p=.015$)."

Q.4.4. I found a lot of the explorative survey/factor analyses difficult to interpret and keep track of. Perhaps one of the SI figures or tables could be moved into the main text to summarize the findings, but that still might only be a partial solution. To reiterate my suggestion from the 1st round of reviews, I still believe that a mixed effects regression framework would provide a more robust statistical framework, but I'm also now realizing that it would provide the additional advantage of substantially improving the readability and interpretability of your dimensional analyses. Rather than reporting endless correlations, you could holistically run a single regression for each of the ADHD/impulsivity variables, directly modeling confounds such as age and IQ. For instance, $BISscore \sim model\ parameters * horizon + age + IQ + (1 | id)$ to use an LMER or BRMS formulation of a model. Then, you could simply plot the regression coefficients and see which predictors don't overlap with zero, accounting for the influence of all predictors simultaneously. I think it's fair to point out that I didn't reiterate this suggestion of using a mixed effects regression framework in subsequent reviews and that I approved of the analysis plan during both the 2nd and 3rd rounds of the review process. But in the current stage 2 manuscript, there is a new problem of a sprawling and confusing overflow of correlation analyses, which provides a new reason for wanting to revisit my previous suggestion of a mixed effects regression framework. I recognize that requesting new analyses at the 2nd stage might not be in the spirit of a pre-registered report, so I can be quite flexible here, so long as the authors find a solution to make these analyses more readable.

R.4.4: We thank the reviewer for these suggestions. We would like to point out that we have no indication to believe that the methods we chose would not be valid and needed to be replaced with 'more robust' methods. In fact, given that this is largely comparisons with summary statistics for each subject (e.g. parameter estimates), these methods are likely to make near identical predictions. Moreover, they would not affect the number of tests and analyses we are conducting. Having said this, we would like to point out that there is a key reason why we are unable to respond to this suggestion: We have clearly laid out our approach in the initial methods, and at this stage of the registered report, we are unable to change these methods. We have thus decided to stick with our current approach. However, we appreciate the potentially confusing nature of many comparisons and results. We have thus decided to provide an additional subfigure, which summarises all of these findings in one comprehensive 'correlation matrix' (cf. Figure 5e,f). We hope this is clear enough and provides the clarity that was hoped for. We would like to point out that we keep the original correlation plots, as they give a better impression of the effect sizes (as mentioned above R.4.1.3) and are much more informative than simply plotting regression coefficients.

Minor comments

Q.4.5.1. "exploration itself is not a homogenous concept" could benefit from references. More precisely, UCB is one algorithmic implementation of uncertainty-directed exploration, the latter of which is the strategy.

R.4.5.1: We apologise for this omission but unfortunately, we are unable to alter the introduction at this stage.

Q.4.5.2. "Even though such mechanisms are suboptimal," All exploration strategies are generally suboptimal, since optional solutions to the explore-exploit problem are usually only available in limited cases (e.g., infinite time horizons)

R.4.5.2: We apologise for the lack of clarity, we meant suboptimal compared to the other exploration strategies used by humans. Unfortunately, we are unable to alter the introduction, but we mention this in the discussion.

Discussion: *'We and others have previously shown that humans deploy a multitude of different strategies for exploration^{22-25,54,55} that all approximate an optimal exploration strategy, which is intractable in open-ended decision problems.'*

Q.4.5.3. The claim that value-free exploration is cheap is quite reasonable, but alternative explanations exist. For instance, having a more stochastic policy has been related to increased computational costs (Gershman, 2020) required to represent the policy. Previous work has also related more cost-effective decision-making with increased repetition of previous choices (Wu et al., 2021). Perhaps this is worth mentioning in the discussion.

R.4.5.3: We believe that the argument of increased computational costs for more stochastic policies aligns well with our claim, but does not affect the value-free random exploration, for which a key aspect is the omission of any information. However, we agree that there are alternative strategies that may co-exist, such as the ones mentioned here. We thus mention this in our revised discussion section.

Discussion: *'In addition, alternative exploration strategies, such as repeating one's previous choice could provide additional insight⁶⁵.'*

Q.4.5.4. The description of the task could be more descriptive. Describing it in terms of a 3-armed bandit influenced by the Wilson horizon task would be helpful. Also "we used our previously validated Maggie's Farm task" doesn't seem accurate, since isn't this paper the validation of that task?

R.4.5.4: We apologise for the lack of clarity. Maggie's Farm has previously been validated in a lab-setting (cf. Dubois et al., 2021), whereas here we were referring to the validation of the online-version. As the cited sentence is in the introduction, we are unable to change it. Regarding the task, we made added a more intuitive explanation of it, as suggested in Q.1.3. we have additionally mentioned the Wilson's Horizon Task.

Results: *'To capture different forms of exploration, we used our previously lab-validated Maggie's Farm task²³ (cf. Figure 1), which is essentially a 3-armed variant of the Horizon task²⁴.'*

Results: *'Bandits carried either a lot, some or no prior information (i.e., 3, 1 or 0 initial samples; Figure 1d) and had either a standard or a low reward mean (Figure 1c). In effect, there were 4 different types of bandits: the certain-standard bandit (standard mean, 3 initial samples), the standard bandit (standard mean, 1 initial sample), the novel bandit (standard mean, 0 initial samples) and the low-value bandit (low mean). A real-life example would be having to choose between four different ice-cream flavours in some Italian city: chocolate, which you have enjoyed 3 times in the past, Toblerone, which you have enjoyed once in the past, hibiscus, which you have never tried, and spinach, which you have disliked once in the past. The decision horizon (cf. below), represents how often you will come back to this exact same ice-cream shop (e.g. the number of vacation days left, assuming you have once ice-cream per day). On each trial, 3 out of those 4 bandit types were used. In the analysis, the bandit with the highest mean reward of prior samples (either 1 or 3) is referred to as the 'high-value bandit'. [...]. To promote and assess exploration, we manipulated the number of choices per trial (i.e., decision horizon; Figure 1b), similarly to the Horizon Task²⁴. Subjects could perform either one draw, encouraging exploitation (short horizon condition), or six draws, encouraging more substantial explorative behaviour (long horizon condition) as in the latter condition, the newly gained information could subsequently be exploited. Going back to the ice-cream example, knowing that you will come back to the same place many times will push you to explore different flavours (i.e. other than chocolate), as it could help guide your future choices.'*

Q.4.5.5. “Manipulating the number of prior samples and reward allows to capture complex exploration strategies” it’s one method, but has limitations. Also this sentence is grammatically incorrect.

R.4.5.5: We have changed the sentence, and we also detail the limitations and alternative approached in the revised discussion (see R.4.2 for further details).

Results: *‘We manipulated the number of prior samples and the rewards of the bandit. This allowed us to capture complex exploration strategies, because they take expected values and the uncertainty of the expected values into account.’*

Q.4.5.6. It’s not always clear which variables are being compared in the Wilcoxon tests

R.4.5.6: We apologise for this lack of clarity. We now explicitly mention the parameter being analysed when it was not clear, and additionally refer the specific test in the Design Table (as suggested in Q.3.2).

Results: *‘We found, as hypothesised, that subjects chose bandits with a lower expected value (computed as the mean of the bandits’ initial samples) in the long horizon compared to the short horizon, a sign of increased exploration in the condition where they could benefit from it (expected-value of chosen bandit: Wilcoxon signed-rank test: $V=110057$, $p<.001$, Wilcoxon effect size: $r=0.265$; Supplementary Figure 3a). [...] We found that this exploration was goal-directed, with subjects choosing bandits they knew less about (lower number of initial samples, i.e., more informative) in the long horizon (number of initial samples of chosen bandit: $V=160109.5$, $p<.001$, $r=0.796$; Supplementary Figure 3b). [...]. In addition, we found that the prior variance, capturing complex, uncertainty-related exploration was also increased in the long horizon (prior variance fitted parameter: $V=54537$, $p<.001$, $r=.306$; Figure 2b), which supports the notion that the long horizon facilitates all exploration strategies.’*

Q.4.5.7. “Our findings thus match our preregistered hypotheses of an increase in different exploration forms in the long horizon” This statement could be clarified, in terms of which exploration forms. I’m also not sure if “exploration forms” is the appropriate technical term here

R.4.5.7: We agree with the reviewer and have changed that.

Results: *‘Our findings thus match our preregistered hypotheses of an increase in exploration in the long horizon.’*

Q.4.5.8. “, we found that the first reward (i.e., apple size) obtained in the long horizon [...] was indeed lower than the reward obtained on the first draw in the short horizon”. Is this true in general or idiosyncratic of the task? I mean this in terms of whether the horizon-related changes in exploration would always induce higher rewards, or if it is specific to the reward structure of the task

R.4.5.8: Subjects played the same task in the short and in the long horizon, so it is not related to the reward structure of the task, but to the strategy the participants used. This has been clarified.

Results: *‘In alignment with the above analyses, we observed that subjects obtained a lower reward (i.e., apple size) in the first draw of the long horizon (i.e., when we observed increased exploration) compared to the single draw in the short horizon ($V=131612$, $p<.001$, $r=0.53$; Supplementary Figure 3c; Hypothesis 1.2.a. in Design Table).’*

Q.4.5.9. “we examined the winning model’s fitted parameters” Please name the model for clarity

R.4.5.9: We thank the reviewer for this suggestion, which we have now implemented.

Results: *‘Next, we were interested to assess which exploration strategies were deployed more in the long horizon, which is why we examined the winning model’s (Thompson+ η + ϵ) fitted parameters.’*

Q.4.5.10. “which supports the notion that the long horizon facilitates all exploration strategies”. This is too general of a claim, since the set of potential exploration strategies is potentially infinite and not all tested in this study.

R.4.5.10: This has now been reworded.

Results: *'In addition, we found that the prior variance, capturing complex, uncertainty-related exploration was also increased in the long horizon (prior variance fitted parameter: $V=54537$, $p<.001$, $r=.306$; Figure 2b), which supports the notion that the long horizon facilitates the exploration strategies we assessed in this task.'*

Q.4.5.11. "After having established that the general task behaviour was aligned with our previous findings". It's not very helpful to state that the task aligned with the author's previous findings, since it doesn't inform the reader of whether they are also consistent with other findings in the field, or whether the previous work by the authors is an exception.

R.4.5.11: This sentence was simply formulated to provide a transition between two different analysis sections in the results section. The previous 3 pages were detailing and embedding these results. We have now changed this sentence to avoid any potential confusion

Results: *'Next, we looked at the link between impulsivity and exploration.'*

Q.4.5.12. It's unclear how age and IQ were controlled for in the correlation results. If there is a section in the supplementary materials already, then you should point the reader to the section in the main text

R.4.5.12: This is detailed in the Methods section which we now refer the reader to. We also mention this in the Design Table, which we have now added to the main results section.

Results: *'[...] we found a significant association between the BIS total score and the ϵ -greedy parameter ($r=0.171$, $p<.001$, Figure 3a; controlling for age and IQ: $r=0.117$, $p=.005$; Hypothesis 2.1. in Design Table; cf. Methods for details) [...]'.*

Methods: *'For the correlations, we used the Pearson correlation coefficient and we performed both a bivariate correlation as well as a partial correlation to control for age and IQ²². The IQ score was computed as the sum of the correct answers on the ICAR sample test⁶⁹.'*

Q.4.5.13. "we found a main effect of impulsivity on the low-value bandit..." what is the DV?

R.4.5.13: As detailed in the methods section, the dependent variable was the % of low-value bandit choices. This has now been clarified.

Results: *'In line with those results, when performing a repeated-measures ANOVA with the horizon as within-subject factor, we found a main effect of impulsivity on how frequently the low-value bandit was chosen (BIS main effect: $F(1,578)=18.103$, $p<.001$, partial eta squared $\eta^2=0.03$; horizon main effect: $F(1,578)=113.614$, $p<.001$, $\eta^2=0.164$; BIS-by-horizon interaction: $F(1,578)=0.773$, $p=.380$, $\eta^2=0.001$ '.*

Methods: *'Additionally, we also performed repeated-measures ANOVAs with within factor horizon and a between subjects variable [impulsivity/ADHD-symptoms] to assess these effects further.'*

Q.4.5.14. I wonder if the "uncertainty-related distress factor" is related to learning, for instance, correlated with the Kalman filter parameters.

R.4.5.14: We did not observe any association between this factor and either σ_0 or Q_0 , which are the relevant parameters for the Kalman filter. We report this in our revised results section.

Results: *'We did not observe any significant association (after correcting for multiple comparisons) in neither in the model parameters (ϵ : $r=0.107$, $p_{cor}=.119$, $p_{unc}=.01$; accounting for age and IQ: $r=0.072$, $p_{cor}=.997$, $p_{unc}=.083$; η : $r=0.001$, $p_{cor}=1$, $p_{unc}=.99$; accounting for age and IQ: $r=-0.002$, $p_{cor}=1$, $p_{unc}=.97$; σ_0 : $r=-0.006$, $p_{cor}=1$, $p_{unc}=.877$; accounting for age and IQ: $r=0.009$, $p_{cor}=1$, $p_{unc}=.821$; Q_0 : $r=0.054$, $p_{cor}=1$, $p_{unc}=.197$; accounting for age and IQ: $r=0.048$, $p_{cor}=1$, $p_{unc}=.251$ '.*

Q.4.5.15. "This means that value-free random exploration is being used in a skilful and goal-directed approach to drive exploration, when it is useful." Perhaps a better way to state this is that participants adapt to the demands of the task

R.4.5.15: We thank the reviewer for this suggestion, which we have now implemented.

Discussion: *'This means that participants adapt their usage of value-free random exploration to the demands of the task.'*

Q.4.5.16. “We deployed a multi-armed bandit task which we have recently developed” Perhaps a more informative description of the task is as a 3-armed variant of the Wilson Horizon task.

R.4.5.16: We unfortunately are not able to change the Methods at this stage, but we have implemented this in the results.

Results: *‘To capture different forms of exploration, we used our previously lab-validated Maggie’s Farm task²³ (cf. Figure 1), which is essentially a 3-armed variant of the Horizon task²⁴.’*

Grammatical issues

Q.4.6.1. There are many mistakes confusing “which” and “that”, and omitting or incorrectly using determinants such as “the” or “a”. This doesn’t hurt the interpretability of results, but also doesn’t reflect a polished manuscript.

R.4.6.1: We have carefully revised the manuscript and asked a native English speaker to help resolve these points.

Q.4.6.2 “Whether and if so, which exploration mechanism “ ?

Q.4.6.3. “ exploit the recent advances in [theories of?] exploration to investigate empirically”

Q.4.6.4. “To provide a clear answer, we use a pre-registered...” this isn’t exactly what a pre-registration is for

Q.4.6.5. “We thus advance on a method...” which method?

R.4.6.2-5: We apologise for the imperfect wording, we have made minimally invasive adjustments as these sentences are from the introduction (see R.4.1.2).

Q.4.6.6. “For this, we used the total score on the Barratt Impulsiveness Scale ... “ this sentence is too long

R.4.6.6: We have now split it into 2 shorter sentences.

Results: *‘For this, we used the total score on the Barratt Impulsiveness Scale (BIS), the most commonly administered self-report measure for impulsiveness⁴⁹. We assessed its link to the model parameter and behavioural measure of value-free random exploration, the ϵ -greedy parameter and the low-value bandit picking frequency.’*

Q.4.6.7. “it needs to be highlighted that value-free random exploration is used by all participants in a goal directed manner by increasing its utilization in the long horizon”

R.4.6.7: This has now been reformulated.

Discussion: *‘Importantly, value-free random exploration is used by all participants in a goal-directed manner (i.e., they used it more when exploration was beneficial).’*

Q.4.6.8. “Heuristics captured similarly in 2nd best model” This is a section title in the SI that didn’t really make sense

R.4.6.8: This now been changed.

Supplementary Material: *‘Analysis of 2nd winning model’.*

I again sign my review

Charley Wu

References

Cogliati Dezza, I., Cleeremans, A., & Alexander, W. (2019). Should we control? The interplay between cognitive control and information integration in the resolution of the exploration-exploitation dilemma. *Journal of Experimental Psychology. General*, 148(6), 977–993.

Gershman, S. J. (2018). Deconstructing the human algorithms for exploration. *Cognition*, 173, 34–42.

Gershman, S. J. (2020). Origin of perseveration in the trade-off between reward and complexity. *Cognition*, 204, 104394.

Tomov, M. S., Truong, V. Q., Hundia, R. A., & Gershman, S. J. (2020). Dissociable neural correlates of uncertainty underlie different exploration strategies. *Nature Communications*, 11(1), 2371.

Wu, C. M., Schulz, E., Pleskac, T. J., & Speekenbrink, M. (2021). Time to explore: Adaptation of exploration under time pressure. <https://doi.org/10.31234/osf.io/dsw7q>

Wu, C. M., Schulz, E., Speekenbrink, M., Nelson, J. D., & Meder, B. (2018). Generalization guides human exploration in vast decision spaces. *Nature Human Behaviour*, 2(12), 915–924.

Reviewers' Comments:

Reviewer #1 (Remarks to the Author):

The authors have generally responded very well to my queries and I think the ms should be almost ready for publication. However, there is one passage of the revised Discussion on p25 which is confusing. They state on the one hand that this form of random exploration was not linked to cognitive flexibility (cf supplementary information)... supporting the idea that cognitive flexibility and planning inabilities might be aspects of a different neurocognitive construct (Nb I assume here that is meant different from random exploration).

However, they go on to say that random exploration may be more closely related to mechanisms often subsumed as set-shifting executive functions which also involve flexible shifting between options.

This is very problematic as the Friedman and Miyake construct of cognitive flexibility to which they refer is based largely on measuring set-shifting - so there is a non seq here somewhere which may relate to the authors' own measure of cognitive flexibility reported in the supplementary materials. Many readers will be confused by this apparent discrepancy. This section therefore needs clarification as well as stylistic polishing. ("which also involve to flexibly shift"needs to be simplified to "which also involves flexible shifting")

Reviewer #2 (Remarks to the Author):

The authors have satisfactorily addressed my concerns

Reviewer #3 (Remarks to the Author):

This is a very thorough revision that perfectly addresses all of my comments. Congratulations on a fine RR.

Reviewer #4 (Remarks to the Author):

Thank you for the detailed responses to my previous comments. I also deeply apologize if any suggestions seemed targeted towards parts of the manuscript that are not allowed to be altered. However, I feel the clarifications added to the discussion have addressed my concerns and have greatly improved the theoretical clarity of the paper. I would be happy to accept this manuscript for publication, barring one minor request.

I think the statement about the winning model being "well distinguishable from other models" with "relatively high confidence regarding its generative origins" should be phrased differently, to accurately reflect the non-trivial degree of model mimicry. This should be mentioned as a potential limitation of this paradigm for disentangling the different exploration strategies being tested here. However, it does not need to question the validity of the main conclusion of the paper, since the authors have shown in their rebuttal that the parameter estimates for epsilon also hold for the other model. Thus, an honest reckoning of potential limitations of the paradigm (in distinguishing the great number of different exploration strategies being tested), while still acknowledging the robustness of the main results (across alternative models) seems like a fair resolution.

REVIEWER COMMENTS

Reviewer #1 (Remarks to the Author)

Q.1.0. The authors have generally responded very well to my queries and I think the ms should be almost ready for publication. However, there is one passage of the revised Discussion on p25 which is confusing. They state on the one hand that this form of random exploration was not linked to cognitive flexibility (cf supplementary information)... supporting the idea that cognitive flexibility and planning inabilities might be aspects of a different neurocognitive construct (Nb I assume here that is meant different from random exploration).

However, they go on to say that random exploration may be more closely related to mechanisms often subsumed as set-shifting executive functions which also involve flexible shifting between options.

This is very problematic as the Friedman and Miyake construct of cognitive flexibility to which they refer is based largely on measuring set-shifting - so there is a non seq here somewhere which may relate to the authors' own measure of cognitive flexibility reported in the supplementary materials. Many readers will be confused by this apparent discrepancy. This section therefore needs clarification as well as stylistic polishing. ("which also involve to flexibly shift" needs to be simplified to "which also involves flexible shifting")

R.1.0: We thank the reviewer for pointing out this inconsistency, which must have emerged over the different revisions. We agree that this section was not coherent and we have changed this paragraph (and the further discussion in the subsequent paragraph) accordingly.

Discussion: 'This form of exploration is the computationally least demanding as it simply ignores all existing information. This is well aligned with a notion of impulsivity as 'acting without thinking', which is also captured in the motor impulsivity scale of the BIS. The latter showed a much closer association with value-free exploration than the other attentional and non-planning impulsivity BIS subscores, which capture the inability to concentrate or a lack of forethought. We did not find a significant association between this form of exploration and a measure of global cognitive flexibility (cf. Supplementary Information), supporting the idea that cognitive flexibility and planning inabilities might be different neurocognitive constructs. However, it would be interesting to investigate whether value-free random exploration is related to more specific tasks, such as set shifting, inhibition or other decision making and learning tasks, given that cognitive flexibility in itself is a relatively heterogeneous construct⁵⁵. [...] This form of exploration may be also related to brain circuits generally seen to be linked to noradrenaline functioning (for a detailed discussion of noradrenaline and executive functions, see Chamberlain & Robbins⁵⁶). In particular, anterior cingulate cortex would be a candidate as it is heavily innervated by noradrenaline and linked and linked to similar exploratory behaviour³³. In addition, fronto-striatal loops including orbito-frontal and dorso-lateral prefrontal cortex may also be involved, as they have often been found to be involved in tasks that are modulated by noradrenaline related to set shifting^{56,57}. However, the precise neural processes underlying value-free random exploration needs to be examined in more detail.'

Reviewer #2 (Remarks to the Author):

Q.2.0. The authors have satisfactorily addressed my concerns

R.2.0: We thank the reviewer for their positive evaluation of our revisions.

Reviewer #3 (Remarks to the Author):

Q.3.0. This is a very thorough revision that perfectly addresses all of my comments. Congratulations on a fine RR.

R.3.0: We thank the reviewer for this very positive feedback and for their help during the whole process.

Reviewer #4 (Remarks to the Author):

Q.4.0. Thank you for the detailed responses to my previous comments. I also deeply apologize if any suggestions seemed targeted towards parts of the manuscript that are not allowed to be altered. However, I feel the clarifications added to the discussion have addressed my concerns and have greatly improved the theoretical clarity of the paper. I would be happy to accept this manuscript for publication, barring one minor request.

I think the statement about the winning model being “well distinguishable from other models” with “relatively high confidence regarding its generative origins” should be phrased differently, to accurately reflect the non-trivial degree of model mimicry. This should be mentioned as a potential limitation of this paradigm for disentangling the different exploration strategies being tested here. However, it does not need to question the validity of the main conclusion of the paper, since the authors have shown in their rebuttal that the parameter estimates for epsilon also hold for the other model. Thus, an honest reckoning of potential limitations of the paradigm (in distinguishing the great number of different exploration strategies being tested), while still acknowledging the robustness of the main results (across alternative models) seems like a fair resolution.

R.4.0: We thank for the reviewer for their positive evaluation of our previous revisions and for the current suggestion which we have now implemented in the discussion. We hope that the revised manuscript now strikes a good balance between the shortcomings of the task and the robustness of the findings.

Discussion: ‘The winning model, combining complex Thompson with novelty (η) and value-free random (ϵ) exploration, was not entirely distinguishable from the 2nd winning model, combining complex UCB with novelty and value-free random exploration, but was well distinguishable from other models (cf. confusion matrix, Supplementary Figure 6b) with relatively high confidence regarding its generative origins (cf. inversion matrix, Supplementary Figure 6c). This suggests that the two complex exploration strategies make similar predictions in our task, preventing us to disentangle them properly. However, we capture similar amounts of value-free random exploration, irrespective of the complex model used, demonstrating the robustness of our result. Our results therefore show that subjects supplemented complex strategies (UCB or Thompson sampling) with two heuristic strategies. Given that we find an association between value free exploration and impulsivity irrespective of the complex model used, this does not impact the conclusions in the given study.’